# DIFFUSION POLICY POLICY OPTIMIZATION

Allen Z. Ren[1], Justin Lidard[1], Lars L. Ankile[2,3], Anthony Simeonov[3],
Pulkit Agrawal[3], Anirudha Majumdar[1], Benjamin Burchfiel[4], Hongkai Dai[4], Max Simchowitz[3,5]

[1]Princeton University  [2]Harvard University  [3]Massachusetts Institute of Technology
[4]Toyota Research Institute  [5]Carnegie Mellon University

## ABSTRACT

We introduce *Diffusion Policy Policy Optimization*, **DPPO**, an algorithmic framework including best practices for fine-tuning diffusion-based policies (e.g. Diffusion Policy (Chi et al., 2024b)) in continuous control and robot learning tasks using the policy gradient (PG) method from reinforcement learning (RL). PG methods are ubiquitous in training RL policies with other policy parameterizations; nevertheless, they had been conjectured to be less efficient for diffusion-based policies. Surprisingly, we show that **DPPO** achieves the strongest overall performance and efficiency for fine-tuning in common benchmarks compared to other RL methods for diffusion-based policies and also compared to PG fine-tuning of other policy parameterizations. Through experimental investigation, we find that **DPPO** takes advantage of unique synergies between RL fine-tuning and the diffusion parameterization, leading to structured and on-manifold exploration, stable training, and strong policy robustness. We further demonstrate the strengths of **DPPO** in a range of realistic settings, including simulated robotic tasks with pixel observations, and via zero-shot deployment of simulation-trained policies on robot hardware in a long-horizon, multi-stage manipulation task. Webpage with code: **diffusion-ppo.github.io**.

## 1 INTRODUCTION

Large-scale pre-training with additional fine-tuning has become a ubiquitous pipeline in the development of language and image foundation models (Brown et al., 2020; Radford et al., 2021; Ouyang et al., 2022; Ruiz et al., 2023). Though behavior cloning with expert data (Pomerleau, 1988) is rapidly emerging as dominant paradigm for pre-training *robot policies* (Florence et al., 2019; 2022; Zhao et al., 2023; Lee et al., 2024; Fu et al., 2024), their performance can be suboptimal (Osa et al., 2018) due to expert data being suboptimal or expert data exhibiting limited coverage of possible environment conditions. As robot policies entail interaction with their environment, reinforcement learning (RL) (Sutton and Barto, 2018) is a natural candidate for further optimizing their performance beyond the limits of demonstration data. However, RL fine-tuning can be nuanced for pre-trained policies parameterized as diffusion models (Ho et al., 2020), which have emerged as a leading parameterization for action policies (Chi et al., 2024b; Reuss et al., 2023; Pearce et al., 2023), due in large part to their high training stability and ability to represent complex distributions (Rombach et al., 2022; Poole et al., 2022; Kong et al., 2020; Ho et al., 2022).

**Contribution 1** (**DPPO**). We introduce *Diffusion Policy Policy Optimization* (**DPPO**), a generic framework as well as a set of carefully chosen design decisions for fine-tuning a diffusion-based robot learning policy via popular policy gradient methods (Sutton et al., 1999; Schulman et al., 2017) in reinforcement learning.

The literature has already studied improving/fine-tuning diffusion-based policies (*Diffusion Policy*) using RL (Psenka et al., 2023; Wang et al., 2022; Hansen-Estruch et al., 2023), and has applied policy gradient (PG) to fine-tuning non-interactive applications of diffusion models such as text-to-image generation (Black et al., 2023; Clark et al., 2023; Fan et al., 2024). Yet PG methods have been believed to be inefficient in training Diffusion Policy for continuous control tasks (Psenka et al., 2023; Yang et al., 2023). On the contrary, we show that for a Diffusion Policy pre-trained from expert demonstrations, our methodology for *fine-tuning* via PG updates yields robust, high-performing policies with favorable training behavior.

Figure 1: We introduce **DPPO**, *Diffusion Policy Policy Optimization*, that fine-tunes pre-trained Diffusion Policy using policy gradient. Extensive experiments in simulation and hardware show **DPPO** affords structured exploration and training stability during policy fine-tuning, and the final policy exhibits strong robustness and generalization. **DPPO** improves policy performance across benchmarks, including ones with pixel observations and with long-horizon rollouts that have been very challenging to solve using previous RL methods.

**Contribution 2** *(Demonstration of **DPPO**'s Performance).* We show that for *fine-tuning* a pre-trained Diffusion Policy, **DPPO** shows strong training stability across tasks and marked improvements in final performance in challenging robotic tasks in comparison to a range of alternatives, including those based on off-policy Q-learning (Wang et al., 2022; Hansen-Estruch et al., 2023; Yang et al., 2023; Psenka et al., 2023) and weighted regression (Peng et al., 2019; Peters and Schaal, 2007; Kang et al., 2024), other demo-augmented RL methods (Ball et al., 2023; Nakamoto et al., 2024; Hu et al., 2023), as well as common policy parameterizations such as Gaussian and Gaussian Mixture models.

The above finding might be surprising because PG methods do not appear to take advantage of the unique capabilities of diffusion sampling (e.g., guidance (Janner et al., 2022; Ajay et al., 2023)). Through careful investigative experimentation, however, we find a **unique synergy** between RL fine-tuning and diffusion-based policies.

**Contribution 3** *(Understanding the mechanism of **DPPO**'s success).* We complement our results with numerous investigative experiments that provide insight into the mechanisms behind **DPPO**'s strong performance. Compared to other common policy parameterizations, we provide evidence that **DPPO** engages in *structured exploration* that takes better advantage of the "manifold" of training data, and finds policies that exhibit greater robustness to perturbation.

Through ablations, we further show that our design decisions overcome the speculated limitation of PG methods for fine-tuning Diffusion Policy. Finally, to justify the broad utility of **DPPO**, we verify its efficacy across both simulated and real environments, and in situations when either ground-truth states or pixels are given to the policy as input.

**Contribution 4** *(Tackling challenging robotic tasks and settings).* We show **DPPO** is effective in challenging robotic and control settings, including pixel observations and long-horizon manipulation tasks with sparse reward. We deploy a policy trained in simulation via **DPPO** on real hardware in zero-shot, which exhibits smoother behavior than the baseline and transfers better to the real-world.

## 2 RELATED WORK

**Policy optimization and its application to robotics.** Policy optimization methods update an explicit representation of an RL policy — typically parameterized by a neural network — by taking gradients through action likelihoods. Following the seminal policy gradient (PG) method (Williams, 1992; Sutton et al., 1999), there have been a range of algorithms that further improve training stability and sample efficiency such as DDPG (Lillicrap et al., 2015) and PPO (Schulman et al., 2015). PG methods have been broadly effective in training robot policies (Andrychowicz et al., 2020; Hwangbo et al., 2019; Kaufmann et al., 2023; Chen et al., 2022b), largely due to their training stability with high-dimensional continuous action spaces, as well as their favorable scaling with parallelized simulated environments. Given the challenges of from-scratch exploration in long-horizon tasks, PG has seen great success in fine-tuning a baseline policy trained from demonstrations (Rajeswaran et al., 2017; Torne et al., 2024; Peng et al., 2021). Our experiments find **DPPO** performing on-policy PG often achieves stronger final performance in manipulation tasks, especially ones with long horizon

and high-dimensional action space, than off-policy Q-learning methods (Ball et al., 2023; Nakamoto et al., 2024; Hu et al., 2023).

**Learning and improving diffusion-based policies.** Diffusion-based policies (Chi et al., 2024b; Reuss et al., 2023; Ankile et al., 2024a; Ze et al.; Wang et al., 2024a; Sridhar et al., 2023; Pearce et al., 2023) have shown recent success in robotics and decision making applications. Most typically, these policies are trained from human demonstrations through a supervised objective, and enjoy both high training stability and strong performance in modeling complex and multi-modal trajectory distributions. As demonstration data are often limited and/or suboptimal, there have been many approaches proposed to improve the performance of diffusion-based policies. One popular approach has been to guide the diffusion denoising process using objectives such as reward signal or goal conditioning (Janner et al., 2022; Ajay et al., 2023; Venkatraman et al., 2023; Chen et al., 2024). More recent work has explored techniques including Q-learning and weighted regression, either from purely offline estimation (Chen et al., 2022a; Wang et al., 2022; Ding and Jin, 2023), and/or with online interaction (Hansen-Estruch et al., 2023; Psenka et al., 2023; Yang et al., 2023).

**Policy gradient through diffusion models.** RL techniques have been used to fine-tune diffusion models such as ones for text-to-image generation (Fan and Lee, 2023; Fan et al., 2024; Black et al., 2023; Wallace et al., 2023). Black et al. (2023) treat the denoising process as an MDP and apply PPO updates. We build upon these earlier findings by embedding the denoising MDP into the environmental MDP of the dynamics in control tasks, forming a two-layer "Diffusion Policy MDP". Though Psenka et al. (2023) have already shown how PG can be taken through Diffusion Policy by propagating PG through both MDPs, they conjecture that it is likely to be ineffective due to large action variance caused by the increased effective horizon induced from the denoising steps. Our results contravene this supposition for diffusion-based policies in the fine-tuning setting.

## 3 PRELIMINARIES

**Markov Decision Process.** We consider a *Markov Decision Process* (MDP) $\mathcal{M}_{\text{ENV}} := (\mathcal{S}, \mathcal{A}, P_0, P, R)$ with states $s \in \mathcal{S}$, actions $a \in \mathcal{A}$, initial state distribution $P_0$, transition probabilities $P$, and reward $R$. At each timestep $t$, the agent (e.g., robot) observes the state $s_t \in \mathcal{S}$, takes an action $a_t \sim \pi(a_t \mid s_t) \in \mathcal{A}$, transitions to the next state $s_{t+1}$ according to $s_{t+1} \sim P(s_{t+1} \mid s_t, a_t)$ while receiving the reward $R(s_t, a_t)$[1]. Fixing the MDP $\mathcal{M}_{\text{ENV}}$, we let $\mathbb{E}^\pi$ (resp. $\mathbb{P}^\pi$) denote the expectation (resp. probability distribution) over trajectories $(s_0, a_0, ..., s_T, a_T)$ with length $T+1$, with initial state distribution $s_0 \sim P_0$ and transition operator $P$. We aim to train a policy to optimize the cumulative reward, discounted by a function $\gamma(\cdot)$: $\mathcal{J}(\pi_\theta) = \mathbb{E}^{\pi_\theta, P_0}[\sum_{t \geq 0} \gamma(t) R(s_t, a_t)]$

**Policy optimization.** The *policy gradient method* (e.g., REINFORCE (Williams, 1992)) allows for improving policy performance by approximating the gradient of this objective w.r.t. the policy parameters: $\nabla_\theta \mathcal{J}(\pi_\theta) = \mathbb{E}^{\pi_\theta, P_0}[\sum_{t \geq 0} \nabla_\theta \log \pi_\theta(a_t|s_t) r_t(s_t, a_t)]$, $r_t(s_t, a_t) := \sum_{\tau \geq t} \gamma(\tau) R(s_\tau, a_\tau)$ where $r_t$ is the discounted cumulative future reward from time $t$, $\gamma$ is the discount factor that depends on the time-step, and $\nabla_\theta \log \pi_\theta(a_t|s_t)$ denotes the gradient of the logarithm of the *likelihood* of $a_t \mid s_t$. To reduce the variance of the gradient estimation, a state-value function $\hat{V}^{\pi_\theta}(s_t)$ can be learned to approximate $\mathbb{E}[r_t]$. The estimated advantage function $\hat{A}^{\pi_\theta}(s_t, a_t) := r_t(s_t, a_t) - \hat{V}^{\pi_\theta}(s_t)$ substitutes $r_t(s_t, a_t)$.

**Diffusion models.** A denoising diffusion probabilistic model (DDPM) (Nichol and Dhariwal, 2021; Ho et al., 2020; Sohl-Dickstein et al., 2015) represents a continuous-valued data distribution $p(\cdot) = p(x^0)$ as the reverse denoising process of a forward noising process $q(x^k|x^{k-1})$ that iteratively adds Gaussian noise to the data. The reverse process is parameterized by a neural network $\varepsilon_\theta(x_k, k)$, predicting the added noise $\varepsilon$ that converts $x_0$ to $x_k$ (Ho et al., 2020). Sampling starts with a random sample $x^K \sim \mathcal{N}(0, \mathrm{I})$ and iteratively generates the denoised sample:

$$x^{k-1} \sim p_\theta(x^{k-1}|x^k) := \mathcal{N}(x^{k-1}; \mu_k(x^k, \varepsilon_\theta(x^k, k)), \sigma_k^2 \mathrm{I}). \tag{3.1}$$

Above, $\mu_k(\cdot)$ is a fixed function, independent of $\theta$, that maps $x^k$ and predicted $\varepsilon_\theta$ to the next mean, and $\sigma_k^2$ is a variance term that abides by a fixed schedule from $k = 1, \ldots, K$. We refer the reader to Chan (2024) for an in-depth survey.

---

[1] For simplicity, we overload $R(\cdot, \cdot)$ to denote both the random variable reward and its distribution.

**Diffusion models as policies.** *Diffusion Policy* (DP; see Chi et al. (2024b)) is a policy $\pi_\theta$ parameterized by a DDPM which takes in $s$ as a conditioning argument, and parameterizes $p_\theta(a^{k-1} \mid a^k, s)$ as in (3.1). DPs can be trained via behavior cloning by fitting the conditional noise prediction $\varepsilon_\theta(a^k, s, k)$ to predict the added noise. Notice that unlike more standard policy parameterizations such as unimodal Gaussian policies, DPs do not maintain an explicit likelihood of $p_\theta(a^0 \mid s)$. In this work, we adopt the common practice of training DPs to predict an **action chunk** — a sequence of actions a few time steps (denoted $T_p$) into the future — to promote temporal consistency. For fair comparison, our diffusion and non-diffusion baselines use the same chunk size.

## 4  **DPPO**: Diffusion Policy Policy Optimization

**Directly applying policy gradient does not work.** At first glance, applying policy gradient to a new policy parameterization should be straightforward — the update simply uses the likelihood of the sampled action under the policy, $\pi_\theta(a_t|s_t)$. However, diffusion as a multi-step denoising process introduces challenges: while the intermediate denoised action likelihood $\pi_\theta(a_t^k|s_t, a_t^{k+1})$ can be readily evaluated, the likelihood of the final denoised action $\pi_\theta(a_t^{k=0}|s_t)$ can only be approximated (Song et al., 2020b). In Appendix D.6 we show that differentiating through the approximated likelihood leads to training instability and fails to optimize the pre-trained policies. Next we take the perspective of "denoising process as MDP" to address the issue.

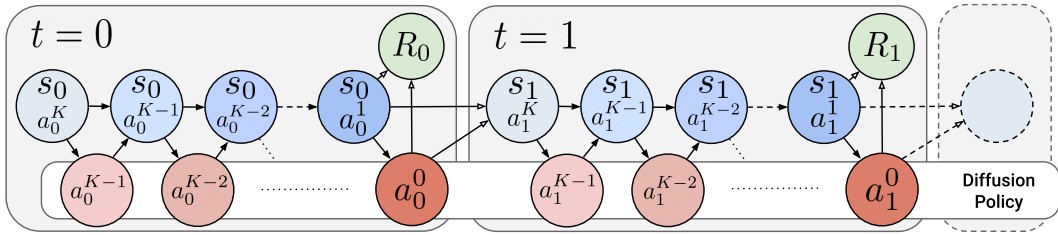

Figure 2: We treat the denoising process in Diffusion Policy as an MDP, and the whole environment episode can be considered as a chain of such MDPs. Now the entire chain ("Diffusion Policy MDP", $\mathcal{M}_{\text{DP}}$) involves a Gaussian likelihood at each (denoising) step and thus can be optimized with policy gradient. Blue circle denotes the state and red circle denotes the action in $\mathcal{M}_{\text{DP}}$.

**Two-Layer "Diffusion Policy MDP".** As observed in Black et al. (2023) and Psenka et al. (2023), a denoising process can be represented as a multi-step MDP in which policy likelihood of each denoising step can be obtained directly. We extend this formalism by embedding the Diffusion MDP into the environmental MDP, obtaining a larger "Diffusion Policy MDP" denoted $\mathcal{M}_{\text{DP}}$, visualized in Fig. 2. Below, we use the notation $\delta$ to denote a Dirac distribution and $\otimes$ to denote a product distribution. Recall the environment MDP $\mathcal{M}_{\text{ENV}} := (\mathcal{S}, \mathcal{A}, P_0, P, R)$ in Section 3. The Diffusion MDP $\mathcal{M}_{\text{DP}}$ uses indices $\bar{t}(t, k) = tK + (K - k - 1)$ corresponding to $(t, k)$, which increases in $t$ but (to keep the indexing conventions of diffusion) *decreases* lexicographically with $K - 1 \geq k \geq 0$. The states, actions and rewards are

$$\bar{s}_{\bar{t}(t,k)} = (s_t, a_t^{k+1}), \quad \bar{a}_{\bar{t}(t,k)} = a_t^k, \quad \bar{R}_{\bar{t}(t,k)}\big(\bar{s}_{\bar{t}(t,k)}, \bar{a}_{\bar{t}(t,k)}\big) = \begin{cases} 0 & k > 0 \\ R(s_t, a_t^0) & k = 0 \end{cases},$$

where the bar-action at $\bar{t}(t, k)$ is the action $a_t^k$ after one denoising step. Reward is only given at times corresponding to when $a_t^0$ is taken. The initial state distribution is $\bar{P}^0 = P_0 \otimes \mathcal{N}(0, \mathbf{I})$, corresponding to $s_0 \sim P_0$ is the initial distribution from the environmental MDP and $a_0^K \sim \mathcal{N}(0, \mathbf{I})$ independently. Finally, the transitions are

$$\bar{P}(\bar{s}_{\bar{t}+1} \mid \bar{s}_{\bar{t}}, \bar{a}_{\bar{t}}) = \begin{cases} (s_t, a_t^k) \sim \delta_{(s_t, a_t^k)} & \bar{t} = \bar{t}(t, k), k > 0 \\ (s_{t+1}, a_{t+1}^K) \sim P(s_{t+1} \mid s_t, a_t^0) \otimes \mathcal{N}(0, \mathbf{I}) & \bar{t} = \bar{t}(t, k), k = 0 \end{cases}.$$

That is, the transition moves the denoised action $a_t^k$ at step $\bar{t}(t, k)$ *into the next state* when $k > 0$, or otherwise progresses the environment MDP dynamics with $k = 0$. The pure noise $a_t^K$ is considered part of the *environment* when transitioning at $k = 0$. In light of (3.1), the policy in $\mathcal{M}_{\text{DP}}$ is

$$\bar{\pi}_\theta(\bar{a}_{\bar{t}(t,k)} \mid \bar{s}_{\bar{t}(t,k)}) = \pi_\theta(a_t^k \mid a_t^{k+1}, s_t) = \mathcal{N}(a_t^k; \mu(a_t^{k+1}, \varepsilon_\theta(a_t^{k+1}, k+1, s_t)), \sigma_{k+1}^2 \mathbf{I}). \quad (4.1)$$

Fortunately, (4.1) is a *Gaussian likelihood*, which can be evaluated analytically and is amenable to the policy gradient updates[2]:

$$\nabla_\theta \bar{\mathcal{J}}(\bar{\pi}_\theta) = \mathbb{E}^{\bar{\pi}_\theta, \bar{P}, \bar{P}^0}\Big[\sum_{\bar{t}\geq 0} \nabla_\theta \log \bar{\pi}_\theta(\bar{a}_{\bar{t}} \mid \bar{s}_{\bar{t}})\bar{r}(\bar{s}_{\bar{t}}, \bar{a}_{\bar{t}})\Big], \quad \bar{r}(\bar{s}_{\bar{t}}, \bar{a}_{\bar{t}}) := \sum_{\tau \geq \bar{t}}\gamma(\tau)\bar{R}(\bar{s}_\tau, \bar{a}_\tau). \quad (4.2)$$

Evaluating the above involves sampling through the denoising process, which is the usual "forward pass" that samples actions in Diffusion Policy; as noted above, the inital state can be sampled from the enviroment via $\bar{P}^0 = P_0 \otimes \mathcal{N}(0, \mathbf{I})$, where $P_0$ is from the environment MDP.

## 4.1 INSTANTIATING **DPPO** WITH PROXIMAL POLICY OPTIMIZATION

We apply Proximal Policy Optimization (PPO) (Schulman et al., 2017; Engstrom et al., 2019; Huang et al., 2022; Achiam, 2018), a popular improvement of the vanilla policy gradient update.

**Definition 4.1** (Generalized PPO, clipping variant). Consider a general MDP. Given an advantage estimator $\hat{A}(s, a)$, the PPO update (Schulman et al., 2017) is the sample approximation to

$$\nabla_\theta \mathbb{E}^{(s_t,a_t)\sim\pi_{\theta_{\text{old}}}} \min\Big(\hat{A}^{\pi_{\theta_{\text{old}}}}(s_t, a_t)\frac{\pi_\theta(a_t \mid s_t)}{\pi_{\theta_{\text{old}}}(a_t \mid s_t)}, \hat{A}^{\pi_{\theta_{\text{old}}}}(s_t, a_t)\,\text{clip}\Big(\frac{\pi_\theta(a_t \mid s_t)}{\pi_{\theta_{\text{old}}}(a_t \mid s_t)}, 1 - \varepsilon, 1 + \varepsilon\Big)\Big),$$

where $\varepsilon$, the clipping ratio, controls the maximum magnitude of the policy updated.

We instantiate PPO in our diffusion MDP with $(s, a, t) \leftarrow (\bar{s}, \bar{a}, \bar{t})$. Our advantage estimator respects the two-level nature of the MDP: let $\gamma_{\text{ENV}} \in (0, 1)$ be the environment discount and $\gamma_{\text{DENOISE}} \in (0, 1)$ be the denoising discount. Consider the environment-discounted return:

$$\bar{r}(\bar{s}_{\bar{t}}, \bar{a}_{\bar{t}}) := \sum_{t' \geq t}\gamma_{\text{ENV}}^{t'-t}\bar{r}(\bar{s}_{\bar{t}(t',0)}, \bar{a}_{\bar{t}(t',0)}), \quad \bar{t} = \bar{t}(t, k),$$

since $\bar{R}(\bar{t}) = 0$ at $k > 0$. This fact also obviates the need of estimating the value at $k > 1$ and allows us to use the following denoising-discounted advantage estimator[3]:

$$\hat{A}^{\pi_{\theta_{\text{old}}}}(\bar{s}_{\bar{t}}, \bar{a}_{\bar{t}}) := \gamma_{\text{DENOISE}}^k\Big(\bar{r}(\bar{s}_{\bar{t}}, \bar{a}_{\bar{t}}) - \hat{V}^{\bar{\pi}_{\theta_{\text{old}}}}(\bar{s}_{\bar{t}(t,0)})\Big)$$

The denoising-discounting has the effect of downweighting the contribution of noisier steps (larger $k$) to the policy gradient (see study in Appendix D.2). Lastly, we choose the value estimator to *only depend* on the "s" component of $\bar{s}$:  $\hat{V}^{\bar{\pi}_{\theta_{\text{old}}}}(\bar{s}_{\bar{t}(t,0)}) := \tilde{V}^{\bar{\pi}_{\theta_{\text{old}}}}(s_t)$,  which we find leads to more efficient and stable training compared to also estimating the value of applying the denoised action $a_t^{k=1}$ (part of $\bar{s}_{\bar{t}(t,0)}$) as shown in Appendix D.2.

**Best Practices for DPPO.** We summarize a number of best practices for **DPPO**; precise details are given in Appendix B. **(1)** We achieve substantial efficiency gains by fine-tuning the last few steps of the DDPM, whilst in many cases obtaining performance comparable to fine-tuning all steps. **(2)** For additional efficiency gains, one may fine-tune the Denoising Diffusion Implicit Model (DDIM) (Song et al., 2020a) instead. **(3)** We propose clipping the diffusion noise schedule at a larger-than-standard noise level to encourage exploration and training stability.

## 5 PERFORMANCE EVALUATION OF **DPPO**

We study the performance of **DPPO** in popular RL and robotics benchmarking environments. For comparisons, we consider (1) alternative RL methods for fine-tuning diffusion policies (Section 5.1), (2) alternative RL methods that leverage expert data (Section 5.2), (3) policy optimization using alternative policy parameterizations (Gaussian/GMM) (Section 5.3) additionally in multi-stage manipulation tasks including hardware evaluation (Section 5.4). These comparisons highlight the effectiveness of (1) policy gradient as a RL method for diffusion, (2) **DPPO** as a general framework

---

[2](Psenka et al., 2023) proposes a similar derivation but does not consider the denoising process as a MDP. See further clarification in Appendix A.

[3]In practice, we use Generalized Advantage Estimation (GAE, Schulman et al. (2015)) that better balances variance and bias in estimating the advantage. We present the simpler form here.

for pre-training plus fine-tuning, (3) diffusion as a RL parameterization (4) especially in challenging tasks, respectively.

**Environments: OpenAI Gym.** We first consider three population OpenAI GYM locomotion benchmarks (Brockman et al., 2016) : {Hopper-v2, Walker2D-v2, HalfCheetah-v2}. All policies are pre-trained with the full medium-level datasets from D4RL (Fu et al., 2020) with **state** input and action chunk size $T_a = 4$. We use the original **dense** reward setup in fine-tuning.

**Environments: Robomimic.** Next we consider four simulated robot manipulation tasks from the ROBOMIMIC benchmark (Mandlekar et al., 2021), {Lift, Can, Square, Transport}, ordered in increasing difficulty. These are more representative of real-world robotic tasks, and Square and Transport (Fig. 3) are considered very challenging for RL training. Both **state** and **pixel** policy input are considered. State-based and pixel-based policies are pre-trained with demonstrations provided by ROBOMIMIC. We consider $T_a = 4$ for Can, Lift, and Square, and $T_a = 8$ for Transport. They are then fine-tuned with **sparse** reward upon task completion.

**Environments: Furniture-Bench & real furniture assembly.** Finally, we demonstrate solving longer-horizon, multi-stage robot manipulation tasks from the FURNITURE-BENCH (Heo et al., 2023) benchmark. We consider three simulated furniture assembly tasks, {One-leg, Lamp, Round-table}. We consider two levels of randomness over initial state distribution, Low and Med, defined by the benchmark. All policies are pre-trained with 50 human demonstrations collected in simulation and $T_a = 8$. They are then fine-tuned with **sparse** (indicator of task stage completion) reward. We also evaluate the **zero-shot sim-to-real performance** with One-leg.

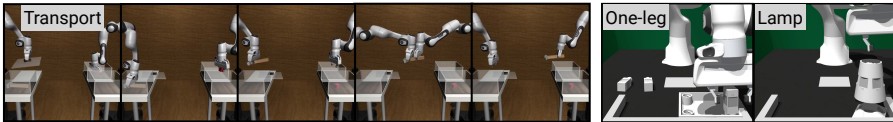

Figure 3: **Long-horizon robot manipulations tasks** including (left) the bimanual Transport from ROBOMIMIC and (right) FURNITURE-BENCH tasks (full rollouts visualized in Fig. 23).

### 5.1 COMPARISON TO DIFFUSION-BASED RL ALGORITHMS

We compare **DPPO** to an extensive list of RL methods for fine-tuning diffusion-based policies. **DRWR** and **DAWR** are *our own, novel* baselines based on reward-weighted regression (Peters and Schaal, 2007) and advantage-weighted regression (Peng et al., 2019). The remaining methods, **DIPO** (Yang et al., 2023), **IDQL** (Hansen-Estruch et al., 2023), **DQL** (Wang et al., 2022), and **QSM** (Psenka et al., 2023), are existing in the literature. We evaluate on the three OpenAI GYM tasks and the four ROBOMIMIC tasks with **state** input; see Appendix F.3 for further details.

Overall, **DPPO** performs consistently, exhibits great training stability, and enjoys strong fine-tuning performance across tasks. In the GYM tasks (Figure 4, top row), **IDQL** and **DIPO** exhibit competitive performance, while the other methods often perform worse and train less stably. **DPPO** is the strongest performer in the ROBOMIMIC tasks (Figure 4, bottom row), especially in the challenging Transport tasks. Surprisingly, **DRWR**, are strong baselines in {Lift, Can, Square} but underperforms in Transport, while all other baselines fare worse still. We postulate that the other baselines, using off-policy updates and propagating biased gradients from the Q function to the actor, suffers from even greater training instability in sparse-reward ROBOMIMIC tasks given continuous action space plus large action chunk sizes (see furtuer studies in Appendix D.2).

### 5.2 COMPARISON TO OTHER DEMO-AUGMENTED RL ALGORITHMS

We compare **DPPO** with recently proposed RL methods for training robot policies (not necessarily diffusion-based) leveraging offline data, including **RLPD** (Ball et al., 2023), **Cal-QL** (Nakamoto et al., 2024), and **IBRL** (Hu et al., 2023). These methods add expert data in the replay buffer and performs off-policy updates (**IBRL** and **Cal-QL** also do pre-training), which significantly improves efficiency v.s. **DPPO** in HalfCheetah-v2. However, in sparse-reward manipulation tasks including Can and Square, **DPPO** achieves much better final performance than all three methods; **RLPD** and **Cal-QL** fail to learn at all and **IBRL** saturates at lower success levels. See Appendix D.1, containing Fig. 11, for further discussion.

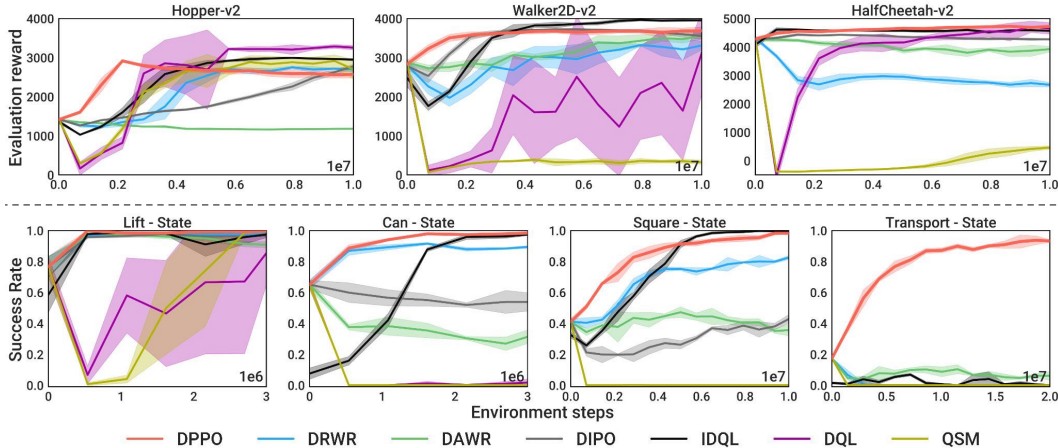

Figure 4: **Comparing to other diffusion-based RL algorithms.** Top row: GYM tasks (Brockman et al., 2016) averaged over five seeds. Bottow row: ROBOMIMIC tasks (Mandlekar et al., 2021), averaged over three seeds with **state** observation. **DPPO** curves are slightly thicker for better visibility.

## 5.3 COMPARISON TO OTHER POLICY PARAMETERIZATIONS

We compare **DPPO** with popular RL policy parameterizations: unimodal Gaussian with diagonal covariance (Sutton et al., 1999) and Gaussian Mixture Model (GMM (Bishop and Nasrabadi, 2006)), using either MLPs or Transformers (Vaswani et al., 2017), and also fine-tuned with the PPO objective. We compare these to **DPPO**-MLP and **DPPO**-UNet, which use either MLP or UNet as the network backbone. We evaluate on the four tasks from ROBOMIMIC (Lift, Can, Square, Transport) with both **state** and **pixel** input. With state input, **DPPO** pre-trains with 20 denoising steps and then fine-tunes the last 10. With pixel input, **DPPO** pre-trains with 100 denoising steps and then fine-tunes 5 DDIM steps.

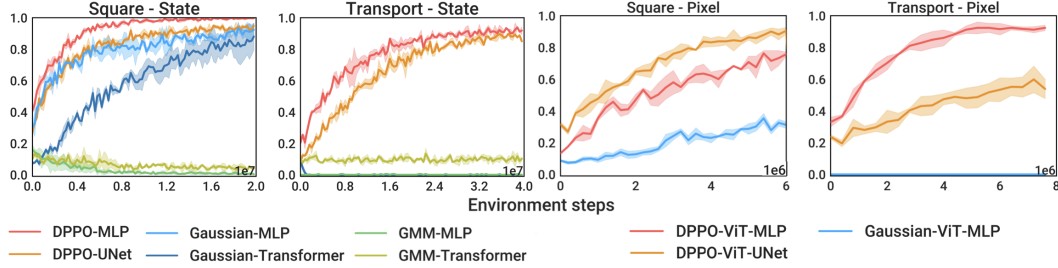

Figure 5: **Comparing to other policy parameterizations** in the more challenging Square and Transport tasks from ROBOMIMIC, with **state** (left) or **pixel** (right) observation. Results are averaged over three seeds.

Fig. 5 display results for the more challenging Square and Transport — we defer the results in Lift and Can to Fig. 20. With **state** input, **DPPO** outperforms Gaussian and GMM policies, with faster convergence to ∼100% success rate in Lift and Can, and greater final performance on Square and the challenging Transport, where it reaches > 90%. UNet and MLP variants perform similarly, with the latter training somewhat more rapidly. With **pixel** inputs, we use a Vision-Transformer-based (ViT) image encoder introduced in Hu et al. (2023) and an MLP head and compare the resulting variants **DPPO**-ViT-MLP and Gaussian-ViT-MLP (we omit GMM due to poor performance in state-based training). While the two are comparable on Lift and Can, **DPPO** trains more quickly and to higher accuracy on Square, and *drastically outperforms* on Transport, whereas Gaussian does not improve from its 0% pre-trained success rate. **To our knowledge, DPPO is *the first RL algorithm* to solve Transport from either state or pixel input to high (>50%) success rates.**

## 5.4 Evaluation on Furniture-Bench, and sim-to-real transfer

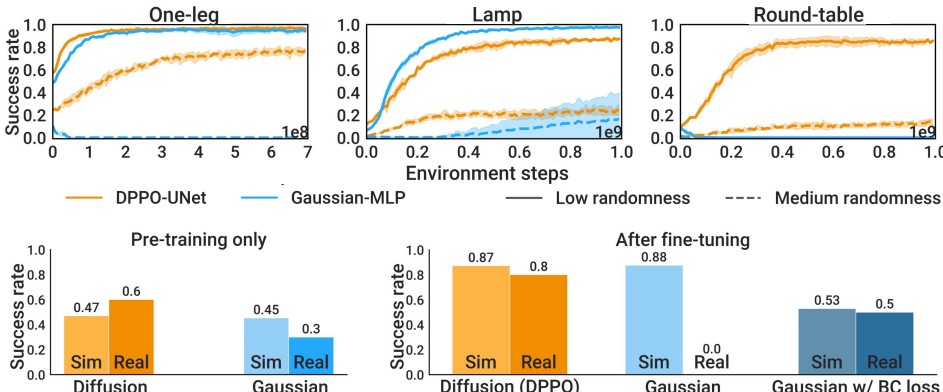

Figure 6: (Top) **DPPO** vs. Gaussian-MLP baseline in **simulated** FURNITURE-BENCH **tasks**. Results are averaged over three seeds. (Bottom) **Sim-to-real transfer results in** `One-leg`.

Here we evaluate **DPPO** on the long-horizon manipulation tasks from FURNITURE-BENCH (Heo et al., 2023). We compare **DPPO** to Gaussian-MLP, the overall most effective baseline from Section 5.3. Fig. 6 (top row) shows the evaluation success rate over fine-tuning iterations. **DPPO** exhibits strong training stability and improves policy performance in all six settings. Gaussian-MLP collapses to zero success rate in all three tasks with `Med` randomness (except for one seed in `Lamp`) and `Round-table` with `Low` randomness. Note that we are only using 50 human demonstrations for pre-training; we expect **DPPO** can leverage additional human data (better state space coverage) to further improve in `Med`, which is corroborated by ablation studies in Appendix D.3.

**Sim-to-real transfer.** We evaluate **DPPO** and Gaussian policies trained in the simulated `One-leg` task on physical hardware zero-shot (i.e., **no real data fine-tuning / co-training**) over 20 trials. Please see additional simulation training and hardware details in Appendix F.8. Fig. 6 (bottom row) shows simulated and hardware success rates after pre-training and fine-tuning. Notably, **DPPO** improves the real-world performance to 80% (16 out of 20 trials). Though the Gaussian policy achieves a high success rate in simulation after fine-tuning (88%), it fails entirely on hardware (0%). Supplemental video suggests it exhibits volatile and jittery behavior. For stronger comparison, we also fine-tune the Gaussian policy with an auxiliary behavior-cloning loss (Torne et al., 2024) such that the fine-tuned policy is encouraged to stay close to the base policy. However, this limits fine-tuning and only leads to 53% success rate in simulation and 50% in reality. *Qualitatively*, we find fine-tuned policies to be more robust and exhibit more corrective behaviors than pre-trained-only policies, especially during the insertion stage of the task; such behaviors are visualized in Fig. 24 with representative hardware rollouts.

## 5.5 Summary of ablation findings

Our ablation studies (c.f. Appendix D.2) find that: **(1)** for challenging tasks, using a value estimator which depends on environment state but is *independent of denoised action* is crucial for performance; we conjecture that this is related to the high stochasticity of Diffusion Policy; **(2)** there is a sweet spot for clipping the denoising noise level for **DPPO** exploration, trading off between too little exploration and too much action noise; **(3)** **DPPO** is resilient to fine-tuning fewer-than-$K$ denoising steps, yielding improved runtime and comparable performance; **(4)** **DPPO** yields improvements over Gaussian-MLP baselines for varying levels of expert demonstration data, and achieves comparable final performance and sample efficiency when **training from scratch** in GYM environments.

## 6 Understanding the performance of **DPPO**

The improvement of **DPPO** over popular Gaussian and GMM methods in Section 5.3 comes as a surprise initially as **DPPO** solves a much longer Diffusion Policy MDP (Section 4) than the origi-

nal environment MDP that other methods solve. This leads us to study the factors contributing to **DPPO**'s strong performance through a series of investigate experiments below.

We use the `Avoid` environment from D3IL benchmark (Jia et al., 2024), where a robot arm needs to reach the other side of the table while avoiding an array of obstacles (Fig. 7, top-left). The action space is the 2D target location of the end-effector. D3IL provides a set of expert demonstrations that covers different possible paths to the goal line — we consider three subsets of the demonstrations, M1, M2, and M3 in Fig. 7, each with two distinct modes. We pre-train MLP-based Diffusion, Gaussian, and GMM policies ($T_a = 4$ unless noted) with the demonstrations. For fine-tuning, we assign (sparse) reward when the robot reaches the goal line from the topmost mode. Gaussian and GMM policies are also fine-tuned with the PPO objective.

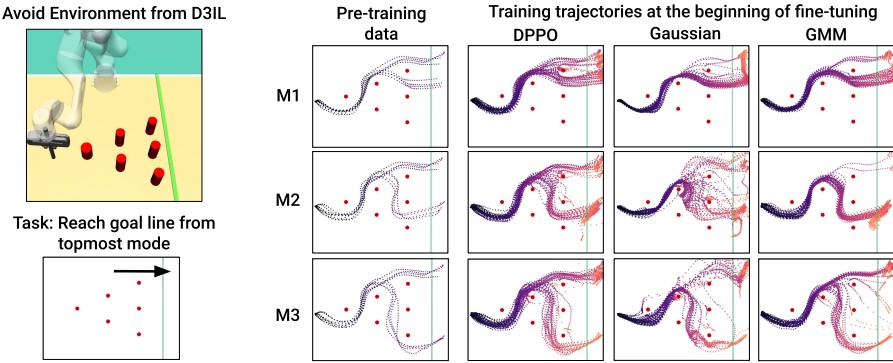

Figure 7: (Left) We use the `Avoid` environment from Jia et al. (2024) to visualize the **DPPO**'s exploration tendencies. The task is to reach the green goal line from the topmost mode. (Right) **Structured exploration.** We show sampled trajectories at the *first iteration of fine-tuning* for DPPO, Gaussian, and GMM after pre-training on three sets of expert demonstrations, M1, M2, and M3.

**Benefit 1: Structured, on-manifold exploration.** Fig. 7 (right) shows the sampled trajectories (with exploration noise) from **DPPO**, Gaussian, and GMM during the first iteration of fine-tuning. **DPPO** explores in wide coverage **around the expert data manifold**, whereas Gaussian generates less structured exploration noise (especially in M2) and GMM exhibits narrower coverage. Unlike Gaussian policy that adds noise only to the final sampled action, diffusion adds multiple rounds of noise through denoising. Each denoising step expands the coverage with new noise while also pushing the newly denoised action towards the expert data manifold (Permenter and Yuan, 2023). Moreover, the combination of diffusion parameterization with the denoising of *action chunks* means that policy stochasticity in **DPPO** is **structured in both action dimension and time horizon**.

**Benefit 2: Training stability from multi-step denoising process.** In Fig. 8 (left), we run fine-tuning after pre-training with M2 and *attempt to de-stabilize fine-tuning* by gradually adding noise to the action during the fine-tuning process (see Appendix F.9 for details). We find that Gaussian and GMM's performance both collapse, while with **DPPO**, the performance is robust to the noise if at least four denoising steps are used. This property also allows **DPPO** to apply significant noise to the sampled actions, simulating an imperfect low-level controller to facilitate sim-to-real transfer in Section 5.4. In Fig. 8 (right), we also find **DPPO** enjoys greater training stability when fine-tuning long action chunks, e.g., up to $T_a = 16$, while Gaussian and GMM can fail to improve at all.

Fig. 9 visualizes how **DPPO** affects the multi-step denoising process. Over fine-tuning iterations, the action distribution gradually converges through the denoising steps — the iterative refinement is largely preserved, as opposed to, e.g., "collapsing" to the optimal actions at the first fine-tuned denoising step or the final one. We postulate this contributes to the training stability of **DPPO**.

**Benefit 3: Robust and generalizable fine-tuned policy.** **DPPO** also generates final policies robust to perturbations in dynamics and the initial state distribution. In Fig. 10, we again add noise to the actions sampled from the fine-tuned policy (no noise applied during training) and find that **DPPO** policy exhibits strong robustness to the noise compared to the Gaussian policy. **DPPO** policy also converges to the (near-)optimal path from a larger distribution of initial states. This finding echoes theoretical guarantees that Diffusion Policy, capable of representing complex multi-modal

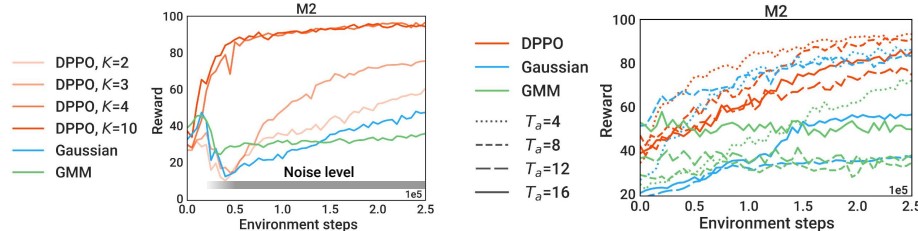

Figure 8: **Training stability.** Fine-tuning performance (averaged over five seeds, standard deviation not shown) after pre-training with M2. (Left) Noise is injected into the applied actions after a few training iterations. (Right) The action chunk size $T_a$ is varied.

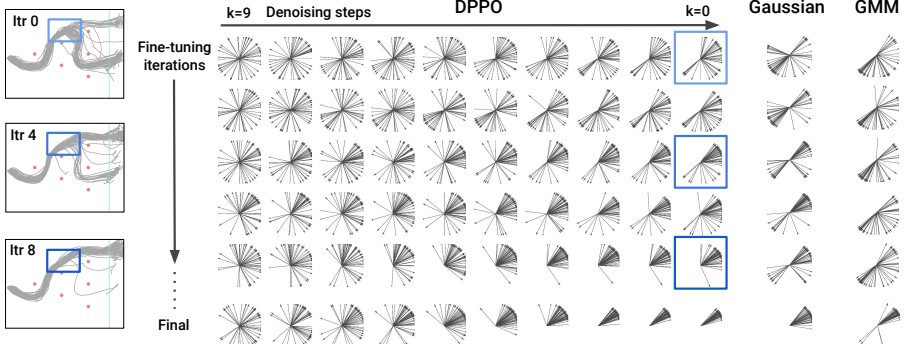

Figure 9: **Preserving the iterative refinement.** The 2D actions from 50 trajectories at the branching point *through fine-tuning* iterations after pre-training with M2. For **DPPO**, we also visualize the action distribution through the final denoising steps at each fine-tuning iteration.

data distribution, can effectively deconvolve noise from noisy states (Block et al., 2024), a property used in Chen et al. (2024) to stabilize long-horizon video generation.

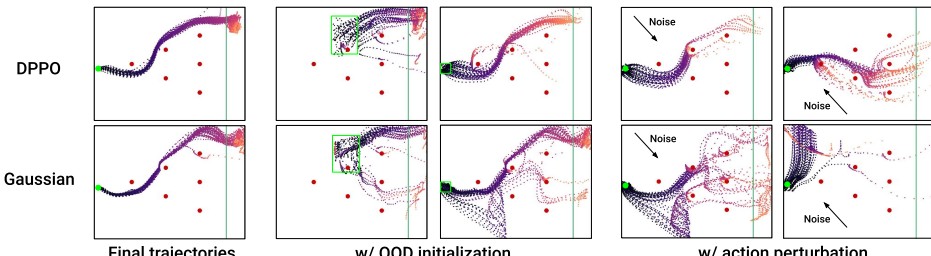

Figure 10: **Policy robustness** *after fine-tuning*. Green dot / box indicates the initial state region.

## 7 CONCLUSION AND FUTURE WORK

We believe **DPPO** will become an important component in the pre-training-plus-fine-tuning pipeline for training general-purpose real-world robotic policies. To this end, we hope in future work to further showcase the promise of **DPPO** for simulation-to-real transfer (Chen et al., 2023b; Liang et al., 2020; Ren et al., 2023; Chi et al., 2024a) in which we fine-tune a vision-based policy that has been pre-trained on a variety of diverse tasks. We expect this pre-training to provide a large and diverse expert data manifold, of which, as we have shown in Section 6, **DPPO** is well-suited to take advantage for better exploration during fine-tuning. We are also excited to understand how **DPPO** can fit together with other decision-making tools such as model-based planning (Janner et al., 2022; Ding et al., 2024) and decision-making aided by video prediction (Chen et al., 2024); these tools may help address the main limitation of **DPPO** — its lower sample efficiency than off-policy methods — and unlocking performing practical RL in physical hardware.

## ACKNOWLEDGMENTS

We would like to thank Lirui Wang and Terry Suh for helpful discussions in the early stage of the project. The authors were partially supported by the Toyota Research Institute (TRI). This article solely reflects the opinions and conclusions of its authors and not TRI or any other Toyota entity.

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

CONTENTS

# A   EXTENDED RELATED WORK

## A.1   RL TRAINING OF ROBOT POLICIES WITH OFFLINE DATA

Here, we discuss related work in training robot policies using RL augmented with offline data to help RL better explore online in sparse reward settings.

One simple form is to use offline data to pre-train the policy, typically using behavior cloning, and then fine-tune the policy online. This is the approach that **DPPO** takes. Often, a regularization loss is applied to constrain the fine-tuned policy to stay close to the base policy, leading to natural fine-tuned behavior and often better learning (Rajeswaran et al., 2017; Zhu et al., 2019; Torne et al., 2024). **DPPO** does not apply regularization at fine-tuning as we find the on-manifold exploration helps **DPPO** maintain natural behavior after fine-tuning Section 5.4. Another popular approach is to learn a *residual* policy with RL on top of the frozen base policy (Alakuijala et al., 2021; Haldar et al., 2023). A closer work to ours is Ankile et al. (2024b), which trains a one-step residual non-diffusion policy with on-policy RL on top of a pre-trained chunked diffusion policy. This approach has the benefit of being fully closed-loop but lacks the structured on-manifold exploration of **DPPO**. Another hybrid approach is from Hu et al. (2023), which uses pre-trained and fine-tuned policies to sample online experiences.

Another popular line of work, instead of training a base policy using offline data, directly adds the data in the replay buffer for online, off-policy learning in a single stage (Hester et al., 2018; Vecerik et al., 2017; Nair et al., 2020). One recent approach from Ball et al. (2023), **RLPD**, further improves sample efficiency from previous off-policy methods incorporating, e.g., critic ensembling. Luo et al. (2024) demonstrates **RLPD** solving real-world manipulation tasks (although generally less challenging than ones solved by **DPPO**).

Other approaches, including **Cal-QL**, build on offline RL to learn from offline data and then switch to online RL while still sampling from offline data (Nakamoto et al., 2024; Hansen-Estruch et al., 2023; Yang et al., 2024). Often the distributional mismatch between offline data and online policy needs to be addressed: **Cal-QL** proposes calibrated conservative Q-learning that learns a offline Q function that lower bounds the true value of the learned policy; Lei et al. (2023) proposes ensemble behavior cloning during pre-training to promote policy diversity (similarly in Wang et al. (2024b)); Lee et al. (2022) proposes prioritizing online samples and then near-on-policy samples from the offline dataset during fine-tuning; Zhang et al. (2023) proposes a similar method akin to Hu et al. (2023) that uses both pre-trained and online policies for collecting new samples.

## A.2   DIFFUSION-BASED RL METHODS

This section discusses related methods that directly train or improve diffusion-based policies with RL methods. The baselines to which we compare in Section 5.1 are discussed below as well, and are highlighted in their corresponding colors. We also refer the readers to Zhu et al. (2023) for an extensive survey on diffusion models for RL.

Most previous works have focused on the **offline** setting with a static dataset. One line of work focuses on state trajectory planning and *guiding* the denoising sampling process such that the sampled actions satisfy some desired objectives. Janner et al. (2022) applies classifier guidance that generates

trajectories with higher predicted rewards. Ajay et al. (2023) introduces classifier-free guidance that avoids learning the value of noisy states. There is another line of work that uses diffusion models as an action policy (instead of state planner) and generally applies Q-learning. **DQL** (Wang et al., 2022) introduces Diffusion Q-Learning that learns a state-action critic for the final denoised actions and backpropagates the gradient from the critic through the entire Diffusion Policy (actor) denoising chain, akin to the usual Q-learning. **IDQL** (Hansen-Estruch et al., 2023), or Implicit Diffusion Q-learning, proposes learning the critic to select the actions at inference time for either training or evaluation while fitting the actor to all sampled actions. Kang et al. (2024) instead proposes using the critic to re-weight the sampled actions for updating the actor itself, similar to weighted regression baselines **DAWR** and **DRWR** introduced in our work. Goo and Niekum (2022) similarly extracts the policy in the spirit of AWR (Peng et al., 2019). Chen et al. (2022a) trains the critic using value iteration instead based on samples from the actor. Finally, Jackson et al. (2024) explores using diffusion guidance to move offline data towards the target trajectory distribution.

We note that methods like **DQL** and **IDQL** can also be applied in the **online** setting. A small amount of work also focuses entirely on the online setting. **DIPO** (Yang et al., 2023) differs from **DQL** and related work in that it uses the critic to update the sampled actions ("action gradient") instead of the actor — the actor is then fitted with updated actions from the replay buffer. **QSM**, or Q-Score Matching (Psenka et al., 2023), suggests that optimizing the likelihood of the entire chain of denoised actions can be inefficient (contrary to our findings in the fine-tuning setting) and instead proposes learning the optimal policy by iteratively aligning the gradient of the actor (i.e., score) with the action gradient of the critic. Rigter et al. (2023) proposes learning a diffusion dynamic model to generate synthetic trajectories for online training of a non-diffusion RL policy.

We note that almost all prior work in diffusion-based RL (offline or online) have relied on approximating the state-action Q function and using it to update the diffusion actor in some form — policy gradient update has been deemed challenging due to the multi-step denoising process (Psenka et al., 2023; Yang et al., 2023). Inaccurate Q values may lead to biased updates to the actor, which can lead to training collapse as it starts with decent pre-training performance but quickly drops to zero success rate as seen in Fig. 4, also failing to recover since then due to the sparse-reward setup. While Q-learning methods generally achieve better sample efficiency when they can solve the task of interest, our focus has been largely on challenging long-horizon robot manipulation tasks where the training stability is much desired.

Lastly, we point out that the role of stochasticity of diffusion/consistency policy for exploration is also explored by Chen et al. (2023a), who find such stochasticity suffices for exploration without additional strategies needed. Our work discovers the similar effect, but also performs extensive investigative experiments in Section 6 and ablation studies in Appendix D.7 to provide affirmative evidence for the distinct exploration strategies induced by diffusion.

**Distinction from the policy gradient formulation in Psenka et al. (2023).** There has been a different formulation introduced in Psenka et al. (2023) Sec. 3 that derives the policy gradient update for diffusion policy. The derivation is based on converting the gradient of the log likelihood of the final denoised action to the sum over log likelihood of individual denoising actions. This formulation, unlike **DPPO**, does not treat the multi-step denoising process as a MDP. In the policy gradient update, Psenka et al. (2023) sums over denoising steps and then takes expectation over environment steps, while **DPPO**'s update (4.2) takes expectation over both denoising and environment steps, potentially leading to better sample efficiency. Moreover, Psenka et al. (2023) does not propose leveraging PPO updates or other modifications to diffusion, and finds such vanilla form of policy gradient update to be ineffective. We formulate **DPPO** independent of their work and find **DPPO** highly effective in fine-tuning settings while also being competitive in training from scratch (Appendix D.3).

**Potential impact beyond robotics.** **DPPO** is a generic framework that can be potentially applied to fine-tuning diffusion-based models in sequential interactive settings beyond robotics. These include: extending diffusion-based text-to-image generation (Black et al., 2023; Clark et al., 2023) to a multi-turn interactive setting with human feedback; drug design/discovery applications (Luo et al., 2022; Huang et al., 2024) with policy search on the molecular level in feedback with simulators (in the spirit of prior non-diffusion-based drug discovery with RL (Popova et al., 2018)); and the

adaptation of diffusion-based language modeling (Sahoo et al., 2024; Lou et al., 2024) to interactive (e.g. with human feedback (Ouyang et al., 2022)), problem-solving and planning tasks.

## B  BEST PRACTICES FOR DPPO

**Fine-tune only the last few denoising steps.**  Diffusion Policy often uses up to $K = 100$ denoising steps with DDPM to better capture the complex data distribution of expert demonstrations. With **DPPO**, we can choose to fine-tune only a subset of the denoising steps instead, e.g., the last $K'$ steps. In Section 4.1 and Section 6 we find this speeds up **DPPO** training and reduces GPU memory usage without sacrificing the asymptotic performance. Instead of fine-tuning the pre-trained model weights $\theta$, we make a copy $\theta_{\mathrm{FT}}$ — $\theta$ is frozen and used for the early denoising steps, while $\theta_{\mathrm{FT}}$ is used for the last $K'$ steps and updated with **DPPO**. Although maintaining the frozen $\theta$ takes extra GPU memory (e.g., about 60MB in `HalfCheetah-v2`), we find that storing the extra intermediate denoised actions and their likelihoods due to higher $K'$ can taking significantly more GPU memory (e.g., 1.1GB extra if $K' = 20$ instead 10 in `HalfCheetah-v2`).

**Fine-tune DDIM.**  Instead of fine-tuning all $K$ or the last few steps of the DDPM, one can also apply Denoising Diffusion Implicit Model (DDIM) (Song et al., 2020a) during fine-tuning, which greatly reduces the number of sampling steps $K^{\mathrm{DDIM}} \ll K$, e.g., as few as 5 steps, and thus potentially improves **DPPO** efficiency as fewer steps are fine-tuned.

$$x^{k-1} \sim p_\theta^{\mathrm{DDIM}}(x^{k-1}|x^k) := \mathcal{N}(x^{k-1}; \mu^{\mathrm{DDIM}}(x^k, \varepsilon_\theta(x^k, k)), \eta\sigma_k^2\mathrm{I}), \quad k = K^{\mathrm{DDIM}}, ..., 0. \quad \text{(B.1)}$$

Although DDIM is typically used as a deterministic sampler by setting $\eta = 0$ in (B.1), we can use $\eta > 0$ for fine-tuning that provides exploration noise and avoids calculating Gaussian likelihood with a Dirac distribution. In practice, we set $\eta = 1$ for training (equivalent to applying DDPM Song et al. (2020a)) and then $\eta = 0$ for evaluation. *We reserve DDIM sampling for our pixel-based experiments and long-horizon furniture assembly tasks, where the efficiency improvements are much desired.*

**Diffusion noise scheduling.**  We use the cosine schedule for $\sigma_k$ introduced in Nichol and Dhariwal (2021), which was originally annealed to a small value on the order of $1E-4$ at $k = 0$. In **DPPO**, the value of $\sigma_k$ also translates to the exploration noise that is crucial to training efficiency. Empirically, we find that clipping $\sigma_k$ to a higher minimum value (denoted $\sigma_{\min}^{\mathrm{exp}}$, e.g., $0.01 - 0.1$) when sampling actions helps exploration (see sensitivity analysis in Appendix D.2). Additionally we clip $\sigma_k$ to be at least $0.1$ (denoted $\sigma_{\min}^{\mathrm{prob}}$) when evaluating the Gaussian likelihood $\log \bar{\pi}_\theta(\bar{a}_{\bar{t}}|\bar{s}_{\bar{t}})$, which improves training stability by avoiding large magnitude.

**Network architecture.**  We study both Multi-layer Perceptron (MLP) and UNet (Ronneberger et al., 2015) as the policy heads in Diffusion Policy. An MLP offers simpler setup and we find it generally fine-tunes more stably with **DPPO**. Moreover, since the UNet applies convolution to the denoised action, we can pre-train and fine-tune with different action chunk size $T_a$ (the number of environment timesteps that the policy predicts future actions with), e.g., 16 and 8. We find that **DPPO** benefits from pre-training with larger $T_a$ (better prediction) and fine-tuning with smaller $T_a$ (more amenable to policy gradient)[4].

## C  ADDITIONAL DETAILS OF DPPO IMPLEMENTATION

**Pseudocode.**  The pseudocode for **DPPO** is presented in Algorithm 1. DPPO takes as input a diffusion policy $\pi_\theta$ trained using behavior cloning loss $\mathcal{L}_{\mathrm{BC}}$. The policy is then fine-tuned using a PPO-style loss (Schulman et al., 2017) with careful treatment of the denoising process (Section 4).

---

[4]With fully-connected layers in MLP, empirically we find that using different chunk sizes for pre-training and fine-tuning with MLP leads to training instability.

**Pre-training.** The diffusion policy $\pi_\theta$ is pre-trained using a behavior cloning loss (Ho et al., 2020):

$$\mathcal{L}_{\text{BC}}(\theta) = \mathbb{E}^{(s_t, a_t^0) \sim \mathcal{D}_{\text{off}}} \big[ \|\varepsilon_t - \varepsilon_\theta(a_t^0, s_t, k)\|^2 \big], \tag{C.1}$$

where $\mathcal{D}_{\text{off}}$ is the offline dataset and $\varepsilon_\theta$ is the policy network predicting the sampled noise added to $a_t^0$ based on the noisy action. We use the cosine noise schedule from Nichol and Dhariwal (2021).

---

**Algorithm 1 DPPO**

---

1: Pre-train diffusion policy $\pi_\theta$ with offline dataset $\mathcal{D}_{\text{off}}$ using BC loss $\mathcal{L}_{\text{BC}}(\theta)$ Eq. (C.1).
2: Initialize value function $V_\phi$.
3: **for** iteration = 1, 2, ... **do**
4:      Initialize rollout buffer $\mathcal{D}_{\text{itr}}$.
5:      $\pi_{\theta_{\text{old}}} = \pi_\theta$.
6:      **for** environment = 1, 2, ..., N *in parallel* **do**
7:          Initialize state $\bar{s}_{\bar{t}(0,K)} = (s_0, a_0^K)$ in $\mathcal{M}_{\text{DP}}$.
8:          **for** environment step t = 1, ..., T, denoising step $k = K - 1, ..., 0$ **do**
9:              Sample the next denoised action $\bar{a}_{\bar{t}(t,k)} = a_t^k \sim \pi_{\theta_{\text{old}}}$.
10:             **if** $k = 0$ **then**
11:                 Run $a_t^0$ in the environment and observe $\bar{R}_{\bar{t}(t,0)}$ and $\bar{s}_{\bar{t}(t+1,K)}$.
12:             **else**
13:                 Set $\bar{R}_{\bar{t}(t,k)} = 0$ and $\bar{s}_{\bar{t}(t,k-1)} = (s_t, a_t^k)$.
14:             Add $(k, \bar{s}_{\bar{t}(t,k)}, \bar{a}_{\bar{t}(t,k)}, \bar{R}_{\bar{t}(t,k)})$ to $\mathcal{D}_{\text{itr}}$.
15:      Compute advantage estimates $A^{\pi_{\theta_{\text{old}}}}(s_{\bar{t}(t,k=0)}, a_{\bar{t}(t,k=0)})$ for $\mathcal{D}_{\text{itr}}$ using GAE Eq. (C.2).
16:      **for** update = 1, 2, ..., num_update **do**          ▷ Based on replay ratio $N_\theta$
17:          **for** minibatch = 1, 2, ..., B **do**
18:             Sample $(k, \bar{s}_{\bar{t}(t,k)}, \bar{a}_{\bar{t}(t,k)}, \bar{R}_{\bar{t}(t,k)})$ and $A^{\pi_{\theta_{\text{old}}}}(s_{\bar{t}(t,k)}, a_{\bar{t}(t,k)})$ from $\mathcal{D}_{\text{itr}}$.
19:             Compute denoising-discounted advantage $\hat{A}_{\bar{t}(t,k)} = \gamma_{\text{DENOISE}}^k A^{\pi_{\theta_{\text{old}}}}(s_{\bar{t}(t,0)}, a_{\bar{t}(t,0)})$.
20:             Optimize $\pi_\theta$ using policy gradient loss $\mathcal{L}_\theta$ Eq. (C.3).
21:             Optimize $V_\phi$ using value loss $\mathcal{L}_\phi$ Eq. (C.4).
22: **return** converged policy $\pi_\theta$.

---

**Environment-step advantage estimation.** We use Generalized Advantage Estimation (GAE) (Schulman et al., 2015) with parameter $\lambda$ for advantage estimation in Algorithm 1. GAE-$\lambda$ approximates the advantage function using the series

$$\hat{A}_{\bar{t}(t,k=0)}^\lambda = \sum_{l=0}^\infty (\gamma\lambda)^l \bar{\delta}_{\bar{t}(t+l,k=0)}, \quad \text{where } \bar{\delta}_{\bar{t}(t,k)} = \bar{R}_{\bar{t}(t,k)} + \gamma_{\text{ENV}} V_\phi(\bar{s}_{\bar{t}(t+1,k)}) - V_\phi(\bar{s}_{\bar{t}(t,k)}).$$

$$\tag{C.2}$$

Notably, GAE-$\lambda$ interpolates between a one-step temporal difference ($\hat{A}_{\bar{t}(t,k)}^{\lambda=0} = \bar{R}_{\bar{t}(t,k)} + \gamma_{\text{ENV}} V_\phi(\bar{s}_{\bar{t}(t+1,k)}) - V_\phi(\bar{s}_{\bar{t}(t,k)})$) and the Monte Carlo return of the episode relative to the baseline ($\hat{A}_{\bar{t}(t,k)}^{\lambda=1} = \sum_{l=0}^{T-t} \gamma_{\text{ENV}}^l \bar{R}_{\bar{t}(t+l,k)} - V_\phi(\bar{s}_{\bar{t}(t,k)})$). We refer the reader to Table 8 for additional details on GAE parameter selection and Section D.2 for ablations on the choice of advantage estimator.

Note that in Eq. (C.2) we are only concerned with $k = 0$, i.e., the final denoising step. In **DPPO** formulation, we only need to calculate the advantage for $k = 0$ (i.e., environment steps), but not for intermediate denoising steps. We only need to apply denoising discounting to the calculated advantages so they can be applied to each denoising step $k$.

**Fine-tuning.** During RL fine-tuning, we update the policy $\pi_\theta$ using the clipped objective:

$$\mathcal{L}_\theta = \mathbb{E}^{\mathcal{D}_{\text{itr}}} \min \left( \hat{A}^{\bar{\pi}_{\theta_{\text{old}}}}(\bar{s}_{\bar{t}}, \bar{a}_{\bar{t}}) \frac{\bar{\pi}_\theta(\bar{s}_{\bar{t}}, \bar{a}_{\bar{t}})}{\bar{\pi}_{\theta_{\text{old}}}(\bar{s}_{\bar{t}}, \bar{a}_{\bar{t}})}, \ \hat{A}^{\pi_{\theta_{\text{old}}}}(\bar{s}_{\bar{t}}, \bar{a}_{\bar{t}}) \operatorname{clip}\left( \frac{\bar{\pi}_\theta(\bar{s}_{\bar{t}}, \bar{a}_{\bar{t}})}{\bar{\pi}_{\theta_{\text{old}}}(\bar{s}_{\bar{t}}, \bar{a}_{\bar{t}})}, 1 - \varepsilon, 1 + \varepsilon \right) \right). \tag{C.3}$$

If we choose to fine-tune only the last $K'$ denoising steps, then we sample only those from $\mathcal{D}_{\text{itr}}$.

Finally, we train the value function to predict the future discounted sum of rewards (i.e., discounted returns):

$$\mathcal{L}_\phi = \mathbb{E}^{\mathcal{D}_{\text{itr}}}\big[\|\sum_{l=0}^{T-t}\gamma_{\text{ENV}}^l \bar{R}_{\bar{t}(t+l,k)} - V_\phi(s_t)\|^2\big]. \tag{C.4}$$

Similar to all baselines in Appendix F.3, we denote $N_\theta$ and $N_\phi$ the replay ratio for the actor (diffusion policy) and the value critic in **DPPO**; in practice we always set $N_\theta = N_\phi$. Similar to usual PPO implementations (Huang et al., 2022), the batch updates in an iteration terminate when the KL divergence between $\pi_\theta$ and $\pi_{\theta_{\text{old}}}$ reaches 1.

**Large batch size.** Since the gradient update in **DPPO** involves expectation over both environment steps and denoising steps, we use a larger batch size compared to, e.g., PPO training with Gaussian policy parameterization. Roughly we use the batch size from Gaussian training times the number of the fine-tuned denoising steps; in some cases like ROBOMIMIC we also observe that a much smaller batch size (close to that of Gaussian training) can be used and significantly improves sample efficiency.

**Gradient clipping ratio.** We also find the PPO clipping ratio, $\varepsilon$, can affect the training stability significantly in **DPPO** (as well as in Gaussian and GMM policies) especially in sparse-reward manipulation tasks. In practice we find that, a good indicator of the amount of clipping leading to optimal training efficiency, is to aim for a clipping fraction (fraction of individual samples being clipped in a batch) of 10% to 20%. For each method in different tasks, we vary $\varepsilon$ in $\{.1, .01, .001\}$ and choose the highest value that satisfies the clipping fraction target. Empirically we also find that, using a higher $\varepsilon$ for earlier denoising steps in **DPPO** further improves training stability in manipulation tasks. Denote $\varepsilon_k$ the clipping value at denoising step $k$, and in practice we set $\varepsilon_{k=(K-1)} = 0.1\varepsilon_{k=0}$, and it follows an exponential schedule among intermediate $k$.

# D ADDITIONAL EXPERIMENTAL RESULTS

## D.1 COMPARING TO OTHER DEMO-AUGMENTED RL METHODS

We compare **DPPO** with recently proposed RL methods for training robot policies (not necessarily diffusion-based) leveraging offline data, including **RLPD** (Ball et al., 2023), **Cal-QL** (Nakamoto et al., 2024), and **IBRL** (Hu et al., 2023). These methods add expert data in the replay buffer and performs off-policy updates (**IBRL** and **Cal-QL** also pretrain with behavior cloning and offline RL objectives, respectively), which significantly improves sample efficiency vs. **DPPO** in HalfCheetah-v2 (Fig. 11, top left).

Among the FRANKA-KITCHEN settings (Fig. 11, top right), we find **RLPD** and **IBRL** fail to learn well especially with noisier demonstrations from Kitchen-Partial-v0 and Kitchen-Mixed-v0. **Cal-QL** achieves competitive performance but **DPPO** still achieves overall the best performance especially with Kitchen-Complete-v0. We note that **DPPO**, not using any expert data during fine-tuning, can be sensitive to the pre-training performance; we find the incomplete demonstrations in Kitchen-Partial-v0 and Kitchen-Mixed-v0 cause challenge in fully modeling the multi-modality of the data even with diffusion parameterization and prevent **DPPO** from achieving (near-)perfect fine-tuning performance.

Nonetheless, we believe the expert demonstrations from ROBOMIMIC are most reflective of pretraining plus fine-tuning in robotics as all demonstrations complete the task despite the varying quality. Fig. 11 bottom row shows the performance of **DPPO** and baselines using either cleaner PH or noisier MH data in Can and Square; **DPPO** exhibits strong final performance. **RLPD** and **Cal-QL** fail to learn at all. Although **IBRL** matches the success rates with **DPPO** in Square, we find that it saturates at lower success levels while **DPPO** continues to reach $\sim 100\%$ success rates (not shown). **DPPO** also runs significantly faster in wall-clock time than the baselines as it leverages

sampling from highly parallelized environments[5]; thus we cap the number of samples at 1e6 for the baselines in ROBOMIMIC, also since their performance saturates.

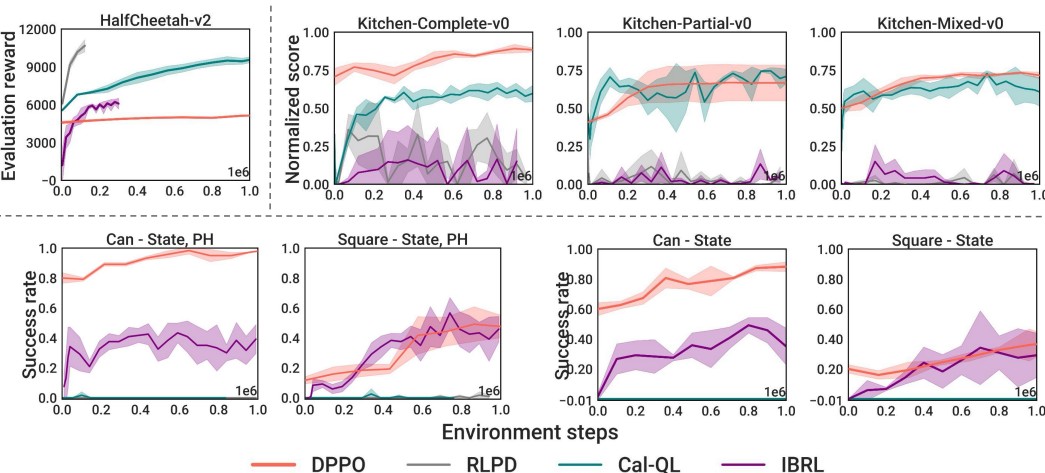

Figure 11: **Comparing to other demo-augmented RL methods.** Results are averaged over five seeds in HalfCheetah-v2 and three seeds in Kitchen-Complete-v0, Kitchen-Partial-v0, Kitchen-Partial-v0, Can, and Square.

We have also experimented with using diffusion policies for **RLPD**, **Cal-QL**, and **IBRL**, and the results are shown in Fig. 12. We use either action chunk size $T_a = 1$ or $T_a = 4$. We see similar results as in Fig. 11 using Gaussian policies that **RLPD** and **Cal-QL** fails to solve the task at all. We believe that the worse performance of **Cal-QL** is due to the offline RL objective (based on learning the state-action Q function) making learning precise continuous actions needed in ROBOMIMIC tasks very difficult, regardless of the policy parameterization, which corroborates our original finding in Section 5.1 when comparing **DPPO** to Q-learning-based diffusion RL methods. Compared to **RLPD** that trains with the SAC objective and expert data in the replay buffer, **IBRL**, using BC pre-training, is able to learn a base policy more effectively and uses it for online data collection. **DPPO** benefits from directly fine-tuning the pre-trained policy (instead of training a new one using experiences from the pre-trained policy), and achieves similar or better sample efficiency before 1e6 steps compared to **IBRL**, and converges to ~100% success rates unlike **IBRL** saturates at lower levels (not shown).

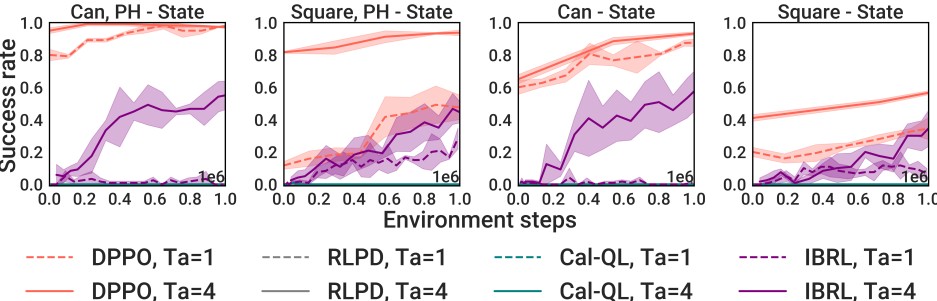

Figure 12: **Using diffusion policy for other demo-augmented RL methods.** Results are averaged over three seeds.

---

[5]Off-policy methods (baselines) usually cannot fully leverage parallelized sampling as the policy is updated less often (e.g., 50 updates per 50 samples instead of 1 update per 1 sample) and the performance can be affected.

## D.2 ABLATION STUDIES ON DESIGN DECISIONS IN **DPPO**

**1. Choice of advantage estimator.** In Section 4.1 we demonstrate how to efficiently estimate the advantage used in PPO updates by learning $\tilde{V}(s_t)$ that only depends on the environment state; the advantage used in **DPPO** is formally

$$\hat{A} = \gamma_{\text{DENOISE}}^k (\bar{r}(\bar{s}_{\bar{t}}, \bar{a}_{\bar{t}}) - \tilde{V}(s_t)).$$

We now compare this choice with learning the value of the full state $\bar{s}_{\bar{t}(t,0)}$ that includes environment state $s_t$ *and* denoised action $a_t^{k=1}$. We additionally compare with the state-action Q-function estimator used in Psenka et al. (2023)[6], $\tilde{Q}(s_t, a_t^{k=0})$, that does not directly use the rollout reward $\bar{r}$ in the advantage.

Fig. 13 shows the fine-tuning results in Hopper-v2 and HalfCheetah-v2 from GYM, and Can and Square from ROBOMIMIC. On the simpler Hopper-v2, we observe that the two baselines, both estimating the value of some action, achieves higher reward during fine-tuning than **DPPO**'s choice. However, in the more challenging tasks, the environment-state-only advantage used in **DPPO** consistently leads to the most improved performance. We believe estimating the accurate value of applying a continuous and high-dimensional action can be challenging, and this is exacerbated by the high stochasticity of diffusion-based policies and the action chunk size. The results here corroborate the findings in Section 5.1 that off-policy Q-learning methods can perform well in Hopper-v2 and Walker2D-v2, but often exhibit training instability in manipulation tasks from ROBOMIMIC.

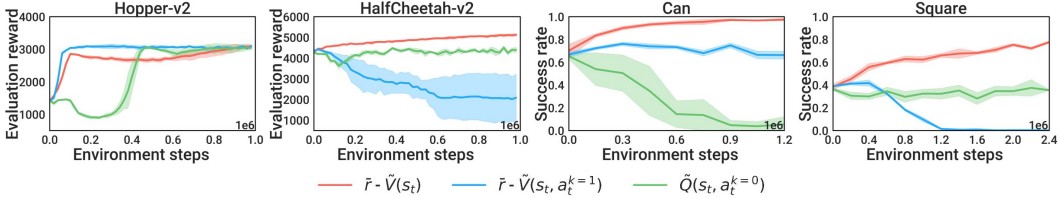

Figure 13: **Choice of advantage estimator.** Results are averaged over five seeds in Hopper-v2 and HalfCheetah-v2 and three seeds in Can and Square.

**Denoising discount factor.** We further examine how $\gamma_{\text{DENOISE}}$ in the **DPPO** advantage estimator affects fine-tuning. Using a smaller value (i.e., more discount) has the effect of downweighting the contribution of earlier denoising steps in the policy gradient. Fig. 14 shows the fine-tuning results in the same four tasks with varying $\gamma_{\text{DENOISE}} \in [0.5, 0.8, 0.9, 1]$. We find in Hopper-v2 and HalfCheetah-v2 $\gamma_{\text{DENOISE}} = 0.8$ leads to better efficiency while smaller $\gamma_{\text{DENOISE}} = 0.5$ slows training. The value does not affect training noticeably in Can. In Square the smaller $\gamma_{\text{DENOISE}} = 0.5$ works slightly better. Overall in manipulation tasks, **DPPO** training seems relatively robust to this choice.

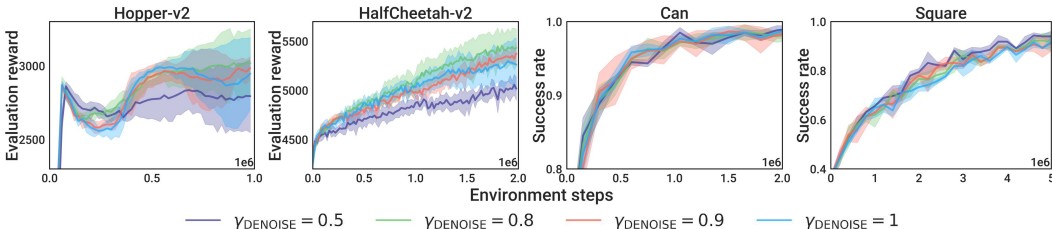

Figure 14: **Choice of denoising discount factor.** Results are averaged over five seeds in Hopper-v2 and HalfCheetah-v2 and three seeds in Can and Square.

---

[6]Psenka et al. (2023) applies off-policy training with double Q-learning (according to its open-source implementation) and policy gradient over the denoising steps. Note that this is a baseline in Psenka et al. (2023) that is conjectured to be inefficient. We follow the same except for applying on-policy PPO updates.

**2. Choice of diffusion noise schedule.** As introduced in Section 4.1, we find it helpful to clip the diffusion noise $\sigma_k$ to a higher minimum value $\sigma_{\min}^{\exp}$ to ensure sufficient exploration. In Figure 15, we perform analysis on varying $\sigma_{\min}^{\exp} \in \{.001, .01, .1, .2\}$ (keeping $\sigma_{\min}^{\text{prob}} = .1$ to evaluate likelihoods). Although in `Can` the choice of $\sigma_{\min}^{\exp}$ does not affect the fine-tuning performance, in `Square` a higher $\sigma_{\min}^{\exp} = 0.1$ is required to prevent the policy from collapsing. We conjecture that this is due to limited exploration causing policy over-optimizing the collected samples that exhibit limited state-action coverage. We also visualize the trajectories at the beginning of fine-tuning in `Avoid` task from D3IL. With higher $\sigma_{\min}^{\exp}$, the trajectories still remain near the two modes of the pre-training data but exhibit a higher coverage in the state space — we believe this additional coverage leads to better exploration. Anecdotally, we find terminating the denoising process early can also provide exploration noise and lead to comparable results, but it requires a more involved implementation around the denoising MDP.

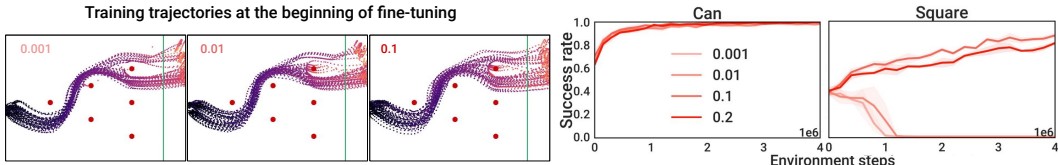

Figure 15: **Choice of minimum diffusion noise.** Results are averaged over three seeds. Note in Left, with higher minimum noise level, the sampled trajectories exhibit wider coverage at the two modes but still maintain the overall structure.

**3. Choice of the number of fine-tuned denoising steps.** We examine how the number of fine-tuned denoising steps in **DPPO**, $K'$, affects the fine-tune performance and wall-clock time in Fig. 16. We show the curves of individual runs (three for each $K'$) instead of the average as their wall-clock times (X-axis) are not perfectly aligned. Generally, fine-tuning too few denoising steps (e.g., 3) can lead to subpar asymptotic performance and slower convergence especially in `Can`. Fine-tuning 10 steps leads to the overall best efficiency. Similar results are also shown in Fig. 19 with `Avoid` task. Lastly, we note that the GPU memory usage scales linearly with $K'$.

We note that the findings here mostly correlate with those from varying the denoising discount factor, $\gamma_{\text{DENOISE}}$. Discounting the earlier denoising steps in the policy gradient can be considered as a soft version of hard limiting the number of fine-tuned denoising steps. Depending on the amount of fine-tuning needed from the pre-trained action distribution, one can flexibly adjust $\gamma_{\text{DENOISE}}$ and $K'$ to achieve the best efficiency.

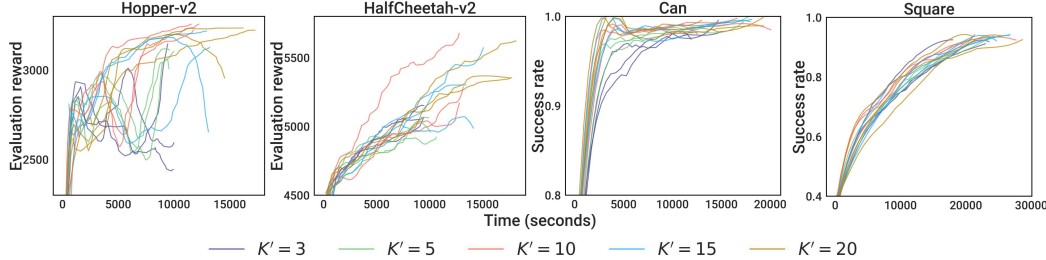

Figure 16: **Choice of number of fine-tuned denoising steps, $K'$.** Individual runs are shown. The curves are smoothed using a Savitzky–Golay filter.

### D.3 EFFECT OF EXPERT DATA

We investigate the effect of the amount of pre-training expert data on fine-tuning performance. In Fig. 17 we compare **DPPO** and Gaussian in `Hopper-v2`, `Square`, and `One-leg` task from FUR­NITURE-BENCH, using varying numbers of expert data (episodes) denoted in the figure. Overall, we find **DPPO** can better leverage the pre-training data and fine-tune to high success rates. Notably, **DPPO** obtains non-trivial performance (60% success rate) on `One-leg` from only 10 episode of demonstrations.

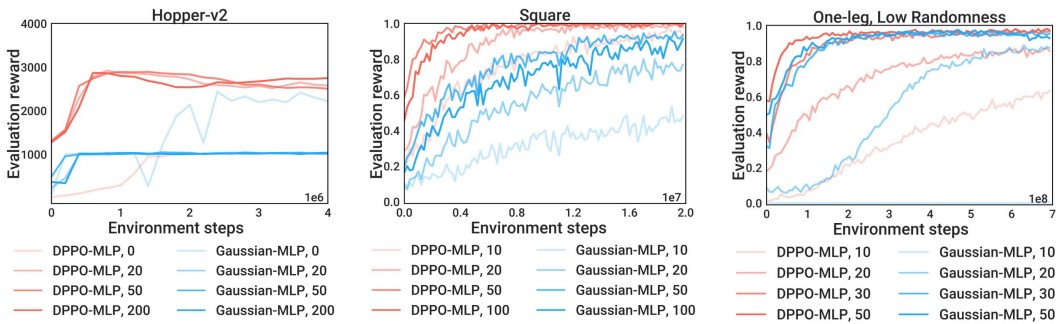

Figure 17: **Varying the number of expert demonstrations.** The numbers in the legends indicates the number of episodes used in pre-training.

**Training from scratch.** In Fig. 18 we compare **DPPO** (10 denoising steps) and Gaussian *trained from scratch* (no pre-training on expert data) in the three OpenAI GYM tasks. As using larger action chunk sizes $T_a$ leads to poor from-scratch training shown in Fig. 17, we focus on single-action chunks $T_a = 1$ (and $T_p = 1$) as is typical in RL benchmarking. Though we find Gaussian trains faster than **DPPO** (expected since **DPPO** solves an MDP with longer effective horizon), **DPPO** still attains reasonable final performance. However, due to the multi-step (10) denoising sampling, **DPPO** takes about $6\times$ wall-clock time compared to Gaussian. We hope that future work will explore how to design the training curriculum of denoising steps for the best balance of training performance and wall-clock efficiency.

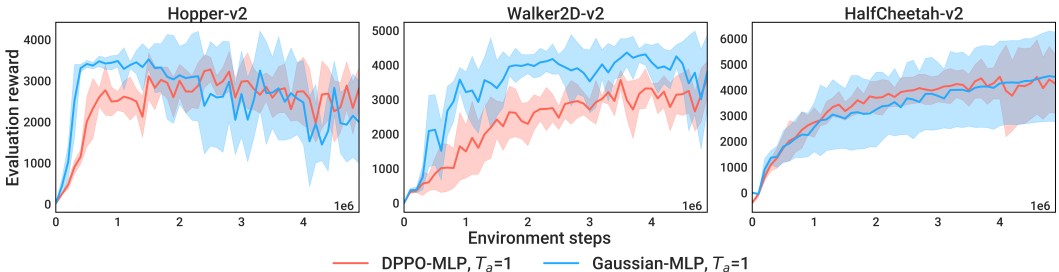

Figure 18: **No expert data / pre-training** with GYM tasks. Results are averaged over five seeds.

## D.4    COMPARING TO OTHER POLICY PARAMETERIZATIONS IN AVOID

Figure 19 depicts the performance of various parameterizations of **DPPO** (with differing numbers of fine-tuned denoising steps, $K'$) to Gaussian and GMM baselines. We study the Avoid task from D3IL, after pre-training with the data from M1, M2, M3 as described in Section 6. We find that, for $K' \in \{15, 20\}$, **DPPO** attains the highest performance of all methods and trains the quickest in terms of environment steps; on M1, M2, it appears to attain the greatest terminal performance as well. $K' = 10$ appears slightly better than, but roughly comparable to, the Gaussian baseline, with GMM and $K' < 10$ performing less strongly.

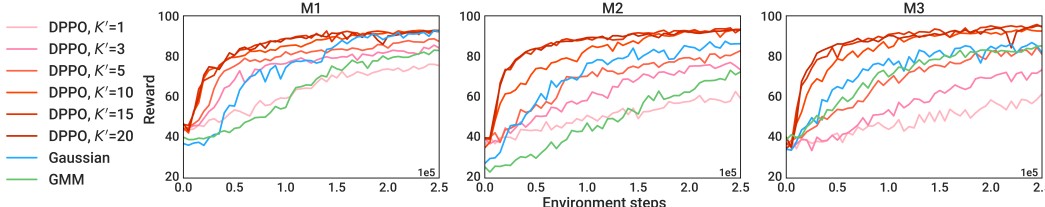

Figure 19: Fine-tuning performance (averaged over five seeds, standard deviation not shown) after pre-training with M1, M2, and M3 in **Avoid task from D3IL**. **DPPO** ($K = 20$), Gaussian, and GMM policies are compared. We also sweep the number of fine-tuned denoising steps $K'$ in **DPPO**.

### D.5 COMPARING TO OTHER POLICY PARAMETERIZATIONS IN THE EASIER TASKS FROM ROBOMIMIC

Figure 20 compares the performance of **DPPO** to Gaussian and GMM baslines, across a variety of architectures, and with **state** and **pixel** inputs, in `Lift` and `Can` environments in the ROBOMIMIC suite. Compared to the `Square` and `Transport` (results shown in Section 5), these environments are considered to be "easier", and this is reflected in the greater performance of **DPPO** and Gaussian baselines (GMM still exhibits subpar performance). Nonetheless, **DPPO** still achieves similar or even better sample efficiency compared to Gaussian baseline.

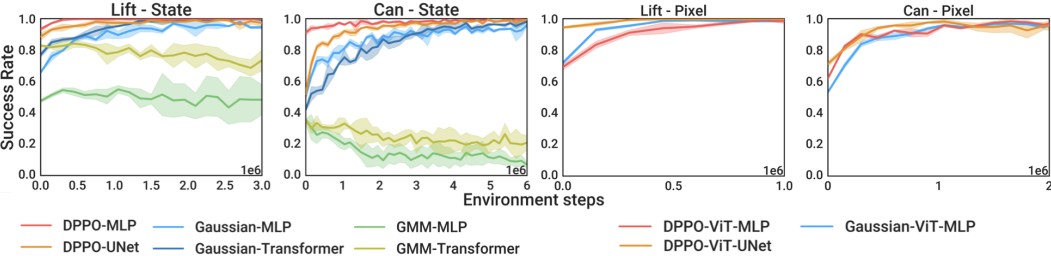

Figure 20: **Comparing to other policy parameterizations** in the easier `Lift` and `Can` tasks from ROBOMIMIC, with **state** (left) or **pixel** (right) observation. Results are averaged over three seeds.

### D.6 COMPARING TO POLICY GRADIENT USING EXACT LIKELIHOOD OF DIFFUSION POLICY

Here we experiment another novel method (which, to our knowledge, has not been explicitly studied in any previous work) for performing policy gradient with diffusion-based policies. Although diffusion model does not directly model the action likelihood, $p_\theta(a_0|s)$, there have been ways to *estimate* the value, e.g., by solving the probability flow ODE that implements DDPM (Song et al., 2020b). We refer the readers to Appendix. D in Song et al. (2020b) for a comprehensive exposition. We follow the official open-source code from Song et al.[7], and implement policy gradient (single-level MDP) that uses the exact action likelihood $\pi_\theta(a_t|s_t)$.

Fig. 21 shows the comparison between **DPPO** and diffusion policy gradient using exact likelihood estimate. Exact policy gradient improves the base policy in `Hopper-v2` but does not outperform **DPPO**. It also requires more runtime and GPU memory as it backpropagates through the ODE. In the more challenging `Can` its success rate drops to zero. Moreover, policy gradient with exact likelihood does not offer the flexibility of fine-tuning fewer-than-$K$ denoising steps or discounting the early denoising steps that **DPPO** offers, which have shown in Appendix D.2 to often improve fine-tuning efficiency.

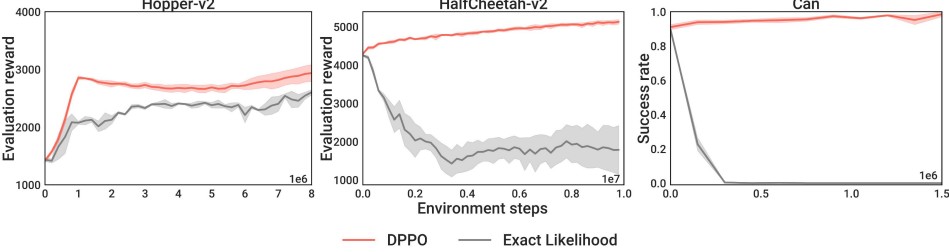

Figure 21: **Comparing to diffusion policy gradient with exact action likelihood.** Results are averaged over five seeds in `Hopper-v2` and `HalfCheetah-v2`, and three seeds in `Can`.

### D.7 ABLATING STRUCTURED EXPLORATION IN **DPPO**

Here we provide additional evidence on how structured exploration of **DPPO** (Section 6) aids RL fine-tuning. While Fig. 19 compares DPPO with Gaussian and GMM policies and shows DPPO

---

[7]https://github.com/yang-song/score_sde_pytorch

trajectories achieve wide coverage and stay near the expert data manifold, in Fig. 22 we ablate such structured exploration within **DPPO**. We use DDIM (Song et al., 2020a) such that actions can be sampled deterministically — this allows us to sample trajectories without adding any noise to intermediate denoising steps but only to the final denoised action ($k = 0$), and compare that to **DPPO** with noise added to all denoising steps. In both cases, we consider the minimum noise level $\sigma_{\min}^{\exp}$ of 0.05 and 0.1. We see in Fig. 22 that with higher noise level, DPPO trajectories cover the expert data modes well without exploring aggressively into new modes, while in the case of only adding noise to the final step, the trajectories become less structured especially in M3.

Then we run both exploration schemes in `Can` and `Square` from ROBOMIMIC, and Fig. 14 right shows the original **DPPO** setup achieves faster convergence than when noise is only added to the final step. This result, on top of results from Section 5.3 showing **DPPO** achiving better sample efficiency than Gaussian and GMM policies, showcases the benefit of structured exploration in fine-tuning.

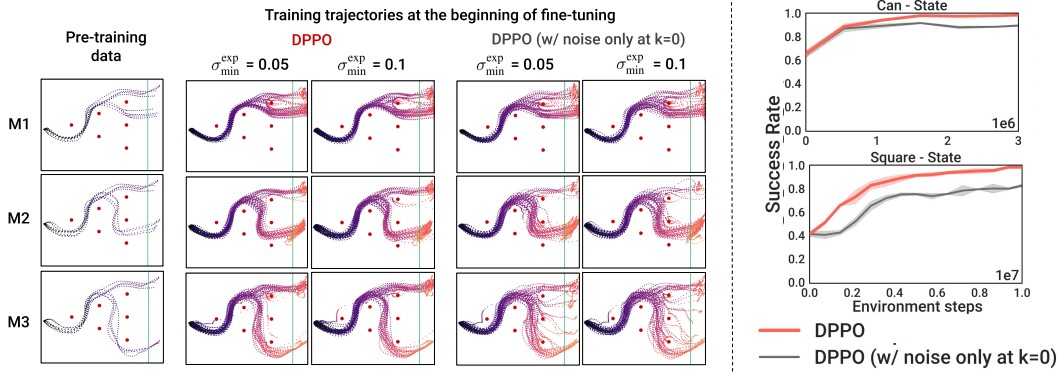

Figure 22: **Ablating Structured Exploration in DPPO.** (Left) Sampled trajectories with noise added to all denoising steps vs. only to the last step $k = 0$ in `Avoid`. (Right) Results are averaged over three seeds in `Can` and `Square`.

## E REPORTING OF WALL-CLOCK TIMES

**Comparing to other diffusion-based RL algorithms Section 5.1.** Table 1 and Table 2 shows the the wall-clock time used in each OpenAI GYM task and ROBOMIMIC task. In GYM tasks, on average **DPPO** trains 41%, 37%, and 12% faster than **DAWR**, **DIPO**, and **DQL**, respectively, which all require a significant amount of gradient updates per sample to train stably. **QSM**, **DRWR**, and **IDQL** trains 43%, 33%, and 7% faster than **DPPO**, respectively. ROBOMIMIC tasks are more expensive to simulate, especially with `Transport` task, and thus the wall-clock difference is smaller among the different methods. All methods use comparable time except for **DIPO** that uses slightly more on average.

| Method | Task | | |
|---|---|---|---|
| | Hopper-v2 | Walker2D-v2 | HalfCheetah-v2 |
| **DRWR** | 11.3 | 12.7 | 10.4 |
| **DAWR** | 30.4 | 30.7 | 27.1 |
| **DIPO** | 27.8 | 27.9 | 26.0 |
| **IDQL** | 16.3 | 16.1 | 15.5 |
| **DQL** | 20.5 | 20.5 | 17.6 |
| **QSM** | 9.6 | 9.9 | 9.7 |
| **DPPO** | 16.6 | 18.3 | 16.8 |

Table 1: **Wall-clock time** in seconds for a single training iteration in **OpenAI GYM tasks** when comparing diffusion-based RL algorithms. Each iteration involves 500 environment timesteps in each of the 40 parallelized environments running on 40 CPU threads and a NVIDIA RTX 2080 GPU (20000 steps total).

| Method | Task | | | |
|--------|------|-----|--------|-----------|
|        | Lift | Can | Square | Transport |
| **DRWR** | 32.5 | 39.5 | 59.8 | 346.1 |
| **DAWR** | 38.6 | 46.0 | 70.5 | 354.3 |
| **DIPO** | 43.9 | 51.6 | 73.3 | 359.7 |
| **IDQL** | 33.8 | 41.7 | 63.7 | 349.9 |
| **DQL**  | 36.9 | 44.4 | 68.5 | 353.5 |
| **QSM**  | 31.8 | 44.5 | 68.7 | 322.5 |
| **DPPO** | 35.2 | 42.0 | 65.6 | 350.3 |

Table 2: **Wall-clock time** in seconds for a single training iteration in **ROBOMIMIC tasks with state input** when comparing diffusion-based RL algorithms. Each iteration involves 4 episodes (1200 environment timesteps for `Lift` and `Can`, 1600 for `Square`, and 3200 for `Transport`) from each of the 50 parallelized environments running on 50 CPU threads and a NVIDIA L40 GPU (60000, 80000, 160000 steps).

**Comparing to other policy parameterizations and architecture** Section 5.3 and Section 5.4. Table 3 and Table 4 shows the wall-clock time used in fine-tuning in each ROBOMIMIC task with state or pixel input, respectively. Gaussian and GMM use similar times and Transformer is slightly more expensive than MLP. On average with state input, **DPPO**-MLP trains 24%, 21%, 24%, and 22% slower than baselines due to the more expensive diffusion sampling. **DPPO**-UNet requires more time with the extensive use of convolutional and normalization layers and trains on average 49% slower than **DPPO**-MLP. On average with pixel input, **DPPO**-ViT-MLP trains 14% slower than Gaussian-ViT-MLP — the difference is smaller than the state input case as the rendering in simulation can be expensive. Table 5 shows the wall-clock time used in FURNITURE-BENCH tasks. **DPPO**-UNet trains 20% slower than Gaussian-MLP on average.

| Method | Task | | | |
|--------|------|-----|--------|-----------|
|        | Lift | Can | Square | Transport |
| Gaussian-MLP | 27.7 | 35.7 | 56.2 | 255.6 |
| Gaussian-Transformer | 29.8 | 37.1 | 57.8 | 266.1 |
| GMM-MLP | 28.0 | 36.2 | 55.2 | 254.5 |
| GMM-Transformer | 29.5 | 37.4 | 58.1 | 260.2 |
| **DPPO**-MLP | 35.6 | 43.3 | 65.0 | 350.5 |
| **DPPO**-UNet | 83.6 | 92.7 | 130.4 | 431.1 |

Table 3: **Wall-clock time** in seconds for a single training iteration in **ROBOMIMIC tasks with state input** when comparing policy parameterizations. Each iteration involves 4 episodes (1200 environment timesteps for `Lift` and `Can`, 1600 for `Square`, and 3200 for `Transport`) from each of the 50 parallelized environments running on 50 CPU threads and a NVIDIA L40 GPU (60000, 80000, 160000 steps).

| Method | Task | | | |
|--------|------|-----|--------|-----------|
|        | Lift | Can | Square | Transport |
| Gaussian-ViT-MLP | 153.6 | 173.1 | 277.0 | 770.0 |
| **DPPO**-ViT-MLP | 194.9 | 202.5 | 328.5 | 871.3 |

Table 4: **Wall-clock time** in seconds for a single training iteration in **ROBOMIMIC tasks with pixel input** when comparing policy parameterizations. Each iteration involves 4 episodes (1200 environment timesteps for `Lift` and `Can`, 1600 for `Square`, and 3200 for `Transport`) from each of the 50 parallelized environments running on 50 CPU threads and a NVIDIA L40 GPU (60000, 80000, 160000 steps).

| Method | Task | | |
|---|---|---|---|
| | One-leg | Lamp | Round-table |
| Gaussian-MLP | 101.8 | 202.8 | 168.7 |
| **DPPO**-UNet | 148.4 | 258.2 | 188.6 |

Table 5: **Wall-clock time** in seconds for a single training iteration in **FURNITURE-BENCH tasks** when comparing policy parameterizations. Each iteration involves 1 episodes (700 environment timesteps for `One-leg`, and 1000 for `Lamp` and `Round-table`) from each of the 1000 parallelized environments running on a NVIDIA L40 GPU (700000, 1000000, 1000000 steps).

# F  ADDITIONAL EXPERIMENTAL DETAILS

| Task / Dataset | | Obs dim - State | Obs dim - Pixel | Act dim | $T$ | Sparse reward ? |
|---|---|---|---|---|---|---|
| GYM | Hopper-v2 | 11 | - | 3 | 1000 | No |
| | Walker2D-v2 | 17 | - | 6 | 1000 | No |
| | HalfCheetah-v2 | 17 | - | 6 | 1000 | No |
| FRANKA-KITCHEN | Kitchen-Complete-v0 | 60 | - | 9 | 280 | Yes |
| | Kitchen-Partial-v0 | 60 | - | 9 | 280 | Yes |
| | Kitchen-Mixed-v0 | 60 | - | 9 | 280 | Yes |
| ROBOMIMIC, state input | Lift | 19 | - | 7 | 300 | Yes |
| | Can | 23 | - | 7 | 300 | Yes |
| | Square | 23 | - | 7 | 400 | Yes |
| | Transport | 59 | - | 14 | 800 | Yes |
| ROBOMIMIC, pixel input | Lift | 9 | 96×96 | 7 | 300 | Yes |
| | Can | 9 | 96×96 | 7 | 300 | Yes |
| | Square | 9 | 96×96 | 7 | 400 | Yes |
| | Transport | 18 | 2×96×96 | 14 | 800 | Yes |
| FURNITURE-BENCH | One-leg | 58 | - | 10 | 700 | Yes |
| | Lamp | 44 | - | 10 | 1000 | Yes |
| | Round-table | 44 | - | 10 | 1000 | Yes |
| D3IL | M1 | 4 | - | 2 | 100 | Yes |
| | M2 | 4 | - | 2 | 100 | Yes |
| | M3 | 4 | - | 2 | 100 | Yes |

Table 6: **Comparison of the different tasks considered.** "Obs dim - State": dimension of the state observation input. "Obs dim - State": dimension of the pixel observation input. "Act dim - State": dimension of the action space. $T$: maximum number of steps in an episode. "Sparse reward ?": whether sparse reward is used in training instead of dense reward.

## F.1  DETAILS OF POLICY ARCHITECTURES USED IN ALL EXPERIMENTS

**MLP.**  For most of the experiments, we use a Multi-layer Perceptron (MLP) with two-layer residual connection as the policy head. For diffusion-based policies, we also use a small MLP encoder for the state input and another small MLP with sinusoidal positional encoding for the denoising timestep input. Their output features are then concatenated before being fed into the MLP head. Diffusion Policy, proposed by Chi et al. (2024b), does not use MLP as the diffusion architecture, but we find it delivers comparable (or even better) pre-training performance compared to UNet.

**Transformer.**  For comparing to other policy parameterizations in Section 5.3, we also consider Transformer as the policy architecture for the Gaussian and GMM baselines. We consider decoder only. No dropout is used. A learned positional embedding for the action chunk is the sequence into the decoder.

**UNet.**  For comparing to other policy parameterizations in Section 5.3, we also consider UNet (Ronneberger et al., 2015) as a possible architecture for DP. We follow the implementation from Chi et al. (2024b) that uses sinusoidal positional encoding for the denoising timestep input, except for using a larger MLP encoder for the observation input in each convolutional block. We find this modification helpful in more challenging tasks.

**ViT.** For pixel-based experiments in Section 5.3 we use Vision-Transformer(ViT)-based image encoder introduced by Hu et al. (2023) before an MLP head. Proprioception input is appended to each channel of the image patches. We also follow (Hu et al., 2023) and use a learned spatial embedding for the ViT output to greatly reduce the number of features, which are then fed into the downstream MLP head.

## F.2 ADDITIONAL DETAILS OF GYM TASKS AND TRAINING IN SECTION 5.1

**Pre-training.** The observations and actions are normalized to $[0, 1]$ using min/max statistics from the pre-training dataset. For all three tasks the policy is trained for 3000 epochs with batch size 128, learning rate of 1e-3 decayed to 1e-4 with a cosine schedule, and weight decay of 1e-6. Exponential Moving Average (EMA) is applied with a decay rate of 0.995.

**Fine-tuning.** All methods from Section 5.1 use the same pre-trained policy. Fine-tuning is done using online experiences sampled from 40 parallelized MuJoCo environments (Todorov et al., 2012). Reward curves shown in Fig. 4 are evaluated by running fine-tuned policies with $\sigma_{\min}^{\exp} = 0.001$ (i.e., without extra noise) for 40 episodes. Each episode terminates if the default conditions are met or the episode reaches 1000 timesteps. Detailed hyperparameters are listed in Table 7.

## F.3 DESCRIPTIONS OF DIFFUSION-BASED RL ALGORITHM BASELINES IN SECTION 5.1

**DRWR:** This is a **customized** reward-weighted regression (RWR) algorithm Peters and Schaal (2007) that fine-tunes a pre-trained DP with a supervised objective with higher weights on actions that lead to higher reward-to-go $r$.

The reward is scaled with $\beta$ and the exponentiated weight is clipped at $w_{\max}$. The policy is updated with experiences collected with the current policy (no buffer for data from previous iteration) and a replay ratio of $N_\theta$. No critic is learned.

$$\mathcal{L}_\theta = \mathbb{E}^{\bar{\pi}_\theta, \varepsilon_t}\Big[\min(e^{\beta r_t}, w_{\max})\|\varepsilon_t - \varepsilon_\theta(a_t^0, s_t, k)\|^2\Big].$$

**DAWR:** This is a **customized** advantage-weighted regression (AWR) algorithm Peng et al. (2019) that builds on **DRWR** but uses TD-bootstrapped Sutton and Barto (2018) advantage estimation instead of the higher-variance reward-to-go for better training stability and efficiency. **DAWR** (and **DRWR**) can be seen as approximately optimizing (4.2) with a Kullback–Leibler (KL) divergence constraint on the policy Peng et al. (2019); Black et al. (2023).

The advantage is scaled with $\beta$ and the exponentiated weight is clipped at $w_{\max}$. Unlike **DRWR**, we follow (Peng et al., 2019) and trains the actor in an off-policy manner: recent experiences are saved in a replay buffer $\mathcal{D}$, and the actor is updated with a replay ratio of $N_\theta$.

$$\mathcal{L}_\theta = \mathbb{E}^{\mathcal{D}, \varepsilon_t}\big[\min(e^{\beta \hat{A}_\phi(s_t, a_t^0)}, w_{\max})\|\varepsilon_t - \varepsilon_\theta(a_t^0, s_t, k)\|^2\big].$$

The critic is updated less frequently (we find diffusion models need many gradient updates to fit the actions) with a replay ratio of $N_\phi$.

$$\mathcal{L}_\phi = \mathbb{E}^{\mathcal{D}}\big[\|\hat{A}_\phi(s_t, a_t^0) - A(s_t, a_t^0)\|^2\big],$$

where $A$ is calculated using TD($\lambda$), with $\lambda$ as $\lambda_{\mathrm{DAWR}}$ and the discount factor $\gamma_{\mathrm{ENV}}$.

**DIPO (Yang et al., 2023):** This baseline applies "action gradient" that uses a learned state-action Q function to update the actions saved in the replay buffer, and then has DP fitting on them without weighting.

Similar to **DAWR**, recent experiences are saved in a replay buffer $\mathcal{D}$. The actions ($k = 0$) in the buffer are updated for $M_{\mathrm{DIPO}}$ iterations with learning rate $\alpha_{\mathrm{DIPO}}$.

$$a_t^{m+1, k=0} = a_t^{m, k=0} + \alpha_{\mathrm{DIPO}}\nabla_\phi \hat{Q}_\phi(s_t, a_t^{m, k=0}), \ m = 0, \ldots, M_{\mathrm{DIPO}} - 1.$$

The actor is then updated with a replay ratio of $N_\theta$.

$$\mathcal{L}_\theta = \mathbb{E}^{\mathcal{D}}\big[\|\varepsilon_t - \varepsilon_\theta(a_t^{M_{\mathrm{DIPO}}, k=0}, s_t, k)\|^2\big].$$

The critic is trained to minimize the Bellman residual with a replay ratio of $N_\phi$. Double Q-learning is also applied.

$$\mathcal{L}_\phi = \mathbb{E}^{\mathcal{D}}\big[\|(R_t + \gamma_{\mathrm{ENV}}\hat{Q}_\phi(s_{t+1}, \bar{\pi}_\theta(a_{t+1}^{k=0}|s_{t+1})) - \hat{Q}_\phi(s_t, a_t^{m=0, k=0})\|^2\big]$$

**IDQL (Hansen-Estruch et al., 2023):** This baseline learns a state-action Q function and state V function to choose among the sampled actions from DP. DP fits on new samples without weighting.

Again recent experiences are saved in a replay buffer $\mathcal{D}$. The state value function is updated to match the expected Q value with an expectile loss, with a replay ratio of $N_\psi$.

$$\mathcal{L}_\psi = \mathbb{E}^{\mathcal{D}}\big[|\tau_{\text{IDQL}} - \mathbb{1}(\hat{Q}_\phi(s_t, a_t^0) < \hat{V}_\psi^2(s_t))|\big].$$

The value function is used to update the Q function with a replay ratio of $N_\phi$.

$$\mathcal{L}_\phi = \mathbb{E}^{\mathcal{D}}\big[\|(R_t + \gamma_{\text{ENV}}\hat{V}_\psi(s_{t+1}) - \hat{Q}_\phi(s_t, a_t^0)\|^2\big].$$

The actor fits all sampled experiences without weighting, with a replay ratio of $N_\theta$.

$$\mathcal{L}_\theta = \mathbb{E}^{\mathcal{D}}\big[\|\varepsilon_t - \varepsilon_\theta(a_t^0, s_t, k)\|^2\big].$$

At inference time, $M_{\text{IDQL}}$ actions are sampled from the actor. For training, Boltzmann exploration is applied based on the difference between $Q$ value of the sampled actions and and the $V$ value at the current state. For evaluation, the greedy action under $Q$ is chosen.

**DQL (Wang et al., 2022):** This baseline learns a state-action Q function and backpropagates the gradient from the critic through the entire actor (with multiple denoising steps), akin to the usual Q-learning.

Again recent experiences are saved in a replay buffer $\mathcal{D}$. The actor is then updated using both a supervised loss and the value loss with a replay ratio of $N_\theta$.

$$\mathcal{L}_\theta = \mathbb{E}^{\mathcal{D}}\big[\|\varepsilon_t - \varepsilon_\theta(a_t^0, s_t, k)\|^2 - \alpha_{\text{DQL}}\hat{Q}_\phi(s_t, \bar{\pi}_\theta(a_t^0|s_t))\big],$$

where $\alpha_{\text{DQL}}$ is a weighting coefficient. The critic is trained to minimize the Bellman residual with a replay ratio of $N_\phi$. Double Q-learning is also applied.

$$\mathcal{L}_\phi = \mathbb{E}^{\mathcal{D}}\big[\|(R_t + \gamma_{\text{ENV}}\hat{Q}_\phi(s_{t+1}, \bar{\pi}_\theta(a_{t+1}^0|s_{t+1})) - \hat{Q}_\phi(s_t, a_t^0)\|^2\big]$$

**QSM (Psenka et al., 2023):** This baselines learns a state-action Q function, and then updates the actor by aligning the score of the diffusion actor with the gradient of the Q function.

Again recent experiences are saved in a replay buffer $\mathcal{D}$. The critic is trained to minimize the Bellman residual with a replay ratio of $N_\phi$. Double Q-learning is also applied.

$$\mathcal{L}_\phi = \mathbb{E}^{\mathcal{D}}\big[\|(R_t + \gamma_{\text{ENV}}\hat{Q}_\phi(s_{t+1}, \bar{\pi}_\theta(a_{t+1}^0|s_{t+1})) - \hat{Q}_\phi(s_t, a_t^0)\|^2\big].$$

The actor is updated as follows with a replay ratio of $N_\theta$.

$$\mathcal{L}_\theta = \mathbb{E}^{\mathcal{D}}\big[\|\alpha_{\text{QSM}}\nabla_a\hat{Q}_\phi(s_t, a_t) - (-\varepsilon_\theta(a_t^0, s_t, k))\|^2\big],$$

where $\alpha_{\text{QSM}}$ scales the gradient. The negative sign before $\varepsilon_\theta$ is from taking the gradient of the mean $\mu$ in the denoising process.

## F.4 DESCRIPTIONS OF RL FINE-TUNING ALGORITHM BASELINES IN SECTION 5.2

In this subsection, we detail the baselines **RLPD**, **Cal-QL**, and **IBRL**. All policies $\pi_\theta$ are parameterized as unimodal Gaussian.

**RLPD (Ball et al., 2023):** This baseline is based on Soft Actor Critic (SAC, Haarnoja et al. (2018)) — it learns an entropy-regularized state-action Q function, and then updates the actor by maximizing the Q function w.r.t. the action.

A replay buffer $\mathcal{D}$ is initialized with offline data, and online samples are added to $\mathcal{D}$. Each gradient update uses a batch of mixed 50/50 offline and online data. An ensemble of $N_{\text{critic}}$ critics is used, and at each gradient step two critics are randomly chosen. The critics are trained to minimize the Bellman residual with replay ratio $N_\phi$:

$$\mathcal{L}_\phi = \mathbb{E}^{\mathcal{D}}\big[\|(R_t + \gamma_{\text{ENV}}\hat{Q}_{\phi'}(s_{t+1}, \pi_\theta(a_{t+1}|s_{t+1})) - \hat{Q}_\phi(s_t, a_t)\|^2\big].$$

The target critic parameter $\phi'$ is updated with delay. The actor minimizes the following loss with a replay ratio of 1:

$$\mathcal{L}_\theta = \mathbb{E}^{\mathcal{D}}\big[-\hat{Q}_\phi(s_t, a_t) + \alpha_{\text{ent}}\log\pi_\theta(a_t|s_t)\big],$$

where $\alpha_{\text{ent}}$ is the entropy coefficient (automatically tuned as in SAC starting at 1).

**Cal-QL (Nakamoto et al., 2024):** This baseline trains the policy $\mu$ and the action-value function $Q^\mu$ in an offline phase and then an online phase. During the offline phase only offline data is sampled for gradient update, while during the online phase mixed 50/50 offline and online data are sampled. The critic is trained to minimize the following loss (Bellman residual and calibrated Q-learning):

$$\mathcal{L}_\phi = \mathbb{E}^{\mathcal{D}}\big[\|(R_t + \gamma_{\text{ENV}}\hat{Q}_{\phi'}(s_{t+1}, \pi_\theta(a_{t+1}|s_{t+1}))) - \hat{Q}_\phi(s_t, a_t)\|^2\big]$$
$$+ \beta_{\text{cql}}(\mathbb{E}^{\mathcal{D}}\big[\max(Q_\phi(s_t, a_t), V(s_t))\big] - \mathbb{E}^{\mathcal{D}}\big[Q_\phi(s_t, a_t)\big]),$$

where $\beta_{\text{cql}}$ is a weighting coefficient between Bellman residual and calibration Q-learning and $V(s_t)$ is estimated using Monte-Carlo returns. The target critic parameter $\phi'$ is updated with delay. The actor minimizes the following loss:

$$\mathcal{L}_\theta = \mathbb{E}^{\mathcal{D}}\big[-\hat{Q}_\phi(s_t, a_t) + \alpha_{\text{ent}}\log\pi_\theta(a_t|s_t)\big],$$

where $\alpha_{\text{ent}}$ is the entropy coefficient (automatically tuned as in SAC starting at 1).

**IBRL (Hu et al., 2023):** This baseline first pre-trains a policy $\mu_\psi$ using behavior cloning, and for fine-tuning it trains a RL policy $\pi_\theta$ initialized as $\mu_\psi$. During fine-tuning recent experiences are saved in a replay buffer $\mathcal{D}$. An ensemble of $N_{\text{critic}}$ critics is used, and at each gradient step two critics are randomly chosen. The critics are trained to minimize the Bellman residual with replay ratio $N_\phi$:

$$\mathcal{L}_\phi = \mathbb{E}^{\mathcal{D}}\big[\|(R_t + \gamma_{\text{ENV}}\max_{a' \in \{a^{IL}, a^{RL}\}}\hat{Q}_{\phi'}(s_{t+1}, a') - \hat{Q}_\phi(s_t, a_t)\|^2\big]$$

where $a^{IL} = \mu_\psi(s_{t+1})$ (no noise) and $a^{RL} \sim \pi_{\theta'}(s_{t+1})$, and $\pi_{\theta'}$ is the target actor. The target critic parameter $\phi'$ is updated with delay. The actor minimizes the following loss with a replay ratio of 1:

$$\mathcal{L}_\theta = -\mathbb{E}^{\mathcal{D}}\big[\hat{Q}_\phi(s_t, a_t)\big].$$

The target actor parameter $\theta'$ is also updated with delay.

### F.5 ADDITIONAL DETAILS OF FRANKA-KITCHEN TASKS AND TRAINING IN SECTION 5.2

**Tasks.** We consider three settings from the D4RL benchmark (Fu et al., 2020): (1) `Kitchen-Complete-v0` containing demonstrations that complete the entire task (four subtasks), (2) `Kitchen-Partial-v0` containing some complete demonstrations and many ones completing only subtasks, and (3) `Kitchen-Mixed-v0` containing incomplete demonstrations only.

**Pre-training.** The observations and actions are normalized to $[0, 1]$ using min/max statistics from the pre-training dataset. No history observation (proprioception or ground-truth object states) is used. All policies are trained with batch size 128, learning rate 1e-4 decayed to 1e-5 with a cosine schedule, and weight decay 1e-6. **DPPO** policies are trained with 8000 epochs. For **IBRL** and **Cal-QL** we follow the hyperparameters from the original implementations — **IBRL** proposes using (1) wider MLP layers and (2) dropout during pre-training, which we follow too. We use $T_a = 4$ for **DPPO**; we also tried to use the same action chunk size with **IBRL**, **RLPD**, and **Cal-QL**, but we find for all of them $T_a = 1$ leads to better performance.

**Fine-tuning.** With **DPPO**, policies are fine-tuned using online experiences sampled from 40 parallelized MuJoCo environments (Todorov et al., 2012), while the baselines use only one environment (matching their original implementations). Episodes terminates when they reach maximum episode lengths (shown in Table 6) or all four subtasks are completed. Detailed hyperparameters are listed in Table 8 — we follow the hyperparameter choices from the original implementations of the baselines.

**Larger variance with DPPO in Fig. 11.** In Fig. 11, it is shown that **DPPO** exhibits a larger variance in normalized score with `Kitchen-Partial-v0` than **Cal-QL**. This is due to **DPPO** solving either 3/4 or 2/4 subtasks in one seed (low variance within the evaluation episodes in one seed) but high variance over seeds, whereas **Cal-QL** has higher variance among evaluation episodes in one seed but on average over seeds it shows lower variance. This also highlights a notable property of **DPPO**: `Kitchen-Partial-v0` and `Kitchen-Mixed-v0` have trajectories only completing subtasks, thus being highly multi-modal. Diffusion policy can sometimes struggle to

learn all the modes from pre-training, and since **DPPO** directly fine-tunes the pre-trained policy, it can fail to converge to 100% success rate at fine-tuning. **Cal-QL** instead learns from all offline data during fine-tuning in an off-policy manner, thus less sensitive to pre-training performance. Nonetheless, with offline data completing tasks consistently despite varying quality (ROBOMIMIC and Kitchen-Complete-v0, which, we believe, are more realistic in the current paradigm of robot manipulation), **DPPO** demonstrates much better final performance than **Cal-QL** and other baselines in Fig. 11.

## F.6 ADDITIONAL DETAILS OF ROBOMIMIC TASKS AND TRAINING IN SECTION 5.3

**Tasks.** We consider four tasks from the ROBOMIMIC benchmark (Mandlekar et al., 2021): (1) Lift: lifting a cube from the table, (2) Can: picking up a Coke can and placing it at a target bin, (3) Square: picking up a square nut and place it on a rod, and (4) Transport: two robot arms removing a bin cover, picking and placing a cube, and then transferring a hammer from one container to another one.

**Pre-training.** ROBOMIMIC provides the Multi-Human (MH) dataset with noisy human demonstrations for each task, which we use to pre-train the policies. The observations and actions are normalized to $[0, 1]$ using min/max statistics from the pre-training dataset. No history observation (pixel, proprioception, or ground-truth object states) is used. All policies are trained with batch size 128, learning rate 1e-4 decayed to 1e-5 with a cosine schedule, and weight decay 1e-6. Diffusion-based policies are trained with 8000 epochs, while Gaussian and GMM policies are trained with 5000 epochs — we find diffusion models require more gradient updates to fit the data well.

**Fine-tuning.** Diffusion-based, Gaussian, and GMM pre-trained policies are then fine-tuned using online experiences sampled from 50 parallelized MuJoCo environments (Todorov et al., 2012). Success rate curves shown in Fig. 4, Fig. 5, and Fig. 20 are evaluated by running fine-tuned policies with $\sigma_{\min}^{\exp} = 0.001$ (i.e., without extra noise) for 50 episodes. Episodes terminates only when they reach maximum episode lengths (shown in Table 6). Detailed hyperparameters are listed in Table 9.

**Pixel training.** We use the wrist camera view in Lift and Can, the third-person camera view in Square, and the two robot shoulder camera views in Transport. Random-shift data augmentation is applied to the camera images during both pre-training and fine-tuning. Gradient accumulation is used in fine-tuning so that the same batch size (as in state-input training) can fit on the GPU. Detailed hyperparameters are listed in Table 10.

## F.7 DESCRIPTIONS OF POLICY PARAMETERIZATION BASELINES IN SECTION 5.3

**Gaussian.** We consider unimodal Gaussian with diagonal covariance, the most commonly used policy parameterization in RL. The standard deviation for each action dimension, $\sigma_{\text{Gau}}$, is fixed during pre-training; we also tried to learn $\sigma_{\text{Gau}}$ from the dataset but we find the training very unstable. During fine-tuning $\sigma_{\text{Gau}}$ is learned starting from the same fixed value and also clipped between $0.01$ and $0.2$. Additionally we clip the sampled action to be within 3 standard deviation from the mean. As discusses in Appendix C, we choose the PPO clipping ratio $\varepsilon$ based on the empirical clipping fraction in each task. This setup is also used in the FURNITURE-BENCH experiments. We note that we spend significant amount of efforts tuning the Gaussian baseline, and our results with it are some of the best known ones in RL training for long-horizon manipulation tasks (exceeding our initial expectations), e.g., reaching $\sim$100% success rate in Lamp with Low randomness.

**GMM.** We also consider Gaussian Mixture Model as the policy parameterization. We denote $M_{\text{GMM}}$ the number of mixtures. The standard deviation for each action dimension in each mixture, $\sigma_{\text{GMM}}$, is also fixed during pre-training. Again during fine-tuning $\sigma_{\text{GMM}}$ is learned starting from the same fixed value and also clipped between $0.01$ and $0.2$.

## F.8 ADDITIONAL DETAILS OF FURNITURE-BENCH TASKS AND TRAINING IN SECTION 5.4

**Tasks.** We consider three tasks from the FURNITURE-BENCH benchmark (Heo et al., 2023): (1) One-leg: assemble one leg of a table by placing the tabletop in the fixture corner, grasping and

inserting the table leg, and screwing in the leg, (2) `Lamp`: place the lamp base in the fixture corner, grasp, insert, and screw in the light bulb, and finally place the lamp shade, (3) `Round-table`: place a round tabletop in the fixture corner, insert and screw in the table leg, and then insert and screw in the table base. See Fig. 23 for the visualized rollouts in simulation.

**Pre-training.** The pre-training dataset is collected in the simulated environments using a Space-Mouse[8], a 6 DoF input device. The simulator runs at 10Hz. At every timestep, we read off the state of the SpaceMouse as $\delta\mathbf{a} = [\Delta x, \Delta y, \Delta z, \Delta\text{roll}, \Delta\text{pitch}, \Delta\text{yaw}]$, which is converted to a quaternion before passed to the environment step and stored as the action alongside the current observation in the trajectory. If $|\Delta\mathbf{a}_i| < \varepsilon\ \forall i$ for some small $\varepsilon = 0.05$ defining the threshold for a no-op, we do not record any action nor pass it to the environment. Discarding no-ops is important for allowing the policies to learn from demonstrations effectively. When the desired number of demonstrations has been collected (typically 50), we process the actions to convert the delta actions stored from the SpaceMouse into absolute pose actions by applying the delta action to the current EE pose at each timestep.

The observations and actions are normalized to $[-1, 1]$ using min/max statistics from the pre-training dataset. No history observation (proprioception or ground-truth object states) is used, i.e., only the current observation is passed to the policy. All policies are trained with batch size 256, learning rate 1e-4 decayed to 1e-5 with a cosine schedule, and weight decay 1e-6. Diffusion-based policies are trained with 8000 epochs, while Gaussian policies are trained with 3000 epochs. Gaussian policies can easily overfit the pre-trained dataset, while diffusion-based policies are more resilient. Gaussian policies also require a very large MLP ($\sim$10 million parameters) to fit the data well.

**Fine-tuning.** Diffusion-based and Gaussian pre-trained policies are then fine-tuned using online experiences sampled from 1000 parallelized IsaacGym environments Makoviychuk et al. (2021). Success rate curves shown in Fig. 6 are evaluated by running fine-tuned policies with $\sigma_{\min}^{\exp} = 0.001$ (i.e., without extra noise) for 1000 episodes. Episodes terminate only when they reach maximum episode length (shown in Table 6). Detailed hyperparameters are listed in Table 11. We find a smaller amount of exploration noise (we set $\sigma_{\min}^{\exp}$ and $\sigma_{\text{Gau}}$ to be 0.04) is necessary for the pre-trained policy achieving nonzero success rates at the beginning of fine-tuning.

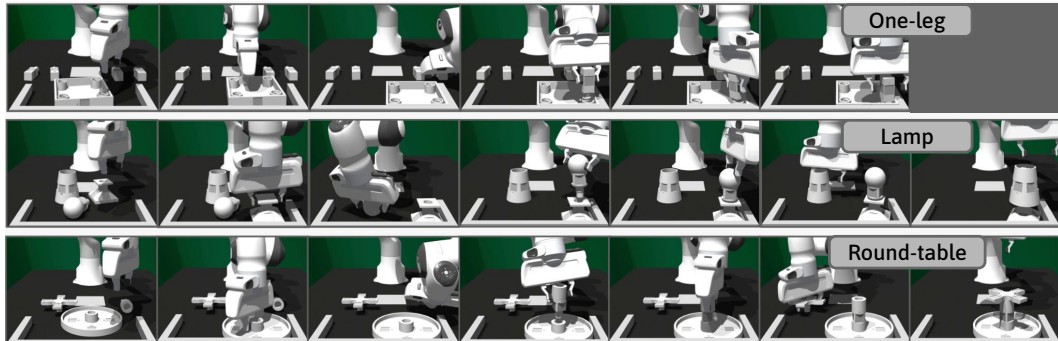

Figure 23: Representative rollouts from simulated FURNITURE-BENCH tasks.

**Hardware setup - robot control.** The physical robot used is a Franka Emika Panda arm. The policies output a sequence of desired end-effector poses in the robot base frame to control the robot. These poses are converted into joint position targets through differential inverse kinematics. We calculate the desired end-effector velocity as the difference between the desired and current poses divided by the delta time $dt = 1/10$. We then convert this to desired joint velocities using the Jacobian and compute the desired joint positions with a first-order integration over the current joint positions and desired velocity. The resulting joint position targets are passed to a low-level joint impedance controller provided by Polymetis (Lin et al., 2021), running at 1kHz.

---

[8]https://3dconnexion.com/us/product/spacemouse-wireless/

**Hardware setup - state estimation.**    To deploy state-based policies on real hardware, we utilize AprilTags (Wang and Olson, 2016) for part pose estimation. The FURNITURE-BENCH (Heo et al., 2023) task suite provides AprilTags for each part and code for estimating part poses from tag detections. The process involves several steps: (1) detecting tags in the camera frame, (2) mapping tag detections to the robot frame for policy compatibility, (3) utilizing known offsets between tags and object centers in the simulator, and (4) calibrating the camera pose using an AprilTag at a known position relative to the robot base. Despite general accuracy, detections can be noisy, especially during movement or partial occlusion, which the `One-leg` task features. Since the task requires high precision, we find the following to help make the estimation reliable enough:

- **Camera coverage:** We find detection quality sensitive to distance and angle between the camera and tag. This issue is likely due to the RealSense D435 camera having mediocre image quality and clarity and the relatively small tags. To remedy this, we opt to use 4 cameras roughly evenly spread out around the scene to ensure that at least one camera has a solid view of a tag on all the parts (i.e., as close as possible with a straight-on view). To find the best camera positions, we start with having a camera in each of the cardinal directions around the scene. Then, we adjust the pose of each to get it as close as possible to the objects while still covering the necessary workspace and capturing the base tag for calibration. Moving the robot arm around the scene to avoid the worst occlusion is also helpful.

- **Lighting:** Even with better camera coverage and placement, detection quality depends on having crisp images. We find proper lighting helpful to improve image quality. In particular, the scene should be well and evenly lit around the scene without causing reflections in either the tag or table.

- **Filtering:** Bad detections can sometimes cause the resulting pose estimate to deviate significantly from the true pose, i.e., jumping several centimeters from one frame to the next. This usually only happens on isolated frames, and thus before "accepting" a given detection, we check if the new position and orientation are within 5 cm and 20 degrees of the previously accepted pose. In addition, we apply low-pass filtering on the detection using a simple exponential average (with $\alpha = 0.25$) to smooth out the high-frequency noise.

- **Averaging:** The objects have multiple tags that can be detected from multiple cameras. After performing the filtering step, we average all pose estimates for the same object across different tags and cameras, which also helps smooth out noise. This alone, however, does not fully cancel the case when a single detection has a large jump, as this can severely skew the average, still necessitating a filtering step. Having multiple cameras benefits this step, too, as it provides more detections to average over.

- **Caching part pose in hand:** A particularly difficult phase of the task to achieve good detections is when the robot transports the table leg from the initial position to the tabletop for insertion. The main problems are that the movement can blur the images, and the grasping can cause occlusions. Therefore, we found it helpful to assume that once the part was grasped by the robot, it would not move in the grasp until the gripper opened. With this, we can "cache" the pose of the part relative to the end-effector once the object is fully grasped and use this instead of relying on detections during the movement.

- **Normalization pitfalls and clipping:** We generally use min-max normalization of the state observations to ensure observations are in $[-1, 1]$. The tabletop part moves very little in the $z$-direction demonstration data, meaning the resulting normalization limits (the minimum and maximum value of the data) can be very close, $x_{\max} - x_{\min} \approx 0$. With these tight limits, the noise in the real-world detection can be amplified greatly as $x_{\mathrm{norm}} = \frac{x - x_{\min}}{x_{\max} - x_{\min}}$. Therefore, ensure that normalization ranges are reasonable. As an extra safeguard, clipping the data to $[-1, 1]$ can also help.

- **Only estimate necessary states:** Despite the `One-leg` task having 5 parts, only 2 are manipulated. Only estimating the pose of those parts can eliminate a lot of noise. In particular, the pose of the 3 legs that are not used and the obstacle (the U-shaped fixture) can be set to an arbitrary value from the dataset.

- **Visualization for debugging:** We use the visualization tool MeshCat[9] extensively for debugging of state estimation. The tool allows for easy visualizations of poses of all relevant objects in the

---

[9] https://github.com/meshcat-dev/meshcat

scene, like the robot end-effector and parts, which makes sanity-checking the implementation far easier than looking at raw numbers.

**Hardware evaluation.** We perform 20 trials for each method. We adopt a single-blind model selection process: at the beginning of each trial, we first randomize the initial state. Then, we randomly select a method and roll it out, but the experimenter does not observe which model is used. We record the success and failure of each trial and then aggregate statistics for each model after all trials are completed.

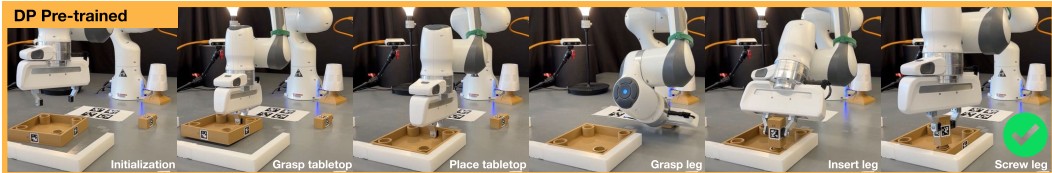

**(A)** Pre-trained Diffusion policy performs successful rollout

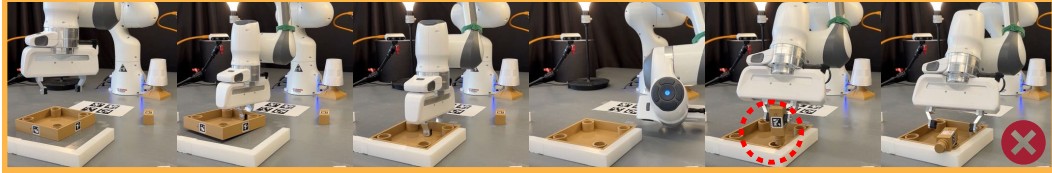

**(B)** Policy pushes peg down without proper alignment with the hole before releasing the peg, making it topple over

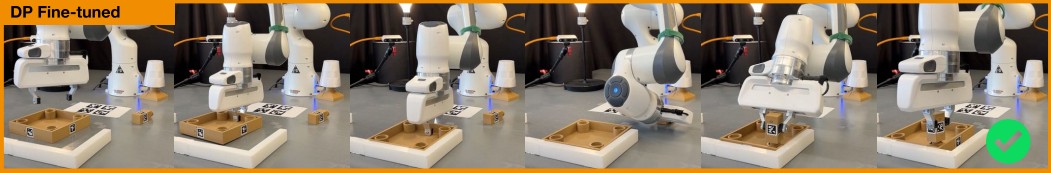

**(C)** Fine-tuned DPPO policy performs successful rollout

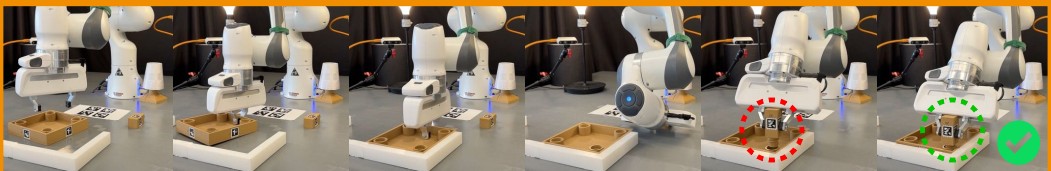

**(D)** Initial peg alignment is off, the policy corrects placement until it is properly inserted in the hole before letting go

Figure 24: **Qualitative comparison of pre-trained vs. fine-tuned `DPPO` policies in real evaluation.** **(A)** Successful rollout with the pre-trained policy. **(B)** Failed rollout with the pre-trained policy due to imprecise insertion. **(C)** Successful rollout with the fine-tuned policy. **(D)** Successful rollout with the fine-tuned policy that requires corrective behavior.

**Domain randomization for sim-to-real transfer.** To facilitate the sim-to-real transfer, we apply additional domain randomization to the simulation training. We record the range of observation noises in hardware without any robot motion and then apply the same amount of noise to state observations in simulation. We find the state estimation in hardware particularly sensitive to the object heights. Also, we apply random noise (zero mean with 0.03 standard deviation) to the sampled action from `DPPO` to simulate the imperfect low-level controller; we find adding such noise to the Gaussian policy leads to zero task success rate while `DPPO` is robust to it (also see discussion in Section 6).

**BC regularization loss used for Gaussian baseline.** Since the fine-tuned Gaussian policy exhibits very jittery behavior and leads to zero success rate in real evaluation, we further experiment with adding a behavior cloning (BC) regularization loss in fine-tuning with the Gaussian baseline. The

combined loss follows

$$\mathcal{L}_{\theta,+\text{BC}} = \mathcal{L}_\theta - \alpha_{\text{BC}} \mathbb{E}^{\pi_{\theta_{\text{old}}}} \Big[ \sum_{k=0}^{K-1} \log \pi_{\theta_{\text{pre-trained}}}(a_t^k | a_t^{k+1}, s_t) \Big],$$

where $\pi_{\theta_{\text{pre-trained}}}$ is the frozen BC-only policy. The extra term encourages the newly sampled actions from the fine-tuned policy to remain high-likelihood under the BC-only policy. We set $\alpha_{\text{BC}} = 0.1$. However, although this regularization reduces the sim-to-real gap, it also significantly limits fine-tuning, leading to the fine-tuning policy saturating at 53% success rate shown in Fig. 6.

### F.9 ADDITIONAL DETAILS OF AVOID TASK FROM D3IL AND TRAINING IN SECTION 6

**Pre-training.** We split the original dataset from D3IL based on the three settings, M1, M2, and M3; in each setting, observations and actions are normalized to $[0, 1]$ using min/max statistics. All policies are trained with batch size 16 (due to the small dataset size), learning rate 1e-4 decayed to 1e-5 with a cosine schedule, and weight decay 1e-6. Diffusion-based policies are trained with about 15000 epochs, while Gaussian and GMM policies are trained with about 10000 epochs; we manually examine the trajectories from different pre-trained checkpoints and pick ones that visually match the expert data the best.

**Fine-tuning.** Diffusion-based, Gaussian, and GMM pre-trained policies are then fine-tuned using online experiences sampled from 50 parallelized MuJoCo environments (Todorov et al., 2012). Reward curves shown in Fig. 8 and Fig. 19 are evaluated by running fine-tuned policies with the same amount of exploration noise used in training for 50 episodes; we choose to use the training (instead of evaluation) setup since Gaussian policies exhibit multi-modality only with training noise. Episodes terminate only when they reach 100 steps.

**Added action noise during fine-tuning.** In Fig. 8 left, we demonstrate that **DPPO** exhibits stronger training stability when noise is added to the sampled actions during fine-tuning. The noise starts at the 5th iteration. It is sampled from a uniform distribution with the lower limit ramping up to 0.1 and the upper limit ramping up to 0.2 linearly in 5 iterations. The limits are kept the same from the 10th iteration to the end of fine-tuning.

### F.10 LISTED TRAINING HYPERPARAMETERS

| Method | Parameter | Task(s) | | | |
|---|---|---|---|---|---|
| | | GYM | Lift, Can | Square | Transport |
| Common | $\gamma_{\text{ENV}}$ | 0.99 | 0.999 | 0.999 | 0.999 |
| | $\sigma_{\min}^{\exp}$ | 0.1 | 0.1 | 0.1 | 0.08 |
| | $\sigma_{\min}^{\text{prob}}$ | | | 0.1 | |
| | $T_p$ | 4 | 4 | 4 | 8 |
| | $T_a$ | 4 | 4 | 4 | 8 |
| | $K$ | | | 20 | |
| | Actor learning rate | | 1e-4 for **DPPO** and 1e-5 for others (tuned from 1e-4 to 1e-5) | | |
| | Critic learning rate (if applies) | | | 1e-3 | |
| | Actor MLP dims | [512, 512, 512] | [512, 512, 512] | [1024, 1024, 1024] | [1024, 1024, 1024] |
| | Critic MLP dims (if applies) | | | [256, 256, 256] | |
| **DRWR** | $\beta$ | | | 10 | |
| | $w_{\max}$ | | | 100 | |
| | $N_\theta$ | | | 16 | |
| | Batch size | | | 1000 | |
| **DAWR** | $\beta$ | | | 10 | |
| | $w_{\max}$ | | | 100 | |
| | $\lambda_{\text{DAWR}}$ | | | 0.95 | |
| | $N_\theta$ | | | 64 | |
| | $N_\phi$ | | | 16 | |
| | Buffer size | 200000 | 120000 | 120000 | 120000 |
| | Batch size | | | 1000 | |
| DIPO | $\alpha_{\text{DIPO}}$ | | | 1e-4 | |
| | $M_{\text{DIPO}}$ | | | 10 | |
| | $N_\theta$ | | | 64 | |
| | Buffer size | | | 1000000 | |
| | Batch size | | | 1000 | |
| **IDQL** | $M_{\text{IDQL}}$ | 20 | 10 | 10 | 10 |
| | $N_\theta$ | | | 128 | |
| | $N_\phi$ | | | 128 | |
| | Buffer size | 1000000 | 250000 | 250000 | 250000 |
| | Batch size | | | 1000 | |
| **DQL** | $\alpha_{\text{DQL}}$ | | | 1 | |
| | $N_\theta$ | | | 16 | |
| | $N_\phi$ | | | 16 | |
| | Buffer eize | | | 1000000 | |
| | Batch size | | | 1000 | |
| **QSM** | $\alpha_{\text{QSM}}$ | | | 10 | |
| | $N_\theta$ | | | 16 | |
| | $N_\phi$ | | | 16 | |
| | Buffer size | 1000000 | 250000 | 250000 | 250000 |
| | Batch size | | | 1000 | |
| **DPPO** | $\gamma_{\text{DENOISE}}$ | | | 0.99 | |
| | GAE $\lambda$ | | | 0.95 | |
| | $N_\theta$ | 5 | 10 | 10 | 10 |
| | $N_\phi$ | 5 | 10 | 10 | 10 |
| | $\varepsilon$ | | | 0.01 | |
| | Batch size | 50000 | 7500 | 10000 | 10000 |
| | $K'$ | | | 10 | |

Table 7: Fine-tuning hyperparameters for OpenAI GYM and ROBOMIMIC tasks when **comparing diffusion-based RL methods**. We list hyperparameters shared by all methods first, and then method-specific ones.

| Method | Parameter | Task(s) | | | |
| --- | --- | --- | --- | --- | --- |
| | | `HalfCheetah-v2` | `Kitchen-Complete-v0` | `Kitchen-Partial-v0` | `Kitchen-Mixed-v0` |
| Common | $\gamma_{\text{ENV}}$ | | | 0.99 | |
| **RLPD** | $T_p$ | | | 1 | |
| | $T_a$ | | | 1 | |
| | $N_\phi$ | 20 | 10 | 10 | 10 |
| | $N_{\text{critic}}$ | 10 | 5 | 5 | 5 |
| | Batch size | | | 256 | |
| **Cal-QL** | $T_p$ | | | 1 | |
| | $T_a$ | | | 1 | |
| | $\beta_{\text{cql}}$ | | | 5 | |
| | Batch size | | | 256 | |
| **IBRL** | $T_p$ | | | 1 | |
| | $T_a$ | | | 1 | |
| | $N_\phi$ | | | 5 | |
| | $N_{\text{critic}}$ | | | 5 | |
| | Batch size | | | 256 | |
| **DPPO** | $T_p$ | 1 | 4 | 4 | 4 |
| | $T_a$ | 1 | 4 | 4 | 4 |
| | $\sigma_{\min}^{\exp}$ | | | 0.1 | |
| | $\sigma_{\min}^{\text{prob}}$ | | | 0.1 | |
| | $\gamma_{\text{DENOISE}}$ | | | 0.99 | |
| | GAE $\lambda$ | | | 0.95 | |
| | $N_\theta$ | 5 | 10 | 10 | 10 |
| | $N_\phi$ | 5 | 10 | 10 | 10 |
| | $\varepsilon$ | | | 0.01 | |
| | Batch size | 10000 | 5600 | 5600 | 5600 |
| | $K$ | | | 20 | |
| | $K'$ | | | 10 | |

| Method | Parameter | `Can, PH` | `Square, PH` | `Can, MH` | `Square, MH` |
| --- | --- | --- | --- | --- | --- |
| Common | $\gamma_{\text{ENV}}$ | | | 0.999 | |
| | $T_a$ | | | 1 | |
| | $T_a$ | | | 1 | |
| **RLPD** | $N_\phi$ | | | 3 | |
| | $N_{\text{critic}}$ | | | 5 | |
| | Batch size | | | 256 | |
| **Cal-QL** | $\beta_{\text{cql}}$ | | | 5 | |
| | Batch size | | | 256 | |
| **IBRL** | $N_\phi$ | | | 3 | |
| | $N_{\text{critic}}$ | | | 5 | |
| | Batch size | | | 256 | |
| **DPPO** | $\sigma_{\min}^{\exp}$ | | | 0.1 | |
| | $\sigma_{\min}^{\text{prob}}$ | | | 0.1 | |
| | $\gamma_{\text{DENOISE}}$ | 0.9 | 0.9 | 0.99 | 0.99 |
| | GAE $\lambda$ | | | 0.95 | |
| | $N_\theta$ | | | 10 | |
| | $N_\phi$ | | | 10 | |
| | $\varepsilon$ | | | 0.01 | |
| | Batch size | 6000 | 15000 | 8000 | 20000 |
| | $K$ | | | 20 | |
| | $K'$ | | | 10 | |

Table 8: Fine-tuning hyperparameters for `HalfCheetah-v2`, FRANKA-KITCHEN, `Can`, and `Square` (PH or MH datasets) when **comparing demo-augmented RL methods**. We list hyperparameters shared by all methods first, and then method-specific ones.

| Method | Parameter | Lift, Can | Square | Transport |
|---|---|---|---|---|
| | | | Task | |
| Common | $\gamma_{\text{ENV}}$ | | 0.999 | |
| | $T_a$ | 4 | 4 | 8 |
| | Actor learning rate | 1e-4 | 1e-5 | 1e-5 (decayed to 1e-6) |
| | Critic learning rate | | 1e-3 | |
| | GAE $\lambda$ | | 0.95 | |
| | $N_\theta$ | 10 | 10 | 8 |
| | $N_\phi$ | 10 | 10 | 8 |
| | $\varepsilon$ | | 0.01 (annealed in **DPPO**) | |
| Gaussian, Common | $\sigma_{\text{Gau}}$ | 0.1 | 0.1 | 0.08 |
| | Batch size | 7500 | 10000 | 10000 |
| Gaussian-MLP | Model size | 552K | 2.15M | 1.93M |
| Gaussian-Transformer | Model size | 675K | 1.86M | 1.87M |
| GMM, Common | $M_{\text{GMM}}$ | | 5 | |
| | $\sigma_{\text{GMM}}$ | 0.1 | 0.1 | 0.08 |
| | Batch size | 7500 | 10000 | 10000 |
| GMM-MLP | Model size | 1.15M | 4.40M | 4.90M |
| GMM-Transformer | Model size | 680K | 1.87M | 1.89M |
| **DPPO**, Common | $\gamma_{\text{DENOISE}}$ | | 0.99 | |
| | $\sigma_{\min}^{\exp}$ | 0.1 | 0.1 | 0.08 |
| | $\sigma_{\min}^{\text{prob}}$ | 0.1 | 0.1 | 0.1 |
| | $K$ | | 20 | |
| | $K'$ | | 10 | |
| | Batch size | 75000 | 100000 | 100000 |
| **DPPO**-MLP | Model size | 576K | 2.31M | 2.43M |
| **DPPO**-UNet | Model size | 652K | 1.62M | 1.68M |

Table 9: Fine-tuning hyperparameters for ROBOMIMIC tasks with **state** input when **comparing policy parameterizations**. We list hyperparameters shared by all methods first, and then method-specific ones. Since the different policy parameterizations use different neural network architecture, we list the total model size here instead of the details such as MLP dimensions.

| Method | Parameter | Lift, Can | Square | Transport |
|---|---|---|---|---|
| | | | Task | |
| Common | $\gamma_{\text{ENV}}$ | | 0.999 | |
| | $T_a$ | 4 | 4 | 8 |
| | Actor learning rate | 1e-4 | 1e-5 | 1e-5 (decayed to 1e-6) |
| | Critic learning rate | | 1e-3 | |
| | GAE $\lambda$ | | 0.95 | |
| | $N_\theta$ | 10 | 10 | 8 |
| | $N_\phi$ | 10 | 10 | 8 |
| | $\varepsilon$ | | 0.01 (annealed in **DPPO**) | |
| Gaussian-ViT-MLP | Model size | 1.03M | 1.03M | 1.93M |
| | $\sigma_{\text{Gau}}$ | 0.1 | 0.1 | 0.08 |
| | Batch size | 7500 | 10000 | 10000 |
| **DPPO**-ViT-MLP | Model size | 1.06M | 1.06M | 2.05M |
| | $\gamma_{\text{DENOISE}}$ | | 0.9 | |
| | $\sigma_{\min}^{\exp}$ | 0.1 | 0.1 | 0.08 |
| | $\sigma_{\min}^{\text{prob}}$ | | 0.10 | |
| | $K$ | | 100 | |
| | $K'$ | | 5 (DDIM) | |
| | Batch size | 37500 | 50000 | 50000 |

Table 10: Fine-tuning hyperparameters for ROBOMIMIC tasks with **pixel** input when **comparing policy parameterizations**. We list hyperparameters shared by all methods first, and then method-specific ones. Since the different policy parameterizations use different neural network architecture, we list the total model size here instead of the details such as MLP dimensions.

| Method | Parameter | Task | | |
|---|---|---|---|---|
| | | One-leg | Lamp | Round-table |
| Common | $\gamma_{\text{ENV}}$ | | 0.999 | |
| | $T_a$ | | 8 | |
| | Actor learning rate | | 1e-5 (decayed to 1e-6) | |
| | Critic learning rate | | 1e-3 | |
| | GAE $\lambda$ | | 0.95 | |
| | $N_\theta$ | | 5 | |
| | $N_\phi$ | | 5 | |
| | $\varepsilon$ | | 0.001 | |
| Gaussian-MLP | Model size | 10.64M | 10.62M | 10.62M |
| | $\sigma_{\text{Gau}}$ | | 0.04 | |
| | Batch size | | 8800 | |
| **DPPO**-UNet | Model size | 6.86M | 6.81M | 6.81M |
| | $\gamma_{\text{DENOISE}}$ | | 0.9 | |
| | $\sigma_{\min}^{\exp}$ | | 0.04 | |
| | $\sigma_{\min}^{\text{prob}}$ | | 0.1 | |
| | $K$ | | 100 | |
| | $K'$ | | 5 (DDIM) | |
| | Batch size | | 44000 | |

Table 11: Fine-tuning hyperparameters for FURNITURE-BENCH tasks when **comparing policy parameterizations**. We list hyperparameters shared by all methods first, and then method-specific ones.

