# OpenReview forum: "Diffusion Policy Policy Optimization"
_ICLR.cc/2025/Conference — ICLR 2025 Poster_

### Official Review · Reviewer_sDz9 · 2024-10-30

**Soundness:** 2
**Presentation:** 1
**Contribution:** 2
**Rating:** 6
**Confidence:** 3

**Summary:**

This paper studies how to use PPO to finetune the diffusion-model based policy in an offline-to-online setting. By integrating denoising steps of diffusion model into task MDP, authors could utilize PPO to finetune the diffusion model for improved performance. Extensive efforts are put to find the best practice in fine-tuning diffusion policy, in terms of hyperparameters, network architecture and algorithmic variants. Finally,  the efficacy and the performance of the proposed framework  is demonstrated on various experiments in both simulated and real-world scenarios.

**Strengths:**

This paper contains tons of discoveries related to fine-tuning diffusion policy via PPO. Specifically, numerous aspects like denoising steps to be fine-tuned, network architecture, GAE variants  and effects of expert data are all studied with experimental results reported. As fine-tuning diffusion policy for real-world robot is a very important field, these experiments are welcome and contain significant value  for the community. Last but not least, the authors also provide an empirical explanation for why their method obtains better results than others (i.e. structural exploration).

$~$

## Originality
Utilizing RL method to finetune diffusion model is in great need. And the fruitful experimental results presented in this work  are of great value and originality.

## Quality

Many experiments and ablation studies are conducted and the best practices are summarized from them.

## Clarity

Unfortunately, this paper is not clearly written. In the main papers (1-10 pages), authors’ writing lacks a central logic  and the results are presented without  a focus. I have great trouble understanding the contribution and experimental results during reading.

## Significance
The large amount of experiments conducted is of great value to the community, especially considering there are also real-world experiments.

**Weaknesses:**

Unfortunately, I personally think the presentation of this paper is very poor. With the extensive and fruitful experiments results, the authors fail to present them in a concise and focused manner in the main pages and make me quite confused during reading and comprehending this paper.

Firstly, there are three components that are actively mentioned and studied: the diffusion policy, the PPO method and the best practice for combining them together.

But based on my comprehension, the diffusion policy and PPO used in this work are of no novel changes & contributions.

Combining these 2 together and successfully applying on real-world robotic system, is with great value and potential hidden evils. But  the authors fail to present this to me  in a clear and concise manner: for example, there are  experiments comparing diffusion policy (DP) to gaussian parameterized policy, but the advantage of DP to vanilla parameterization is already well-established and not the contribution of this paper, in my opinion. Since this paper is already over-sized, I suggest the authors to relocate some contents into appendix and focus on your main contribution in the main pages



To summarize, I hope the organization and presentation of this paper could be improved such that the problem formulation, method design and experimental results could be tightly chained and focused on  your main contribution.

**Questions:**

1. Could you change the title "Diffusion Policy Policy Optimization" to provide more contexts to readers? Current title is too short and abstract and your work is not as general as it indicates.
2. I would  suggest deleting/relocating some contents in early chapters as your paper is already over-sized and some of these contents are already known.
4. in line 249, the definition of $\gamma_{\text{ENV}}^t$ seems to be incorrect? I thinks it should be $\gamma_{\text{ENV}}^{t^\prime-t}$? or are there any further treatments of this discount factor?
4. there lacks a pseudo-code for the whole algorithm, please provide one. For example, how is your V function trained?
5. for fine-tuning control policy in RL, most existing algorithms finish in less than 1e6 steps. But the results presented throughout this paper is conducted in 1e6~1e7 steps. Could you truncate the figures to 5e5 such that we could better compare the performance of various methods?

6. Could you provide more comparisons of DPPO against established offline-to-online methods like [1] and [2]?

>[1] Wang, Shenzhi, et al. "Train once, get a family: State-adaptive balances for offline-to-online reinforcement learning." Advances in Neural Information Processing Systems 36 (2024).
>
>[2] Zhang, Haichao, Wei Xu, and Haonan Yu. "Policy Expansion for Bridging Offline-to-Online Reinforcement Learning." The Eleventh International Conference on Learning Representations.

7. Could you provide more explanations about the worse performance of DAWR, DRWR and Cal-QL? In appendix, the result of the performance of Cal-QL on  Can and Square are almost zero, which is surprising to me. Could you elaborate more on this? BTW, what is the performance of Cal-QL when only offline training is performed?

8. for DRWR, in the weight function, could you minus a V function from the estimated return such that its’ more aligned toward DPPO?
9. in line 988-994, you mentioned speedup and less GPU memory usage is obtained by fine-tuning only the last few K denoising steps. But the method is to  use a trainable $\theta$ for the last K steps and a frozen $\theta_{\text{FT}}$ for previous steps. How does this method improve the training speed and reduce GPU memory?

10. Sect. 6 of structural exploration is of much importance and novelty. Could you try ablating on this phenomenon for DP? for example, by tuning the denoising process such that DP itself performs less or more structural exploration and see if the performance is affected?

There are many interesting, fruitful discoveries and experiments in this work. But I find none of them is studied exhaustively or presented clearly.

---

> ### Author Response · Authors · 2024-11-20
> **Response from the authors**
>
> We thank the reviewer for the detailed feedback. We first address the major concerns raised by the reviewer about the paper presentation. Please also see the revised version of the paper. Since we have added or removed some figures or sections in the revision, we will use the updated numbering below unless noted.
>
> ### **Presentation**
> We appreciate the reviewers for helpful suggestions for improving the presentation. In order to make space for additional exposition, we have moved “potential impacts” and the figure displaying a physical hardware trajectory (original Figure 2). We have made the following additions in its place:
> - We acknowledge that the approach section can better motivate why considering the two-level MDP is necessary. At the beginning of Sec. 4, we have expanded on the discussion about naively running policy gradient with diffusion failing to optimize diffusion policy, which is shown in the experiments using exact likelihood in the appendix.
> - We acknowledge that the current manuscript lacks a clear rationale for considering each set of experiments in Sec. 5. We have included such rationales at the beginning of Sec. 5 in the revision so that the motivation of the experiments is made clear. In particular, we will motivate comparison to Gaussian policies, which we elaborate on below.
> - We acknowledge that the implementation details of the DPPO algorithm are missing, and we have added a detailed pseudocode and discussions on the GAE-based advantage estimation in Appendix C.
>
> We welcome any other suggestions from the reviewer on improving the paper presentation further.
>
> ### **Other concerns**
> > Why compare to Gaussian policies: ...***the advantage of DP to vanilla parameterization is already well-established*** and not the contribution of this paper, in my opinion…
>
> We agree that the benefits of Gaussian policies over Diffusion policies are well-documented for BC pre-training [1], and that it is widely appreciated that Diffusion can represent richer, multi-modal distributions over action.
>
> Our comparison to Gaussian policies is to determine (1) whether these benefits translate to RL, but more important (2) are there any synergistic effects when conducting online RL through a Diffusion parameterization, which is not studied in any prior work. We have included this rationale/framing in the revision.
>
> All comparisons to Gaussian policies control for the already-known differences between policy parameterization in pre-training by ensuring that their success rates on BC pre-training is comparable (this is not exact, but we have carefully chosen pre-training checkpoints to control this). Therefore, any improvements in performance are due to the unique advantages of **diffusion as a policy parametrization for online RL fine-tuning**, rather than its ability to fit multi-modal demonstration data. At least for us authors, these surplus improvements of diffusion over Gaussian in RL fine-tuning were initially very surprising.
>
> The improvement of diffusion over Gaussian in benchmarking results also led us to explore both parameterizations in investigative experiments (Sec. 6), and we find that the advantages conferred by DPPO over other parametrizations support the hypothesis that diffusion and RL have synergistic, rather than merely additive, benefits. For example, DPPO (using our designed noise-exploration schedule) **induces a fundamentally different form of exploration** than previously dominant Gaussian parameterization, exploring in **structured and on-manifold** manners near the expert data. This is irrelevant during BC and is a crucial capability of diffusion in RL settings. Finally, we demonstrate the gaps between DPPO and Gaussian in sim-to-real transfer. This too is a finding that is not implied by any past work.
>
> **To summarize: we demonstrate the advantages of DPPO over other policy parameterizations for RL fine-tuning settings, and these advantages imply benefits of the DP parameterization beyond what has already been observed in BC, which had not been studied in any prior work.** We have emphasized this point in the revision.

---

> > ### Author Response · Authors · 2024-11-20
> > **Continued**
> >
> > > Firstly, there are three components that are actively mentioned and studied: the diffusion policy, the PPO method and the best practice for combining them together. But based on my comprehension, the diffusion policy and PPO used in this work are of ***no novel changes & contributions.***
> >
> > We would like to point out another major component of the DPPO framework, which is the two-layer MDP that treats the denoising process as a MDP and embeds it into the environment MDP. This is the novel design proposed by DPPO that allows applying policy gradient methods with the diffusion parameterization.
> >
> > Unrolling the denoising process in diffusion policies had not been considered in prior work. This is because unrolling the MDP increases the number of effective MDP steps by a factor of K, and we know that long-horizon RL problems are frequently more challenging. This led to other methods exploiting techniques either from offline RL [2] or ones inspired by diffusion guidance methods [3]. We also note that the authors of QSM tried a similar approach to ours but based on REINFORCE, and found it to have low performance. We ourselves thought our approach might not work, and indeed, its performance was rather unremarkable for the first few months of experimentation. Thus, the fact that such a method indeed is the most performant comes as a major surprise to us as authors, and we hope, to the community.
> >
> > > Could you ***change the title*** "Diffusion Policy Policy Optimization" to provide more contexts to readers? Current title is too short and abstract and your work is not as general as it indicates.
> >
> > We have changed the title to Training Diffusion Policy with Policy Gradient in the revision. We welcome any other suggestions from the reviewer.
> >
> > > In line 249, the definition of $\gamma^t_\text{ENV}$ seems to be incorrect? I thinks it should be $\gamma^{t'-t}_\text{ENV}$? or are there any further treatments of this ***discount factor***?
> >
> > This is indeed a typo, and it should be $\gamma^{t'-t}_\text{ENV}$. We thank the reviewer for pointing it out, and we have corrected it in the revision.
> >
> > > There lacks a ***pseudo-code*** for the whole algorithm, please provide one. For example, how is your V function trained?
> >
> > We thank the reviewer for the great suggestion. We have added the pseudocode (https://imgur.com/a/JfOixCy) and shown the loss functions used in DPPO in Appendix C. The V function is trained to predict the discounted sum of future rewards, $\sum_{l=0}^{T-t} \gamma^l_\text{ENV} \bar R_{\bar t(t+l, k)}$.
> >
> > > For fine-tuning control policy in RL, most existing algorithms finish in less than 1e6 steps. But the results presented throughout this paper is conducted in 1e6~1e7 steps. Could you ***truncate the figures*** to 5e5 such that we could better compare the performance of various methods?
> >
> > We have truncated Figure 4 (original Figure 5) to smaller ranges — however, we note that our experiment settings differ from others that we consider (1) action chunking and (2) the more challenging Robomimic tasks, which we find taking more samples. In Fig. 11 we compare DPPO with other demo-augmented methods, including RLPD, Cal-QL, and IBRL, and for these comparisons, we use the original experiment setup of these methods (no action chunking), and we have focused on showing the performance up to 1e6 steps.
> >
> > > Could you provide ***more comparisons*** of DPPO against established offline-to-online methods like [1] and [2]?
> >
> > We thank the reviewer for pointing out additional methods for offline-to-online RL. Indeed these methods are relevant to DPPO that the first one addresses the potential distributional shift between pre-training data and fine-tuning environments, and the second one proposes retaining the pre-trained policy during fine-tuning and expanding the policy set with another online one (akin to IBRL considered in our paper). We have cited both and discussed them in the extended related work in the appendix, and we will do our utmost to implement these methods in our manipulation setups before rebuttal ends.

---

> ### Author Response · Authors · 2024-11-20
> **Continued**
>
> > Could you provide more explanations about the ***worse performance of DAWR, DRWR and Cal-QL***? In appendix, the result of the performance of Cal-QL on Can and Square are almost zero, which is surprising to me. Could you elaborate more on this? BTW, what is the performance of Cal-QL when only offline training is performed?
>
> RWR and AWR are known to optimize the reward objective subject to a KL constraint on the policy [4], thus the optimization can be constrained. Furthermore, when using diffusion parameterization, the supervised learning objective in RWR and AWR (score matching) does not optimize the exact likelihood of the data; instead, it is a variational lower bound. Black et al. [5] also observe the suboptimal performance of RWR with image diffusion models.
>
> We have found Cal-QL shows competitive performance in newly added Franka Kitchen tasks (Fig. 11, https://imgur.com/a/PhSV6Gt). However, we find it unable to learn the precise motion needed in Robomimic tasks. In Can and Square, the policy needs to precisely grasp and place the objects while in Franka Kitchen, the requirement is not as high —- the robot can open the microwave, move the kettle, turn on the switch, and open the cabinet (the four subtasks) without precise control of the gripper opening. Upon observing the recorded videos, we find Cal-QL policies can locate the gripper reasonably near the object of interest in Can and Square, but fail to grasp them precisely or move them to locations precisely. In contrast, we find IBRL can perform such behavior better. We hypothesize that the offline RL objective causes the issue, as the learned Q function might not be accurate enough for updating the actor to learn such precise motion. IBRL instead uses a behavior cloning objective for pre-training.
>
> The initial success rate on the figures (x=0) indicates Cal-QL's performance when only offline training is performed, which is zero in Can and Square. Again we believe the offline RL objective failed to learn the precise motion required for these tasks.
>
> If there is another specific setting where the reviewer would like to understand more about the performance of DAWR, DRWR, and Cal-QL, please let us know. Similarly, we find that RLPD does not perform well in Can and Square, which is consistent with the findings in the IBRL paper.
>
> > For DRWR, in the weight function, could you ***minus a V function*** from the estimated return such that its’ more aligned toward DPPO?
>
> Yes, we could have used a value function baseline in the return estimation in DRWR, but we chose not to do so, so it matches the original RWR setup [6]. Instead, we also consider DAWR, which applies TD(\lambda) in the advantage estimation and is likely to exhibit lower variance in return estimation than DRWR.
>
> > In line 988-994, you mentioned ***speedup and less GPU memory usage*** is obtained by fine-tuning only the last few K denoising steps. But the method is to use a trainable  for the last K steps and a frozen FT for previous steps. How does this method improve the training speed and reduce GPU memory?
>
> The reviewer raised a good point that fine-tuning all denoising steps circumvents keeping a copy of the original network. However, we note that the network size itself is fairly small in terms of GPU memory. Reducing the number of fine-tuned denoising steps means we only need to save a subset of the intermediate denoised actions and their likelihood, which we find can take significant memory in an epoch involving 50-100 trajectories. In Fig. 16 we also show a moderate speedup in both number of samples and wall-clock time by not fine-tuning all denoising steps.
>
> For example, we find that in the HalfCheetah task it takes only 60Mb to store the extra model, but fine-tuning 10 denoising steps instead of 20 saves about 1.1Gb used for storing the denoised actions and likelihood. We have added this clarification in the appendix.

---

> ### Author Response · Authors · 2024-11-20
> **Continued**
>
> > Sect. 6 of structural exploration is of much importance and novelty. Could you try ***ablating on this phenomenon for DP***? for example, by tuning the denoising process such that DP itself performs less or more structural exploration and see if the performance is affected?
>
> We thank the reviewer for appreciating the analysis in Sec. 6, which we consider one of the most important findings and contributions of our work. We agree that further investigation of structured exploration can strengthen the paper, and we have added an ablation study on it in Appendix D.7 (also see https://imgur.com/a/2TUw5yw).
>
> We find that, when we use DDIM (thus we can sample deterministically) and then add noise only to the final denoised action (similar to additive noise to Gaussian policy), the sampled trajectories become less structured. In contrast, DPPO adds noise to all denoising steps (with either DDIM or DDPM) and we find the trajectories maintain strong structured exploration near the expert data. When we run both setups in Can and Square tasks, we find DPPO adding noise to all denoising steps shows faster convergence. This result, on top of existing ones in Sec. 5 showing DPPO outperforming Gaussian and GMM policies, demonstrates the effect of structured exploration in faster fine-tuning.
>
> We also welcome any other suggestions from the reviewer on new studies on structured exploration. We believe this is a very important future research direction as our work uncovers that diffusion induces a fundamentally different form of exploration compared to previously dominant Gaussian parameterization.
>
> [1] Diffusion policy: Visuomotor policy learning via action diffusion. Chi et al.
>
> [2] Diffusion Policies as an Expressive Policy Class for Offline Reinforcement Learning. Wang et al.
>
> [3] Learning a Diffusion Model Policy from Rewards via Q-Score Matching. Psenka et al.
>
> [4] AWAC: Accelerating online reinforcement learning with offline datasets. Nair et al.
>
> [5] Training Diffusion Models with Reinforcement Learning. Black et al.
>
> [6] Reinforcement Learning by Reward-weighted Regression For Operational Space Control. Peters et al.

---

> ### Author Response · Authors · 2024-11-24
> **Follow-up from the authors**
>
> As the rebuttal period is closing, the authors would greatly appreciate it if the reviewer can address our responses to the original review. We believe we have adequately addressed the issues raised especially ones around the novelty (also see the meta response) and the overall presentation. If the reviewer requires further clarifications and revisions to the paper, or disagrees with some of our responses, please let us know.

---

### Official Review · Reviewer_WTdz · 2024-10-31

**Soundness:** 3
**Presentation:** 3
**Contribution:** 3
**Rating:** 6
**Confidence:** 4

**Summary:**

The papers introduces Diffusion Policy Policy Optimization (DPPO), an algorithmic framework designed to fine-tune diffusion-based policies using policy gradient optimization (PPO) from reinforcement learning (RL). DPPO enables structured exploration, enhances training stability, and demonstrates the robustness and generalization during deployment. It improves policy performance across various benchmarks, including tasks with pixel-based observations and long-horizon rollouts, which have traditionally been challenging for previous RL methods.

**Strengths:**

(1) The paper introduces a novel dual-layer MDP framework and the use of PPO to fine-tune Diffusion-Policy, enhancing its performance in various tasks.
(2) The paper proposes several best practices for DPPO, such as fine-tuning only the last few denoising steps and using DDIM sampling to replace some denoising steps, improving efficiency and performance.
(3) DPPO shows a strong performance in all benchmark tests, particularly in tasks involving pixel-based observations and long-horizon rollouts.
(4)  DPPO achieves highly efficient zero-shot transfer from simulation to real-world tasks in robotics, demonstrating a minimal sim-to-real performance gap.

**Weaknesses:**

(1) Figure 10 shows that the pre-trained Diffusion Policy can model multimodality in the behavior policy. However, after fine-tuning, the Diffusion Policy gradually converges into a deterministic policy. In this case, does it lose the advantage of expressing multimodal action distributions that Diffusion Policy is known for?

(2) In the experiments, DPPO is only compared against other diffusion-based methods, without any comparison to existing state-of-the-art Offline2Online or Sim2Real algorithms, such as Uni-O4 [1] and O3F [2]. Clearly, there is a significant optimal gap in environments like HalfCheetah.

(3) The motivation for fine-tuning models is typically to reduce the number of online interactions and improve sample efficiency. However, DPPO does not seem to significantly enhance sample efficiency, as its convergence speed is similar to other online reinforcement learning methods.

(4) The v-D4RL [3] dataset could have been considered in the experiments for broader evaluation.

(5) Compared to QSM [4], DPPO does not introduce particularly novel concepts but rather provides engineering or empirical adjustments, such as fine-tuning certain denoising steps and replacing value estimation with advantage estimation.

[1] Kun Lei, et al. Uni-O4: Unifying Online and Offline Deep Reinforcement Learning with Multi-Step On-Policy Optimization
[2] Seunghyun Lee, et al. Offline-to-Online Reinforcement Learning via Balanced Replay and Pessimistic Q-Ensemble
[3] Cong Lu, et al. Challenges and Opportunities in Offline Reinforcement Learning from Visual Observations
[4] Michael Psenka, et al. Learning a Diffusion Model Policy from Rewards via Q-Score Matching

**Questions:**

Please see the weaknesses

---

> ### Author Response · Authors · 2024-11-20
> **Response from the authors**
>
> We thank the reviewer for the feedback and the appreciation of the novelty of our work. We address the concerns raised by the reviewer below. Please also see the revised version of the paper. Since we have added or removed some figures or sections in the revision, we will use the updated numbering below unless noted.
>
> > (5) Compared to QSM [4], DPPO ***does not introduce particularly novel concepts*** but rather provides engineering or empirical adjustments, such as fine-tuning certain denoising steps and replacing value estimation with advantage estimation.
>
> We would like to highlight that our derivation is distinct from the policy gradient derivation presented in QSM. For clarity, we will call this QSM-PG to distinguish from the main method proposed in QSM. DPPO formulation is based on treating the multi-step denoising process as a MDP, whereas QSM-PG does not. The gradient update in QSM-PG (Eq. 9 in [1]) differs from ours in that it sums over the denoising steps and then takes expectation over environment steps on the outside, whereas our update (Eq. 4.2 in ours) takes expectation over both denoising steps and environment steps as we embed the diffusion MDP into the environment MDP. In particular, the QSM-PG update does not treat the denoising process as a MDP. Hence, the updates have a subtly different form than the two-layer MDP in our setup. We have made this distinction clear in the revision.
>
> We also note that our more significant contribution is the **scientific finding** that this relatively simple method, based on policy optimization, significantly outperforms other proposed methods of which the authors are aware in challenging robotic tasks. In Sec. 6 we also show **diffusion induces a fundamentally different form of exploration** from previously dominant Gaussian parameterization --- DPPO exhibits structured and on-manifold exploration, which is a very novel finding of ours and has not been studied in any previous work. Such exploration leads to faster training and also natural behavior after fine-tuning, which is critical to DPPO's successful sim2real transfer.
>
> We have also added ablation study on such exploration in Appendix D.7 (also see https://imgur.com/a/2TUw5yw) in the revision. We find that, when we use DDIM (thus we can sample deterministically) and then add noise only to the final denoised action (similar to additive noise to Gaussian policy), the sampled trajectories become less structured. In contrast, DPPO adds noise to all denoising steps (with either DDIM or DDPM) and we find the trajectories maintain strong structured exploration near the expert data. When we run both setups in Can and Square tasks, we find DPPO adding noise to all denoising steps shows faster convergence. This result, on top of existing ones in Sec. 5 showing DPPO outperforming Gaussian and GMM policies, demonstrates the effect of structured exploration in faster fine-tuning.
>
> > (1) Figure 10 shows that the pre-trained Diffusion Policy can model multimodality in the behavior policy. However, after fine-tuning, the Diffusion Policy gradually converges into a deterministic policy. In this case, does it ***lose the advantage of expressing multimodal action*** distributions that Diffusion Policy is known for?
>
> In the experiments from Figure 10 (now Figure 9), we specifically design the reward function such that the robot needs to reach the goal line **from the topmost mode**. It is natural and desired that the policy will collapse the suboptimal modes from pre-training.
>
> In general, we agree that RL training, without specific treatment of the data modality, can reduce the trajectory entropy. However, we believe while the ability to capture multi-modality is necessary for behavior cloning as the expert data can be multi-modal (the different modes often stem from suboptimality of human demonstrations), it is not necessary to maintain multi-modality for the final policy. As we have shown, DPPO maintains a sufficient amount of multi-modality for exploration during fine-tuning, but is not prevented from collapsing suboptimal modes when desirable for final-policy performance.

---

> ### Author Response · Authors · 2024-11-20
> **Continued**
>
> > (2) In the experiments, DPPO is only compared against other diffusion-based methods, without any comparison to existing state-of-the-art ***Offline2Online or Sim2Real algorithms***, such as Uni-O4 [1] and O3F [2]. Clearly, there is a significant optimal gap in environments like HalfCheetah.
>
> We note that we have compared DPPO with a range of offline2online algorithms, including IDQL (diffusion offline RL pre-training + online fine-tuning) in Sec. 5.1 and Cal-QL (Gaussian offline RL pre-training + online fine-tuning) and IBRL (Gaussian behavior cloning + online fine-tuning) in Sec. 5.2. Among them, IBRL has been shown very effective in training manipulation policies for challenging Robomimic tasks, but still fares worse compared to DPPO (Fig. 11). Cal-QL, which is representative of many offline RL pre-training + online fine-tuning methods and addresses the distributional mismatch between offline data and online policy, struggles to learn precise motion needed in Robomimic and achieves near zero success rate.
>
> In addition, as the focus of our paper is not on sim2real, we do not compare specifically to sim-to-real algorithms that usually include domain-specific designs. DPPO is a general framework allowing RL fine-tuning with diffusion policies that can be combined with sim2real approaches, and we use sim2real as just one test of its effectiveness. In the revision we have also cited Uni-O4 and O3F and discussed them in the extended related work in the appendix, and we will try to compare DPPO with Uni-O4 at least during rebuttal.
>
> The optimality gap in HalfCheetah between on-policy policy gradient methods and off-policy Q-learning methods is known in the literature [2]. We do not anticipate DPPO building on policy gradients to outperform Q-learning methods in such dense-reward tasks.
>
> > (3) The motivation for fine-tuning models is typically to reduce the number of online interactions and improve sample efficiency. However, DPPO ***does not seem to significantly enhance sample efficiency***, as its convergence speed is similar to other online reinforcement learning methods.
>
> We agree that ultimately, one desires a fine-tuning method that is both efficient and performant. However, the first step towards this research goal is a method that can reliably complete more challenging tasks. The purpose of policy optimization is to show that these tasks **can be** solved by RL fine-tuning given enough interaction time. We note that our baselines fail to solve the dexterous and challenging tasks, regardless of wall-clock time allotted.
>
> Furthermore, the main appeal of DPPO is the great training stability much desired for fine-tuning challenging long-horizon robot manipulation tasks. Compared to some of the Q-learning alternatives, DPPO, building on policy gradients, trades off some sample efficiency for greater training stability. Our results have convincingly demonstrated that Q-learning methods can also lead to performance collapse in these tasks due to the biased update from the possibly inaccurate Q function (Fig. 5 and 11).
>
> > (4) The ***v-D4RL*** dataset could have been considered in the experiments for broader evaluation.
>
> We thank the reviewer for suggesting this great dataset, and we have cited it in the appendix, where the D4RL tasks are discussed in the revision. It includes variations in visual distractions that we have not tested in Robomimic tasks. We will perform additional experiments if time permits before the rebuttal ends — we have been prioritizing our efforts in additional pixel-based experiments with Robomimic and ablation studies on noise scheduling.
>
> [1] Learning a Diffusion Model Policy from Rewards via Q-Score Matching. Psenka et al.
>
> [2] Benchmarks for Spinning Up Implementations. https://spinningup.openai.com/en/latest/spinningup/bench.html.

---

> > ### Comment · Reviewer_WTdz · 2024-11-25
> >
> > I appreciate the authors' detailed response to my comments. I looked through the revised paper.  However, I still have a few remaining concerns.
> >
> > 1. The author stress that diffusion induces a fundamentally different form of exploration. The similar idea also be stressed in CPQL[1], such as "our consistency policy is inherently stochastic, and we find in experiments that this inherent randomness allows for sufficient exploration of the environment without the need for additional exploration strategies" .
> >
> > 2. To support the contribution "DPPO exhibits a remarkably small sim-to-real gap"  and "DPPO's successful sim2real transfer",  I think the comparison with some Sim2Real algorithms should be given and analyzed.
> >
> > [1] Boosting Continuous Control with Consistency Policy, AAMAS, 2024.

---

> ### Author Response · Authors · 2024-11-24
> **Follow-up from the authors**
>
> As the rebuttal period is closing, the authors would greatly appreciate it if the reviewer can address our responses to the original review. We believe we have adequately addressed the issues raised especially the one around the novelty (also see the meta response). If the reviewer requires further clarifications or disagrees with some of our responses, please let us know.

---

> ### Author Response · Authors · 2024-11-25
> **Response to the reviewer response**
>
> We thank the reviewer for responding and sharing the additional concerns. We would like to address them below:
>
> > The author stress that diffusion induces a fundamentally different form of exploration. The similar idea also be stressed in CPQL[1], such as "our consistency policy is inherently stochastic, and we find in experiments that this inherent randomness allows for sufficient exploration of the environment without the need for additional exploration strategies" .
>
> We thank the reviewer for pointing to CPQL, which contains the discussion around exploration from consistency policy that we were not aware of. We have now cited CPQL in the extended related work in Appendix A.2 and included a discussion.
>
> We totally acknowledge that consistency policy can leverage inherent stochasticity for exploration, and there is potential benefit of using such inherent stochasticity, as opposed to additive ones used in Gaussian policy, for better exploration.
>
> Our work presents **affirmative evidence** that DPPO engages in structured and on-manifold exploration as shown in the exploration trajectories plotted in Figure 7 (https://imgur.com/a/0YTC98r) and Figure 22 (https://imgur.com/a/2TUw5yw). The ablation study in Figure 22 also demonstrates that adding noise through the multi-step denoising process is critical to faster fine-tuning as opposed to only adding noise to the last step.
>
> To our knowledge, no prior work including CPQL has presented **careful investigative experiments or ablation studies, including the visualizations, to study the strong inductive bias of diffusion** (or other form of policy stochasticity) **for exploration**. In another word, we don’t think that CPQL argues for “a fundamental different form of exploration” (we are not saying that consistency policy does not; it might have such properties, but it is also possible that it behaves similarly to Gaussian with noise that also suffices for exploration but not as structured as diffusion. CPQL paper itself did not make claims around this from our understanding). Our work presents this claim and also supports it with the extensive experiments and visualizations.
>
> Further investigating the exploration strategies induced by consistency policy and examining if there is a key difference of it from diffusion, would be a very interesting future direction.
>
> > To support the contribution "DPPO exhibits a remarkably small sim-to-real gap" and "DPPO's successful sim2real transfer", I think the comparison with some Sim2Real algorithms should be given and analyzed.
>
> We agree with the reviewer that comparisons to other sim2real algorithms potentially better demonstrate the sim2real capabilities of DPPO. We plan to include comparisons to (1) dedicated domain randomization technique [1] and (2) sim2real adaptation technique [2] in future revisions.
>
> Nonetheless, we would like to stress that, we do not claim DPPO being the best sim2real algorithm and necessarily perform better than existing sim2real algorithms. **We do not present DPPO being a sim2real algorithm as a contribution of our work**, and we hope we have not overclaimed in the paper — to make sure this is the case, we have changed the wording of “...exhibits a remarkably small sim-to-real gap” to “...exhibits smoother behavior than the baseline and transfers better to the real-world” in Contribution 4 in the introduction.
>
> Instead, our intention is to **present, for a simple sim2real algorithm (BC pre-training + RL fine-tuning without regularization), diffusion is a better policy parameterization than Gaussian, and DPPO leverages this**. We demonstrate that DPPO policy can transfer to the real-world without extensive treatment addressing the sim2real gap, and compared to the Gaussian policy, which is the almost uniformly chosen policy parameterization for sim2real RL, DPPO exhibits smoother behavior thanks to on-manifold exploration (thus a smaller sim2real gap) and transfers better to the real-world.
>
> Thus in the paper we do not focus on carefully designing procedures for better sim2real, but rather show that, just doing simple policy optimization using diffusion, without any regularization, can transfer well to the real-world. To this end, we have compared DPPO to the same pre-training + fine-tuning pipeline, which is incredibly effective in training policies for challenging manipulation tasks, but using Gaussian policy, and showing the benefits (with additional explanations based on structured exploration studied in Sec 6) and improvements of DPPO.
>
> We believe DPPO has laid algorithmic foundations for better sim2real transfer in challenging robot manipulation tasks, and  incorporating DPPO in dedicated sim2real algorithms may lead to strong sim2real performance in diverse tasks.
>
> [1] “Sim-to-real transfer of robotic control with dynamics randomization” Peng et al.
>
> [2] “Rma: Rapid motor adaptation for legged robots” Kumar et al.

---

> > ### Comment · Reviewer_WTdz · 2024-11-26
> >
> > Thanks for your detailed explanation.  Most of my concerns are addressed. I will raise my score from 5 to 6.

---

### Official Review · Reviewer_7Q1R · 2024-11-02

**Soundness:** 2
**Presentation:** 3
**Contribution:** 2
**Rating:** 6
**Confidence:** 4

**Summary:**

This paper proposes a framework (DPPO) to enable fine-tuning pre-trained diffusion-based policies through policy gradient methods. They compare DPPO with rich baselines in common benchmarks. They conclude best practices for DPPO and conduct ablation studies to understand the effectiveness of DPPO.

**Strengths:**

* This paper conducts rich experiments and analysis. They compare DPPO with other fine-tuning RL methods for diffusion policies and other fine-tuning RL policies. They also conduct several ablation studies and visualization to analyze the effectiveness of DPPO.
* This paper is well-written and well-structured. They conclude the best practice for DPPO and the benefits of such framework.

**Weaknesses:**

* The novelty of the paper is limited. Previous works have proposed the idea that considers a denoising process as multi-step MDP. This paper simply integrate this MDP into the environmental MDP. The tricks to adapt PPO for better fine-tuning are also seen in the previous works like GAE except fine-tuning the last few steps.

**Questions:**

* In the original diffusion policy paper, the performance in the transport is nearly 100% success rate, which conflicts with the results in the paper.
* How about the results compared with the baselines that introduce a residual policy?
* How about the results compared with the baselines that use RLPD or Cal-QL for the pre-trained diffusion policies?

---

> ### Author Response · Authors · 2024-11-20
> **Response from the authors**
>
> We thank the reviewer for the feedback and address the concerns raised by the reviewer below. Please also see the revised version of the paper. Since we have added or removed some figures or sections in the revision, we will use the updated numbering below unless noted.
>
> > ***The novelty of the paper is limited.*** Previous works have proposed the idea that considers a denoising process as multi-step MDP. This paper simply integrate this MDP into the environmental MDP. The tricks to adapt PPO for better fine-tuning are also seen in the previous works like GAE except fine-tuning the last few steps.
>
> We have addressed the overall concern about the novelty of our work in the meta response. We reiterate some of the points here to further address the particular concern from the reviewer.
>
> We are open about the fact that Black et al. [1] interpret the denoising process as an MDP for fine-tuning. However, unrolling the denoising process in diffusion policies had not been considered in prior work. This is because unrolling the MDP increases the number of effective MDP steps by a factor of K, and we know that long-horizon RL problems are frequently more challenging. We ourselves thought our approach might not work, and indeed, its performance was rather unremarkable for the first few months of experimentation. Thus, the fact that such a method works well at the end comes as a major surprise to us as authors, and we hope, to the community. This highlights the main contribution of our work, which is the **scientific finding** that this relatively simple method, based on policy optimization, significantly outperforms other proposed methods of which the authors are aware of in challenging robotic tasks. We also provide empirical explanations in Sec. 6 with a series of investigative experiments, and carefully document all our findings in fair comparisons.
>
> It would be helpful for our revision of the manuscript to better understand the reviewer’s comment: “The tricks to adapt PPO for better fine-tuning are also seen in the previous works like GAE except fine-tuning the last few steps.” While we do propose a number of design choices—most of them, diffusion specific (noise clipping, number steps denoised, etc)—we do not focus on modification of the advantage estimation to any significant degree.
>
> We also would like to highlight the necessary changes to the noise schedule, and the scientific findings that diffusion performs **exploration in a fundamentally different fashion** than Gaussian policies in online RL, which had not been studied in any previous work. We have detailed them in the meta-response.
>
> > In the original diffusion policy paper, the performance in the transport is nearly 100% success rate, ***which conflicts with the results in the paper.***
>
> In our work, we used the Multi-Human (MH) dataset from Robomimic, which consists of noisier human demonstrations than the Proficient-Human (PH) dataset alluded to by the reviewer. In [2], the pre-training result with MH dataset in Transport (state input) is about 46%. Furthermore, to showcase the improvement from RL training, we chose to use a smaller model size and earlier checkpoints so RL training starts at a lower success level. We were careful to make relevant choices so that the Diffusion and Gaussian policies exhibited comparable performance after pre-training. This can be seen from the “0 - steps” data points on the plot (and we will mention this in the appendix in future revisions).

---

> ### Author Response · Authors · 2024-11-20
> **Continued**
>
> > How about the results compared with the baselines that introduce a ***residual*** policy?
>
> We point the reviewer to Ankile et al. [3], which recently (after our submission) proposed learning a residual Gaussian MLP policy on top of a pre-trained diffusion policy. It shows comparable results on Furniture-Bench to ours but requires many domain-specific design choices. However, we believe the Gaussian residual setup does not enjoy the structured exploration indicated by our Sec. 6. Considering a diffusion residual policy and fine-tuning with DPPO can be more promising in the long run. This is especially true because the structured exploration in DPPO may be able to take advantage of experiences collected from multiple tasks and diverse environments. We have cited [3] and discussed this point in the extended related work of the appendix.
>
> > How about the results compared with the baselines that use ***RLPD or Cal-QL for the pre-trained diffusion policies?***
>
> We have run additional experiments for RLPD, Cal-QL, and IBRL using pre-trained diffusion policies with PH/MH dataset from Can and Square tasks from Robomimic (Appendix D.1, Fig. 12, screenshot https://imgur.com/a/QDpS1vU). RLPD and Cal-QL still fail to complete the task, similar to using Gaussian policies. IBRL's performance improves compared to using Gaussian especially when action chunk size 4 is used. DPPO achieves similar or better sample efficiency before 1e6 steps, and reaches ~100% success rate afterwards (not shown) while IBRL saturates at lower levels. We believe that using either offline RL objective or Q learning makes learning precise continuous actions needed in Robomimic tasks difficult, regardless of the policy parameterization, corroborating our original finding in Sec. 5.1 when comparing DPPO to often Q-learning-based diffusion RL methods. We have further discussed this in the meta-response and in the revision.
>
> [1] Training Diffusion Models with Reinforcement Learning. Black et al.
>
> [2] Diffusion policy: Visuomotor policy learning via action diffusion. Chi et al.
>
> [3] From Imitation to Refinement – Residual RL for Precise Assembly. Ankile et al.

---

> ### Author Response · Authors · 2024-11-24
> **Follow-up from the authors**
>
> As the rebuttal period is closing, the authors would greatly appreciate it if the reviewer can address our responses to the original review. We believe we have adequately addressed the issues raised especially the one around the novelty (also see the meta response). If the reviewer requires further clarifications or disagrees with some of our responses, please let us know.

---

> > ### Comment · Reviewer_7Q1R · 2024-11-25
> >
> > Thanks for the response. I think the response mostly addresses my concerns. I maintain my positive score.

---

### Official Review · Reviewer_yZhx · 2024-11-04

**Soundness:** 4
**Presentation:** 4
**Contribution:** 2
**Rating:** 8
**Confidence:** 4

**Summary:**

The authors propose a framework for finetuning a pretrained diffusion policy using on-policy RL, specifically PPO, and show that their method improves performance on long-horizon sparse-reward tasks and works on real hardware.

**Strengths:**

- The authors explore the possibility of fine-tuning diffusion-based policies with on-policy RL and demonstrate some surprising results. Specifically, the authors find improved performance over baselines on long-horizon and sparse reward tasks. The authors also demonstrate how their method can leverage environment parallelization to train faster than off-policy RL methods. Given the abundance of GPU-accelerated simulators these days, I believe this to be a major strength as off-policy RL finetuning methods cannot leverage parallelization in the same way, leaving these high-throughput simulators as a relatively untapped resource until now.

- The authors provide extensive comparisons to a variety of other RL finetuning methods, including two of their own novel baselines. The ablations are also thorough.

- The paper is well written and the quality of the presentation is high.

**Weaknesses:**

- Much of section 6, including the first few paragraphs up to and including Figure 8 are redundant and unnecessary. At this point there are a number of diffusion policy papers applied to various tasks as well as the original diffusion policy paper itself that show that diffusion models consistently outperform GMM and Gaussian policies at capturing the modes of a multi-modal distribution, and so this result is no longer surprising or novel. I'd have rather seen this space dedicated to showing more experimental results that were instead put in the appendix, such as the comparisons in Figure 12 to methods that finetune on additional expert data, or perhaps a more in-depth analysis on why DPPO does better on long-horizon sparse-reward tasks. "How does this method compare to just collecting and training (via BC or whatever else) on additional expert data?" was a question that came to mind while I read the results, which is an important one to discuss and one I imagine other readers will have.

- While the technical contribution is fairly good, the novelty of this approach is relatively low. The proposed method follows the more general and somewhat repetitive trend of "take diffusion policy and try it on task X and potentially finetune with method Y" style papers in the robot learning community.

Despite the somewhat low novelty, I believe that the impressive results and the fact that this paper is one of the first, if not the first, diffusion policy finetuning methods capable of leveraging high-throughput simulators is an important contribution to the robotics and ML communities.

**Questions:**

- What intuition, if any, do the authors have on why DPPO does better on long-horizon sparse-reward tasks? Is there something special about finetuning with PPO as opposed to other finetuning strategies that helps?

- In Figure 6, why does DPPO-UNet get outperformed by DPPO-MLP on state-based tasks? And why is DPPO-UNet not present on the pixel-based versions of the task?

- On lines 264-265 the authors state *"We propose clipping the diffusion noise schedule at a larger-than-
standard noise level to encourage exploration and training stability."*. If more noise encourages exploration, would it not be more beneficial to use the variance-exploding line of diffusion models such as EDM instead of the variance-preserving ones?

- In Figure 7 we see positive sim2real transfer from pretraining only, but negative transfer after fine-tuning. Do the authors have some intuition as to why this is the case?

---

> ### Author Response · Authors · 2024-11-20
> **Response from the authors**
>
> We thank the reviewer for the feedback. First, we would like to comment on comparing DPPO to off-policy methods. We agree that on-policy methods can leverage the parallelized simulation better and potentially achieve better wall-clock time than off-policy methods. Nonetheless, we believe the deeper issue with off-policy methods, typically relying on learning the state-action Q function, is its training instability due to the biased actor update using the approximated Q function, which makes it not suitable for fine-tuning challenging robot manipulation tasks with continuous action space and action chunking. Please see further discussion in the meta-response.
>
> Next, we address the individual concerns raised by the reviewer below. Please also see the revised version of the paper. Since we have added or removed some figures or sections in the revision, we will use the updated numbering below unless noted.
>
> > While the technical contribution is fairly good, ***the novelty of this approach is relatively low***. The proposed method follows the more general and somewhat repetitive trend of "take diffusion policy and try it on task X and potentially finetune with method Y" style papers in the robot learning community.
>
> As discussed in the meta-response, we believe that our core contribution is not just the algorithmic novelty of the two-layer MDP allowing effective policy optimization with diffusion but also to demonstrate that an incredibly natural approach to policy optimization, which has been both overlooked in the prior literature and explicitly conjectured to be ineffective [1], in fact drastically outperforms the prior art in challenging control tasks. Given the ubiquity of diffusion policies in today’s robotics pipelines and the eagerness for effective fine-tuning (due to the limitations of learning from demonstration alone), we believe that uncovering the correct **principles** for fine-tuning diffusion is of core and fundamental research interest. We believe this submission takes a meaningful step towards this goal by demonstrating that policy optimization through the denoising MDP is highly successful (despite concerns of horizon increase) and outperforms methods based on Q-learning and weighted regression.
>
> We note that this finding is highly robust across tasks and imagine that it will spur further investigation into the study of diffusion policy as a parametrization for **RL from scratch** (we have preliminary results in App. D.3, Fig. 18).
>
> Another novel contribution is the scientific finding that diffusion induces a **fundamentally different form of exploration** than previously dominant Gaussian policy parameterization; below, we further clarify the importance of such findings in Section 6.
>
> > Much of section 6, including the first few paragraphs up to and including Figure 8 are redundant and unnecessary. At this point there are a number of diffusion policy papers applied to various tasks as well as the original diffusion policy paper itself that show that diffusion models consistently outperform GMM and Gaussian policies at ***capturing the modes of a multi-modal distribution, and so this result is no longer surprising or novel.***
>
> We agree with the reviewer that many previous papers have shown diffusion models can better capture the multi-modality of data distribution compared to Gaussian and GMM. However, our intention of Figure 8 (now Figure 7) is to demonstrate that DPPO exhibits **structured and on-manifold exploration**. We find that even with exploration noise added to the sampled trajectories, diffusion (DPPO) policy stays near the expert data (i.e., **exploring on the expert data manifold**), which we believe leads to faster RL training and more natural behavior after fine-tuning. Gaussian trajectories can deviate significantly away from the expert data (especially in M2). This finding is novel and significant (as pointed out by Reviewer sDz9), and has not been shown in any previous work. We have further clarified the motivation for such studies in Sec. 6.
>
> We postulate that such exploration leads to a significant training speedup of DPPO compared to Gaussian / GMM policies, as shown with Robomimic and Furniture-Bench tasks. We have added an ablation study in Appendix D.7 showing that adding noise over all denoising steps (as diffusion/DPPO does) induces such structured exploration, but adding noise only to the last denoising step does not.

---

> > ### Author Response · Authors · 2024-11-20
> > **Continued**
> >
> > > in-depth analysis on why DPPO does better on long-horizon sparse-reward tasks…What intuition, if any, do the authors have on ***why DPPO does better on long-horizon sparse-reward tasks?*** Is there something ***special about finetuning with PPO*** as opposed to other finetuning strategies that helps?
> >
> > We believe two aspects are contributing to the strong performance of DPPO in fine-tuning long-horizon manipulation tasks.
> >
> > First, we show empirical explanations in Sec. 6 that DPPO engages in **structured and on-manifold exploration** (i.e., exploring near the expert data) --- in long-horizon sparse-reward tasks, the policy has to exploit the pre-training (expert data) so it can explore near the possible solutions. Staying near the expert data, instead of exploring more randomly in the state space as Gaussian does, improves training efficiency. Fig. 8 shows that DPPO can handle training with **longer action chunk size** than Gaussian policies, which is desired for good pre-training performance in long-horizon tasks [2, 3]. Fig. 10 shows that the DPPO policy is more **robust to OOD states and perturbation in dynamics**, which is much desired during training, too, especially in long-horizon tasks with large state and action space.
> >
> > Second, we firmly believe on-policy methods with policy gradient are much more suited to fine-tuning pre-trained long-horizon manipulation policies than off-policy alternatives, often approximating the state-action Q function. Possibly inaccurate Q function leads to biased updates to the actor, which might lead to training collapse as it starts with good pre-training performance but quickly drops to zero success rate (Fig. 4 bottom row), also failing to recover since then due to the sparse-reward setup. Policy gradient methods trade off some sample efficiency for much better training stability, which is why they are extremely popular in RL for robotics [4, 5]. We have highlighted this intuition in the revised appendix.
> >
> > > "How does this method ***compare to just collecting and training (via BC or whatever else) on additional expert data?***" was a question that came to mind while I read the results, which is an important one to discuss and one I imagine other readers will have.
> >
> > We agree with the reviewer that it will be interesting to compare DPPO (RL) with other fine-tuning methods, such as iterative supervised fine-tuning (e.g., based on DAgger). We point the reviewer to [6], where the results have shown RL systematically outperforms DAgger in Furniture-Bench tasks considered in our work (Fig. 1 right in [6]) — RL performance can converge to ~100% while DAgger saturates around 85%. RL training is necessary for achieving very robust behavior as it penalizes the bad behavior (“pushing the likelihood of the bad actions down”), whereas supervised training does not.
> >
> > > In Figure 6, why does ***DPPO-UNet*** get outperformed by DPPO-MLP on state-based tasks? And why is DPPO-UNet not present on the pixel-based versions of the task?
> >
> > Indeed, we often find DPPO-MLP can achieve higher asymptotic performance than DPPO-UNet. While UNet has shown strong pre-training performance [2], for RL training, we hypothesize that the much simpler MLP architecture is easier to fine-tune. UNet shines when its stronger pre-training performance is required or one would like to pre-train and fine-tune with different action chunk sizes.
> >
> > We omitted DPPO-UNet in the pixel-based Robomimic tasks as we focus on comparing DPPO-MLP and Gaussian-MLP, the two most effective methods from state-based tasks. We have now added the results with DPPO-UNet in Fig. 5 and Fig. 20. UNet performs better than MLP in the Square task but converges more slowly in Transport. We believe further tuning of the parameters can make UNet converge faster in Transport.

---

> > > ### Author Response · Authors · 2024-11-20
> > > **Continued**
> > >
> > > > On lines 264-265 the authors state "We propose clipping the diffusion noise schedule at a larger-than-standard noise level to encourage exploration and training stability.". If more noise encourages exploration, would it not be more beneficial to ***use the variance-exploding line of diffusion models such as EDM*** instead of the variance-preserving ones?
> > >
> > > The main issue with the original noise schedule is that the final noise level is too low (e.g., 1e-4), and the final noise level affects the actual amount of noise applied in the environment much more significantly than earlier noise levels. We think this modification is applicable to either DDPM or variance-exploding ones like EDM, where the final noise level is also annealed to a small amount. Nonetheless, we believe exploring different noise schedules or diffusion parameterization for better RL exploration is a very significant future direction. This paper aims to show that a natural modification of existing implementations suffices (albeit, certainly, room for improvement). Moreover, our method is compatible with diffusion policies that are pre-trained on the more standard variance schedule.
> > >
> > > > In Figure 7 we see positive sim2real transfer from pretraining only, but ***negative transfer after fine-tuning***. Do the authors have some intuition as to why this is the case?
> > >
> > > One possible reason could be that, after fine-tuning, the behavior is slightly more susceptible to sim-to-real gap (e.g., slightly less smooth), that it works better in sim than real. We also think that the performance differences may not be significant statistically.
> > >
> > > [1] Learning a Diffusion Model Policy from Rewards via Q-Score Matching. Psenka et al.
> > >
> > > [2] Diffusion policy: Visuomotor policy learning via action diffusion. Chi et al.
> > >
> > > [3] $\pi_0 $: A Vision-Language-Action Flow Model for General Robot Control. Black et al.
> > >
> > > [4] Learning to walk in minutes using massively parallel deep reinforcement learning. Rudin et al.
> > >
> > > [5] Learning dexterous in-hand manipulation. Andrychowicz et al.
> > >
> > > [6] From Imitation to Refinement – Residual RL for Precise Assembly. Ankile et al.

---

> ### Author Response · Authors · 2024-11-24
> **Follow-up from the authors**
>
> As the rebuttal period is closing, the authors would greatly appreciate it if the reviewer can address our responses to the original review. We believe we have adequately addressed the issues raised especially ones around the novelty and comparison to Gaussian parameterization. If the reviewer requires further clarifications or disagrees with some of our responses, please let us know.

---

> > ### Comment · Reviewer_yZhx · 2024-11-25
> > **Response to Authors' Rebuttal**
> >
> > I thank the authors for investing their time in running additional experiments and for their in-depth response to my questions and concerns. The additional experiments overall add value to the paper and highlight the capabilities of this approach. I maintain my belief that the first portion of section 6 is unnecessary and found the authors' response to this concern lacking. The idea of "on manifold exploration" seems like a manufactured result that isn't really novel and detracts from the overall quality of this paper. It is no surprise that diffusion models better capture the trajectory distribution over GMMs or gaussian policies, which means definitionally the initial trajectories before exploring and finetuning with PPO will look more "on manifold". This is not a novel contribution of DPPO, but rather a well established and known result of applying diffusion models to trajectory data. I have seen figures similar to figure 7 in this paper several times now and I am not convinced anything new is happening here. Nonetheless, I believe this method offers a promising alternative to diffusion + off-policy RL finetuning and that the counterintuitive findings that on-policy RL synergizes well with the diffusion models is worth discussing. Thus, I will raise my score to accept.

---

> ### Author Response · Authors · 2024-11-25
> **Response to the reviewer's response**
>
> We thank the reviewer for the positive response and appreciation of newly added experiments and discussions.
>
> We would like to further address the concern about “on-manifold exploration” since we believe it is one of the core findings of our work and represents an area that has not been thoroughly studied in previous research. We are happy to engage in further discussion — we also acknowledge that there may be some misunderstandings or differences in interpretation between the reviewer and the authors.
>
> ---------
>
> If we understand correctly, the reviewer alludes to plots from the original diffusion policy paper [1] (Fig. 4 in the RSS version) or Fig. 3 of the Diffuser paper that show diffusion well captures the multi-modality of the trajectories. Our studies differ that we are plotting **trajectories with exploration noise added** since we are considering online RL instead of BC or offline RL. We think, before our work, it is not immediately clear that, with additional noise added (using higher noise level with diffusion), diffusion trajectories will remain on-manifold or not; our work shows this is indeed the case and the new ablation study shows such feature leads to faster fine-tuning.
>
> We also highlight two additional details:
> - The structured exploration DPPO shows does not solely depend on the capability of diffusion capturing multi-modality. It also benefits from the strong tendency of diffusion to “push” trajectories back towards the data manifold after the exploration noise from the previous environment step makes the trajectory depart from the manifold. Figure 10 demonstrates this effect — when the robot dynamics is perturbed, DPPO trajectories tend to stay near the data manifold, even after fine-tuning.
> - In the BC case with no noise added, the Gaussian policy will be deterministic and the plot will look like a single trajectory. In our plot, it instead exhibits multi-modality since exploration noise is added and the trajectories diverge. Nonetheless, we show that compared to the multi-modal Gaussian, diffusion stays closer to the expert data. To our knowledge, no previous work has studied the comparison between Gaussian and diffusion in capturing online policy trajectories in robotics or RL.
>
> [1] Diffusion policy: Visuomotor policy learning via action diffusion. Chi et al.
>
> [2] Planning with Diffusion for Flexible Behavior Synthesis. Janner et al.
>
> ----------
>
> We hope the reviewer have understood our intentions. We understand that the reviewer may believe that, it is clear and obvious that even with exploration noise added, diffusion will better stay near expert data. We are open to the differing opinion on this contribution.

---

### Author Response · Authors · 2024-11-20
**Meta response from the authors**

We thank all the reviewers for their constructive feedback. To address the concerns, we have made a series of major revisions to the paper summarized below (the section and figure numbering are consistent with the revised version):

- [Experiment] In Sec. 5.2 and Appendix D.1, we have added additional results with the popular Franka Kitchen tasks from D4RL when comparing DPPO to other demo-augmented RL methods. Methods based on Q-learning work better in these tasks as expected, but they perform much worse in Robomimic tasks where higher precision is required.
- [Experiment] In Sec. 5.3, we have added additional results with UNet as the policy head for DPPO in Robomimic tasks with pixel input.
- [Experiment] In Appendix D.1, we have additional results using diffusion policies for RLPD, Cal-QL, and IBRL, showing that the issue with Q learning persists even with diffusion, corroborating our original findings when comparing to Q-learning-based diffusion RL methods in Sec. 5.1.
- [Experiment] In Appendix D.7, we have added an ablation study on structured exploration of DPPO, showing its contribution in faster fine-tuning.
- [Presentation] At the beginning of Sec. 4, we have expanded on the discussion of naively running policy gradient with diffusion and pointing out the experiments using exact likelihood in the appendix, thus better motivating the novel two-layer MDP setup considered in DPPO.
- [Presentation] In Sec. 4 and Appendix A, we have made the distinction between DPPO’s and the policy gradient’s derivation from QSM [1] clear, highlighting the novelty of the DPPO formulation.
- [Presentation] At the beginning of Sec. 5, we have included rationales for considering each set of experiments to better motivate them and make the transition from the approach section to experimental results more coherent.
- [Presentation] In Appendix C, we have included pseudocode of DPPO and expanded the discussion on our GAE-based advantage estimation.
- [Presentation] To save space, we have moved the paragraph “Potential Impacts beyond Robotics” in the introduction to Appendix A, removed the original Fig. 2 showing the hardware trial, and trimmed discussions at a few places.

Please see the attached revised paper with the changes highlighted in blue. We also would like to address some of the shared concerns from the reviewers here.

## **Novelty of the overall DPPO approach**

We agree that the methodology proposed in this work admits a simple implementation and that it draws on recent ideas in the literature for fine-tuning diffusion models [1, 2]. While the instantiation of DPPO, together with the careful design decisions, is certainly one of our contributions, we argue that our more significant contribution is the **scientific finding** that this relatively simple method, based on policy optimization, **significantly outperforms other proposed methods in challenging robotic tasks**. In other words, our intention is not to combine DP + {an arbitrary RL method} and show that it is performant. Rather, it is to show that, with the right perspective, DP + {one of the simpler and more natural RL methods} matches and often outperformes all of the more complicated alternatives especially in difficult tasks. These contributions rely on a number of conceptual insights that inform the engineering details and which substantially differentiate our approach to those taken in prior works. We also stress that our submission was careful not to overclaim about which aspects were algorithmically novel, and instead emphasize developing a comprehensive scientific understanding of why this particular combination is (to borrow from the messaging of many other papers in the broader literature) “all you need.”

**Finding 1: Performance of DPPO / Surprising effectiveness of a Two-Layer MDP, despite horizon increase:**
Our core finding is that DPPO, a relatively simple instantiation of policy optimization, drastically outperforms existing methods. We are open about the fact that Black et al. [1] interpret the denoising process as an MDP for fine-tuning image diffusion models. However, unrolling the denoising process in diffusion policies had not been considered in prior work. This is because unrolling the MDP increases the number of effective MDP steps by a factor of K, and we know that long-horizon RL problems are frequently more challenging. This led to other methods exploiting techniques either from offline RL [2] or ones inspired by diffusion guidance methods [3]. We also note that the authors of QSM tried a similar approach to ours but based on REINFORCE, and found it to have low performance. We ourselves thought our approach might not work, and indeed, its performance was rather unremarkable for the first few months of experimentation. Thus, the fact that such a method indeed is the **most performant** comes as a major surprise to us as authors, and we hope, to the community.

---

> ### Author Response · Authors · 2024-11-20
> **Continued**
>
> **Finding 2: Diffusion induces a novel form of exploration:** Although PPO had been studied for fine-tuning diffusion models in image generation [1], making it perform well in control tasks requires fundamentally different perspectives. The first is that exploration noise becomes much more important in the data-poor robotics regime than in images, where exploration is rather de-emphasized (e.g., [1] did not discuss the importance of the noise schedule). Indeed, we can think of PPO in an image diffusion model as closer to “alignment” than true exploration.
>
> We show that diffusion policy, due in part to the intrinsic noise in the denoising process, and to our algorithmic modifications, which add additional noise by clipping the noise schedule, induces a form of exploration that is quite unlike what we see with typical Gaussian or GMM policies (Sec. 6, Fig. 7). Diffusion generates exploration trajectories that are **structured and near the expert data manifold**. One intuition is that, unlike Gaussian policy that adds noise only to the final sampled action, diffusion adds multiple rounds of noise through denoising. Each denoising step expands the coverage with new noise while also pushing the newly denoised action towards the expert data manifold [10].
>
> As suggested by Reviewer sDz9, we have included additional ablation experiments (Appendix D.7, https://imgur.com/a/2TUw5yw) to pinpoint the source of this on-manifold exploration — we find that such exploration only emerges when noise is added across denoising steps, instead of merely at the last step as it is done with Gaussian policies. We also stress that backpropagation through the likelihoods, which corresponds to collapsing all steps to a single one and treating the diffusion model as a flow, is highly unstable and non-performant (Appendix D.6).
>
> We also would like to highlight further interpretation of this “on-manifold exploration” in the context of other literature: we can think of the K-steps of diffusion as a variant of a K-layer network, and thus, one interpretation of our method is that of adding noise to the hidden representation of the network. This draws connections to other works that have demonstrated that adding noise to the internal representations leads to control over a greater variety of generations at varying resolutions ([4] StyleGan) and introduces more effective and consistent exploration for solving MuJoCo tasks ([5], OpenAI). Thus, our work suggests that this approach may unlock new paradigms for effective exploration that are well-fitted with expressive action parameterizations, such as diffusion and flow matching, which have been gaining popularity [6, 7]. Indeed, the possibility for on-manifold exploration, guided by model priors, seems rather unexplored in continuous control RL.
>
> Finally, we note that our experiments on BC show that the advantages of DPPO pertain to the performance after RL finetuning, as we do our utmost to match the BC pre-training performance of DPPO and Gaussian policies by selecting checkpoints. Thus, the efficacy of DPPO is shown to be a feature of the synergistic effects between diffusion and policy optimization rather than an added benefit to using a more expressive policy parameterization alone.

---

> ### Author Response · Authors · 2024-11-20
> **Continued**
>
> **Finding 3: Policy gradient (but not Q-learning) succeeds for challenging, high-precision tasks:** The dominant paradigm for robotic fine-tuning was based on learning state-action Q functions, either borrowing ideas from offline/off-policy RL [2, 3] or by appealing to ideas from classifier guidance [3]. Our key insight is that policy gradient methods can be more successful for high-dimensional, continuous action spaces. In the revision, we have included a new set of experiments on the Franka Kitchen tasks from D4RL, where we indeed find Q-learning approaches to be comparatively more successful. However, on Robomimic tasks where high precision is needed, DPPO using policy gradients is much more performant. In Can and Square from Robomimic, the policy needs to precisely grasp and place the objects, while in Franka Kitchen, the requirement is not as high—the robot can open the microwave, move the kettle, turn on the switch, and open the cabinet (the four subtasks) without precise control of the gripper opening. As Q learning backpropagates the gradient of the biased (inaccurate) Q function to the actor, the training can be rather unstable [8]. We also conjecture that the stability of policy gradients pertains to randomized smoothing in optimizing through contact-rich dynamics [9], and this hypothesis is in part corroborated by the ineffectiveness of backpropagating through likelihoods (a first-order method) rather than our use of the REINFORCE trick (a zeroth-order method) via PPO.
>
> On the other hand, we acknowledge that Q-learning methods exhibit favorable sample efficiency when effective; combining the best of both strategies is an exciting direction for future study. For the present submission, however, we want to isolate the finding that policy optimization alone provides tremendous performance improvements in challenging tasks.
>
> ## **Why compare to Gaussian parameterizations in Sec. 5?**
> We agree with the reviewers that the benefits of Gaussian policies over Diffusion policies are well-documented for BC pre-training and that it is widely appreciated that Diffusion can represent richer, multi-modal distributions over actions.
>
> Our comparison to Gaussian policies is to determine (1) whether these benefits translate to RL, but more importantly, (2) if there are any synergistic effects when conducting online RL through a Diffusion parameterization, which has not been studied in any prior work. We have included this rationale/framing in the revision.
>
> All comparisons to Gaussian policies control for the already-known differences between policy parameterization in pre-training by ensuring that their success rates on BC pre-training are comparable (this is not exact, but we have carefully chosen pre-training checkpoints to control this). Therefore, any performance improvements are due to the unique advantages of **diffusion as a policy parametrization for online RL fine-tuning** rather than its ability to fit multi-modal demonstration data. At least for us authors, these surplus improvements of diffusion over Gaussian in RL fine-tuning were initially very surprising, especially since DPPO increases the effective MDP horizon by a factor of the number of denoising steps (making optimization more challenging).
>
> The improvement of diffusion over Gaussian in benchmarking results also led us to explore both parameterizations in investigative experiments (Sec. 6), and we find that the advantages conferred by DPPO over other parametrizations support the hypothesis that diffusion and RL have synergistic, rather than merely additive, benefits. For example, DPPO (using our designed noise-exploration schedule) **induces a fundamentally different form of exploration** than previously dominant Gaussian parameterization, exploring in **structured and on-manifold** manners near the expert data. This is irrelevant during BC and is a crucial capability of diffusion in RL settings. Finally, we demonstrate the gaps between DPPO and Gaussian in sim-to-real transfer. This, too, is a finding that is not implied by any past work.
>
> **To summarize, we demonstrate the advantages of DPPO over other policy parameterizations for RL fine-tuning settings. These advantages imply the benefits of the DP parameterization beyond what has already been observed in BC and had not been studied before.** We have emphasized this point in the revision.

---

> > ### Author Response · Authors · 2024-11-20
> > **Continued**
> >
> > ## **Overall presentation**
> > The reviewers had mixed comments about the paper's presentation—Reviewer yZhx, 7Q1R, and WTdz gave scores of 4, 3, and 3, while Reviewer sDz9 considered the presentation poor and gave a score of 1. We acknowledge Reviewer sDz9's suggestion that the overall presentation can be more coherent. Based on the existing comments, we highlight two main changes already made below. We also invite any other concrete suggestions on improving our exposition further.
> >  - We acknowledge that the approach section can better motivate why considering the two-level MDP is necessary. At the beginning of Sec. 4, we expanded on the discussion of naively running policy gradient with diffusion and pointing out the experiments using exact likelihood in the appendix.
> > - We acknowledge that the current manuscript lacks a clear rationale for considering each set of experiments in Sec. 5. In the revision, we include such rationales at the beginning of Sec. 5 so that the motivation of the experiments is made clear.
> >
> > [1] Training Diffusion Models with Reinforcement Learning. Black et al.
> >
> > [2] IDQL: Implicit Q-Learning as an Actor-Critic Method with Diffusion Policies. Hansen-Estruch et al.
> >
> > [3] Learning a Diffusion Model Policy from Rewards via Q-Score Matching. Psenka et al.
> >
> > [4] A Style-Based Generator Architecture for Generative Adversarial Networks. Karras et al.
> >
> > [5] Parameter Space Noise for Exploration. OpenAI.
> >
> > [6] $\pi_0 $: A Vision-Language-Action Flow Model for General Robot Control. Black et al.
> >
> > [7] Diffusion policy: Visuomotor policy learning via action diffusion. Chi et al.
> >
> > [8] Interpolated policy gradient: Merging on-policy and off-policy gradient estimation for deep reinforcement learning. Gu et al.
> >
> > [9] Do Differentiable Simulators Give Better Policy Gradients? Suh et al.
> >
> > [10] Interpreting and Improving Diffusion Models from an Optimization Perspective. Permenter et al.

---

### Meta-Review · Area_Chair_AEWe · 2024-12-21

**Metareview:**

This paper introduces a technique for fine-tuning diffusion policies with a policy gradient method. Previously, policy gradient updates of diffusion policies were conjectured to have training instability. With the new technique proposed, the authors show that the resulting algorithm called DPPO can fine-tune to superior policies both in simulations and real robots. Extensive comparison is provided. This paper has the potential to be adopted by many papers in the future and as such I recommend this paper for a spotlight presentation.

Minor comments:

> using the policy gradient (PG) method

Change to “Using a policy gradient method”

> As the first glance

Change to “At first glance”

**Additional Comments On Reviewer Discussion:**

Reviewers appreciated the broad range of experiments, results, and analysis. Authors also significantly improved the writing during the rebuttal period, addressing reviewers’ concerns.

---

### Decision · Program_Chairs · 2025-01-22

Accept (Poster)